# Endocytic recycling is central to circadian collagen fibrillogenesis and disrupted in fibrosis

Joan Chang[1,2]*, Adam Pickard[1], Jeremy A Herrera[1], Sarah O'Keefe[2], Richa Garva[1], Matthew Hartshorn[1], Anna Hoyle[1], Lewis Dingle[3], John Knox[1,2], Thomas A Jowitt[1], Madeleine Coy[2], Jason Wong[3], Adam Reid[3], Yinhui Lu[1], Cédric Zeltz[4], Rajamiyer V Venkateswaran[5], Patrick T Caswell[1], Stephen High[2], Donald Gullberg[4], Karl E Kadler[1]*

[1]Wellcome Centre for Cell-Matrix Research, Faculty of Biology, Medicine and Health, University of Manchester, Manchester, United Kingdom; [2]Division of Molecular and Cellular Function, Faculty of Biology, Medicine and Health, University of Manchester, Manchester, United Kingdom; [3]Blond McIndoe Laboratories, Faculty of Biology, Medicine and Health, University of Manchester, Manchester, United Kingdom; [4]Department of Biomedicine and Centre for Cancer Biomarkers, Norwegian Center of Excellence, University of Bergen, Bergen, Norway; [5]Manchester University National Health Service Foundation Trust, Manchester Academic Health Science Centre, Manchester, United Kingdom

*For correspondence:
joan.chang@manchester.ac.uk
(JC);
karl.kadler@manchester.ac.uk
(KEK)

## eLife Assessment

This study describes a novel mechanism for how collagen fibrils are formed. The authors present **compelling** evidence that collagen-I fibrillogenesis relies on a functional endocytic system for recycling collagen-I, with circadian-regulated VPS33b and integrin-α11 being critical for fibril assembly. This is an **important** study for the understanding of the pathophysiology of collagen fibrillogenesis.

**Abstract** Collagen-I fibrillogenesis is crucial to health and development, where dysregulation is a hallmark of fibroproliferative diseases. Here, we show that collagen-I fibril assembly required a functional endocytic system that recycles collagen-I to assemble new fibrils. Endogenous collagen production was not required for fibrillogenesis if exogenous collagen was available, but the circadian-regulated vacuolar protein sorting (VPS) 33b and collagen-binding integrin α11 subunit were crucial to fibrillogenesis. Cells lacking VPS33B secrete soluble collagen-I protomers but were deficient in fibril formation, thus secretion and assembly are separately controlled. Overexpression of VPS33B led to loss of fibril rhythmicity and overabundance of fibrils, which was mediated through integrin α11β1. Endocytic recycling of collagen-I was enhanced in human fibroblasts isolated from idiopathic pulmonary fibrosis, where VPS33B and integrin α11 subunit were overexpressed at the fibrogenic front; this correlation between VPS33B, integrin α11 subunit, and abnormal collagen deposition was also observed in samples from patients with chronic skin wounds. In conclusion, our study showed that circadian-regulated endocytic recycling is central to homeostatic assembly of collagen fibrils and is disrupted in diseases.

## Introduction

Collagen fibrils account for ~25% of total body protein mass (*Smejkal and Fitzgerald, 2017*) and are the largest protein polymers in vertebrates (*Craig et al., 1989*). The fibrils can exceed centimeters in length and are organized into elaborate networks to provide structural support for cells. It is unclear how the fibrils are formed and how this process goes awry in collagen pathologies, such as fibrosis. A key mechanism, cell surface-mediated fibrillogenesis, has been suggested to occur via indirect binding to fibronectin fibrils, or via direct assembly by collagen-binding integrins (*Kadler et al., 2008*; *Musiime et al., 2021*; *Zeltz et al., 2023*). Interestingly, fibrils can be reconstituted in vitro from purified collagen (reviewed in *Kadler et al., 1996*) but the assembly process is not controlled to the extent seen in vivo in terms of number, size, and organization; this suggests a much tighter cellular control over the process. Additional support for tight cellular control of collagen fibril formation comes from electron microscope observations of collagen fibrils at the plasma membrane of embryonic avian and rodent tendon fibroblasts (*Trelstad and Hayashi, 1979*; *Birk and Trelstad, 1986*), where the end (tip) of a fibril is enclosed within a plasma membrane invagination termed a fibripositor (*Canty et al., 2004*). Previously, we identified vacuolar protein sorting (VPS) 33B (a regulator of SNARE-dependent membrane fusion in the endocytic pathway) as a circadian clock-regulated protein involved in collagen homeostasis (*Chang et al., 2020*). VPS33B forms a protein complex with VIPAS39 (VIPAR) (*Hunter et al., 2018*), and mutations in the *VPS33B* gene cause arthrogryposis-renal dysfunction-cholestasis syndrome (*Gissen et al., 2004*). Here, death usually occurs within the first year of birth, accompanied with renal insufficiency, jaundice, multiple congenital anomalies, and predisposition to infection (*Nezelof et al., 1979*). One proposed disease-causing mechanism is abnormal post-Golgi trafficking of lysyl hydroxylase 3 (LH3, *PLOD3*), which catalyzes the hydroxylation of lysyl residues in collagen to form hydroxylysine residues. It also has hydroxylysyl galactosyltransferase and galactosylhydroxylysyl glucosyltransferase activities, which creates attachment sites for carbohydrates that is crucial in stabilizing intramolecular and intermolecular crosslinks within the collagen structure, and thus is essential for collagen homeostasis during development (*Banushi et al., 2016*; *Salo et al., 2006*). All these observations point to a central role for the endosome, situated in proximity of the plasma membrane and acting as a hub for a complex assortment of vesicles, in sorting collagen molecules to different fates.

Previous studies of collagen endocytosis have focused on collagen degradation and signaling, identifying additional collagen-binding proteins such as Endo180 and MRC1 (*Madsen et al., 2007*; *Madsen et al., 2013*; *Madsen et al., 2017*; *Rainero, 2016*; *Lee et al., 2014*), as well as a role for non-integrin collagen receptors involved in mediating signaling activities such as DDR1/DDR2 (reviewed in *Leitinger, 2014*). Here, we showed that collagen uptake by fibroblasts is circadian rhythmic. Importantly, instead of degrading endocytosed collagen, cells utilize endocytic recycling of exogenous collagen-I to assemble new fibrils, even in the absence of endogenous collagen production. Further, we showed that the secretion of soluble collagen protomers is separate from, and can occur independently to, collagen fibril assembly. We identified VPS33B and integrin α11 subunit (part of the collagen-binding integrin α11β1 heterodimer [*Musiime et al., 2021*; *Tiger et al., 2001*; *Zeltz and Gullberg, 2016*]) as central molecules specific to fibril formation. Finally, we showed that in idiopathic pulmonary fibrosis (IPF), a life-threatening disease with unknown trigger/mechanism where lung tissue is replaced with collagen fibrils, integrin α11 subunit, and VPS33B are located at the invasive fibroblastic focus where collagen fibril rapidly accumulates (*Herrera et al., 2019*); IPF fibroblasts also have elevated endocytic recycling of exogenous collagen-I, despite having similar collagen-I expression level to normal fibroblasts. Together, these results provide novel insights into the importance of the circadian clock-regulated endosomal system in normal fibrous tissue homeostasis, where fibrillogenesis occurs with both endogenous collagen and/or recycled scavenged exogenous collagen. This work also highlights that collagen utilization, rather than production (i.e. assembled into fibrils vs. protomeric secretion), is central to the maintenance of homeostasis, and dysregulation leads to disease.

## Results

### Collagen-I is taken up into punctate structures within the cell, and reassembled into fibrils

To study collagen-I endocytosis and the fate of endocytosed collagen, we made Cy3- or Cy5-labeled collagen-I (Cy3-colI, Cy5-colI) from commercial rat tail collagen (*Doyle, 2018*). We confirmed the helicity of these collagens with an expected molecular weight corresponding to a heterotrimeric mature collagen-I without the propeptides, and showed that the process of fluorescence labeling did not alter the collagen trimer secondary structure or stability (*Figure 1—figure supplements 1–3*). We incubated Cy3-colI with explanted murine tail tendons for 3 days, then imaged the core of the tendon. Cells in tendon (nuclei marked with Hoechst stain) showed a clear uptake of Cy3-colI (*Figure 1A*, yellow box, indicated by yellow arrowheads). Interestingly, fibrillar Cy3 fluorescence signals that extend across cells were also observed, suggestive of these being extracellular fibrils (*Figure 1A*, gray box, indicated by white arrows). Pulse-chase experiments were performed where Cy3-colI was added to the tendons for 3 days, and then removed from the media. This was followed by 5FAM-colI for a further 2 days (*Figure 1—figure supplement 4*). The results showed distinct areas where only Cy3-colI was observed (*Figure 1—figure supplement 4*, yellow box); this persistence indicated that not all collagen endocytosed by cells is directed for degradation. 5FAM-positive and 5FAM/Cy3-positive fibril-like structures were also observed, confirming that collagen-I taken up by tissues is reincorporated into the matrix (*Figure 1—figure supplement 4*, gray boxes). To confirm these findings in 2D cultures, we incubated immortalized murine tail tendon fibroblast cultures (iTTFs) with labeled Cy3-colI. Flow cytometry analysis revealed that most cells had taken up Cy3-colI after overnight incubation (*Figure 1—figure supplement 5*). Further time course analyses revealed a time-dependent and concentration-dependent increase of collagen-I uptake (*Figure 1B*). Cells incubated with Cy3-ColI for 1 hr followed by trypsinization were then analyzed using flow cytometry imaging (flow imaging), revealing that collagen-I is endocytosed by the cells into distinct puncta (*Figure 1C*). Similar results were obtained using Cy5-labeled collagen-I, indicating that the fluorescence label itself did not alter cellular response (*Figure 1—figure supplements 6–8*). To confirm that the labeled collagen-I is in fact endocytosed into the fibroblasts and not only associated with the cell surface when cells are still attached, we performed live imaging on cells incubated with Cy3-colI. Time lapse images showed Cy3-colI congregate in structures within the cell, as indicated with white arrows (*Figure 1—figure supplement 9*). Additionally, we transduced iTTF with GFP-tagged Rab5 and confirmed co-localization of Cy5-colI with Rab5-positive intracellular structures (*Figure 1D*, yellow arrows). We then incubated iTTFs with Cy3-colI for 1 hr, before trypsinizing and replating the cells to ensure any Cy3-colI signal detected hereon originated from the endocytosed pool. Immunofluorescence (IF) staining indicated co-localization of endogenous collagen-I with Cy3-labeled collagen-I in extracellular fibrillar structures after 72 hr, compared to a striking lack of extracellular collagen-I (endogenous or labeled) at 18 hr (*Figure 1E* and *Figure 1—figure supplement 9*). These results demonstrated that exogenous collagen-I can be taken up by cells both in vitro and in tissues ex vivo, and recycled into fibrils.

We considered the possibility that the fluorescent fibril-like structures were the result of spontaneous cell-free fibrillogenesis of the added Cy3-colI. Thus, we incubated Cy3-colI at concentrations spanning the 0.4 µg/mL critical concentration for cell-free in vitro fibril formation (*Kadler et al., 1987*), with or without cells (*Figure 1F*). In cell-free cultures, Cy3-positive fibrils were observed at 5 µg/mL, 1 µg/mL, and 0.5 µg/mL in a dose-dependent manner, with no discernible Cy3-positive fibrils at 0.1 µg/mL Cy3-colI (*Figure 1F*, right panel; *Figure 1G*). However, in the presence of cells, Cy3-positive fibrillar structures were observed at the cell surfaces at all concentrations of Cy3-colI examined, including 0.1 µg/mL (*Figure 1F*, zoom-ins of red box expanded to the left, highlighted by white arrows; *Figure 1H and I*). These results indicated that the cells actively took up and recycled Cy3-colI into extracellular fibrils.

### The endocytic pathway controls collagen-I secretion and fibril assembly

We then investigated the route that collagen-I is taken up. Prior research identified macropinocytosis (*Nazemi et al., 2024*), which can be modeled with high-molecular weight dextran. We hypothesized that there will be co-localization of labeled collagen-I and labeled 70 kDa dextran within the cell if they are entering through the same route, however surprisingly we see little co-localization (*Figure 2—figure supplement 1*). We then performed a receptor saturation experiment, where

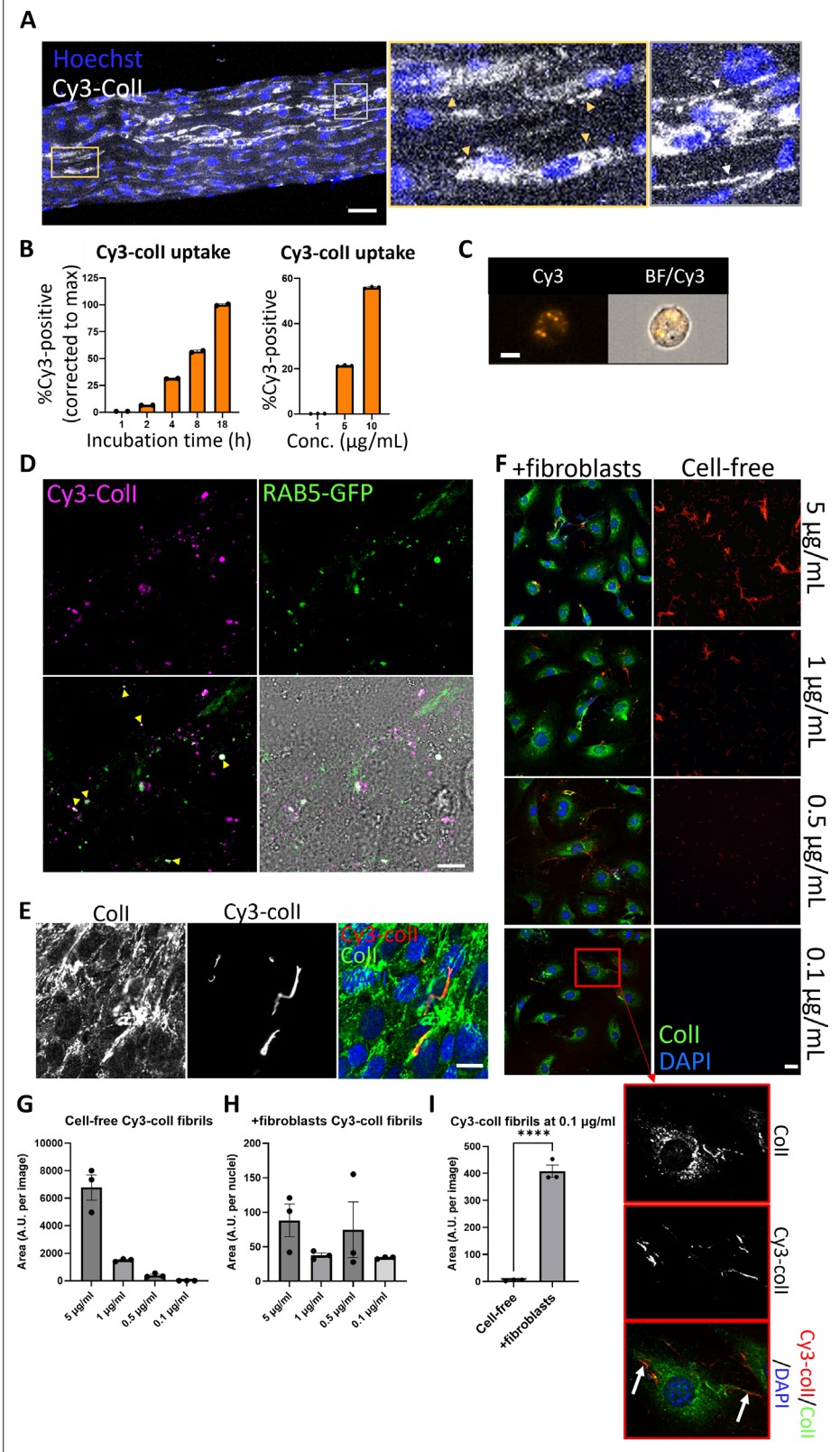

**Figure 1.** Collagen-I is endocytosed and reassembled into fibrils. (**A**) Fluorescent images of tail tendon incubated with Cy3-colI for 5 days, showing the presence of collagen-I within the cells, and fibril-like fluorescence signals outside of cells. Hoechst stain was used to locate cells within the tendon. Area surrounded by yellow box expanded on the right, and cells with Cy3-colI present intracellularly pointed out by yellow triangles. Area

*Figure 1 continued on next page*

*Figure 1 continued*

surrounded by gray box expanded on the right, and fibril-like fluorescence signals indicated with white arrows. Scale bar = 50 µm. Representative of N=3. (**B**) Bar chart showing an increase of percentage of fluorescent iTTFs incubated with 1.5 µg/mL Cy3-colI over time (left), and an increase of percentage of fluorescent iTTFs incubated with increasing concentration of Cy3-colI for 1 hr (right), suggesting a non-linear time-dependent and dose-dependent uptake pattern. N=3. (**C**) Flow cytometry imaging of iTTFs incubated with 5 µg/mL Cy3-labeled collagen-I for 1 hr, showing that collagen-I is taken up by cells and held in vesicular-like structures. Images acquired using ImageStream at ×40 magnification. Scale bar = 10 µm. Cy3 – Cy3 channel, BF/Cy3 – merged image of BF and Cy3. Representative of >500 cells images collected per condition. (**D**) Fluorescent images of iTTFs transduced with Rab5-GFP and incubated with Cy3-labeled collagen-I. Yellow arrows point to labeled collagen co-localizing with Rab5 in intracellular structures. Representative of N=3. Scale bar = 10 µm. (**E**) Fluorescent images of iTTFs incubated with 5 µg/mL Cy3-colI for 1 hr, trypsinized and replated in fresh media, and further cultured for 72 hr. Top labels denote the fluorescence channel corresponding to proteins detected. Merged image color channels as denoted on top left. Representative of N>3. Scale bar = 20 µm. (**F**) Fluorescent image series of Cy3-colI incubated at different concentrations for 72 hr, either cell-free (right panel), or with iTTFs (+fibroblasts, left panel). Representative of N=3. Scale bar = 20 µm. Red box – zoomed out to the bottom left and separated according to fluorescence channel. White arrows highlighting Cy3-positive fibrils assembled by fibroblasts when incubated with 0.1 µg/mL Cy3-colI. (**G**) Quantification of the area of Cy3-positive fibrils in cell-free cultures, quantified per image area. N=3. (**H**) Quantification of the area of Cy3-positive fibrils in +fibroblasts cultures, corrected to number of nuclei per image area. N=3. (**I**) Comparison of total area of Cy3-positive fibrils in cell-free and +fibroblast cultures at 0.1 µg/mL concentration, as quantified per image area. N=3. ****p<0.0001.

The online version of this article includes the following figure supplement(s) for figure 1:

**Figure supplement 1.** Circular dichroism spectra of unlabeled (black), Cy3-labeled (orange, Cy3-colI), and Cy5-labeled (blue, Cy5-colI) collagen-I in acetic acid showing the helical positive peak at 223 nm in each of the spectra.

**Figure supplement 2.** Mass photometry of unlabeled (black), Cy3-labeled (orange, Cy3-colI), and Cy5-labeled (blue, Cy5-colI) collagen-I in acetic acid.

**Figure supplement 3.** Temperature-induced thermal unfolding of collagen-I monitored at 223 nm.

**Figure supplement 4.** Fluorescent images of tail tendon either not incubated with fluorescently labeled collagen-I (control, bottom), or incubated with fluorescently labeled collagen-I for 5 days – Cy3-colI as added for the first 3 days, removed, and then with 5FAM-labeled collagen-I (FAM-colI) added in the last 2 days.

**Figure supplement 5.** Representative dot plots from flow cytometry analysis representing Cy3 gates used in control and iTTFs incubated with 1 µg/mL Cy3-labeled collagen (Cy3-colI) for 18 hr.

**Figure supplement 6.** Representative dot plots from flow cytometry analysis representing Cy5 gates used in control and iTTFs incubated with 1 µg/mL Cy5-labeled collagen (Cy5-colI) for 18 hr.

**Figure supplement 7.** Bar chart showing a progressive increase of percentage of fluorescent iTTFs incubated with 1.5 µg/mL Cy5-colI over time (left), and an increase of percentage fluorescent iTTFs incubated with increasing concentration of Cy5-colI for 1 hr (right), suggesting a non-linear time-dependent and dose-dependent uptake pattern.

**Figure supplement 8.** Flow cytometry imaging of iTTFs incubated with 5 µg/mL Cy5-labeled collagen-I for 1 hr, showing that collagen-I is taken up by cells and held in vesicular structures.

**Figure supplement 9.** Fluorescent image series of iTTFs incubated with Cy3-labeled collagen-I at 0 min (t=0 min), 69 min (t=69 min), and 107 min (t=107 min) after addition (rows) and viewed from different angles (columns), with cell mask (cyan) to distinguish the cell volume.

**Figure supplement 10.** iTTFs were incubated with Cy3-colI (magenta) for 1 hr, trypsinized, replated, and allowed to stick down for 18 hr, before being fixed and stained with collagen-I (ColI, green) and counterstained with DAPI (blue).

unlabeled collagen-I was added in excess to labeled collagen-I (10:1) during the 1 hr incubation, before flow imaging analysis of the fibroblasts. Flow imaging showed that labeled collagen was at the cell periphery and not intracellular when saturation occurs, suggestive of a receptor-mediated process (*Figure 2—figure supplement 2*). Thus, we used Dyngo4a (*McCluskey et al., 2013*) to inhibit clathrin-mediated endocytosis, where Dyngo4a treatment (20 µM) leads to ~60% reduction in Cy3-colI uptake relative to control (*Figure 2—figure supplement 3*), without affecting cell viability (*Figure 2—figure supplement 4*). We then investigated the effects of Dyngo4a treatment on the ability of wild-type (WT) fibroblasts to assemble collagen fibrils. Cells were treated with Dyngo4a for 48 hr before fixation and IF against collagen-I antibody. A significant reduction (average 60%) in the number of collagen-I

fibrils assembled was observed (*Figure 2A*), indicative of endocytosis playing a key role in collagen-I fibrillogenesis. Conditioned media (CM) from these treated cells were also collected and assessed. To our surprise, the amount of soluble collagen-I secreted was also greatly reduced, with minimal impact on total secretion (*Figure 2B*, *Figure 2—figure supplement 5*). Quantitative PCR (qPCR) analyses revealed that *Col1a1* mRNA was significantly reduced in Dyngo4a-treated cells (*Figure 2—figure supplement 6*), suggesting a potential feedback mechanism between endocytosis and collagen-I synthesis, secretion, and fibrillogenesis. Interestingly, while fibronectin (FN1) mRNA was significantly lower in Dyngo4a-treated cells (*Figure 2—figure supplement 6*), and the intensity of FN1 signal appears lower, the amount of FN1 fibrils deposited (as determined by the area occupied by fibrils) was not significantly impacted (*Figure 2C*). Taken together, these results showed that inhibiting endocytosis in fibroblasts does not lead to accumulation of soluble collagen-I protomers in the extracellular space. Rather, endocytosis impacts collagen fibrillogenesis and the transcriptional control on collagen-I and fibronectin.

## Collagen-I endocytosis is circadian clock regulated, and recycling alone can generate fibrils

Previously, we have shown that clock-synchronized fibroblasts synthesize collagen fibrils in a circadian rhythmic manner (*Chang et al., 2020*), the results here thus far indicate an involvement of endocytosis in collagen fibrillogenesis. Therefore, we hypothesized that collagen-I endocytosis may also be circadian clock regulated. Time-series flow cytometry analyses of fibroblasts incubated with Cy3-colI revealed that the level of Cy3-colI endocytosed by the cells is rhythmic, with a periodicity of 23.8 hr as determined by MetaCycle analysis (*Figure 2D*). When corrected to the running average, the rhythmic nature of Cy3-colI uptake is accentuated (*Figure 2—figure supplement 7*). We noted that, when compared to the number of fibrils produced over time, the peak time of uptake happens before peak fibril numbers (*Figure 2E*). These data suggest that the cells may be endocytosing exogenous collagen under circadian control and holding it in the endosomal compartment, then trafficking the collagen to the plasma membrane for fibril formation.

To eliminate the possibility that the fluorescent fibril-like structures were due to attachment of fluorescently labeled collagen protomers to pre-existing fibrils already deposited by cells, fibrils already deposited by cells, we performed an siRNA-mediated knockdown against *Col1a1* (siCol1a1) to target endogenous collagen production. Fibroblasts were then analyzed by IF using anti-collagen-I or anti-fibronectin (FN1) antibodies. Control (scrambled siRNAs, scr) cells synthesized collagen-I and fibronectin (*Figure 3A*, top row) with defined fibrillar structures. In contrast, cells treated with siCol1a1 synthesized lower levels of collagen-I, with significantly few collagen-I fibrillar structures as well as lower intracellular collagen-I signal (*Figure 3A*, bottom row; quantification plots). An ~90% reduction in *Col1a1* mRNA was confirmed using qPCR (*Figure 3—figure supplement 1*).

Flow cytometry analysis of cells incubated with Cy3-colI revealed that siCol1a1 fibroblasts retained the ability to endocytose exogenous collagen (*Figure 3—figure supplement 2*). To assess if siCol1a1 fibroblasts can assemble a collagen fibril with exogenous collagen, Cy3-colI was incubated with scr or siCol1a1 cells for 1 hr, before trypsinization and replating to ensure no cell surface-associated Cy3-colI remains. Within 3 days, Cy3-positive collagen fibrils were observed in both scr (*Figure 3B*, left column) and siCol1a1 cells (*Figure 3B*, right column). The observation of fibrillar Cy3 signals in siCol1a1 cells showed that the cells can repurpose collagen into fibrils without the requirement for intrinsic collagen-I production (*Figure 3B*, red arrow). The cells were also probed for collagen-I using an anti-collagen-I antibody with a secondary antibody conjugated to Alexa Fluor 488 (*Figure 3B*, second row). In scr cells we detected multiple fibrillar structures that did not contain Cy3-colI (*Figure 3B*, white arrows), indicating fibrils derived from endogenous collagen-I. In the siCol1a1 cells we detected a more diffuse signal and puncta within the cell, but no discernible Alexa Fluor 488-only collagen-I fibrils (*Figure 3B*). These results indicate that fibroblasts can assemble collagen-I fibrils from recycled exogenous collagen-I. Importantly, siCol1a1 cells treated with Dyngo4a could not effectively form Cy3-collagen-containing fibrils under these conditions, indicative of a requirement for endocytic recycling of exogenous collagen-I for fibrillogenesis, when endogenous levels are insufficient (*Figure 3C*). To confirm these findings, we isolated primary tendon fibroblasts from the Col1a2-CreERT2::Col1a1-fl/fl (termed CKO) mouse that had either been treated with tamoxifen (CKO+) to produce fibroblasts that cannot synthesize collagen-I or from untreated mice to yield matched controls (CKO-). Fibroblasts

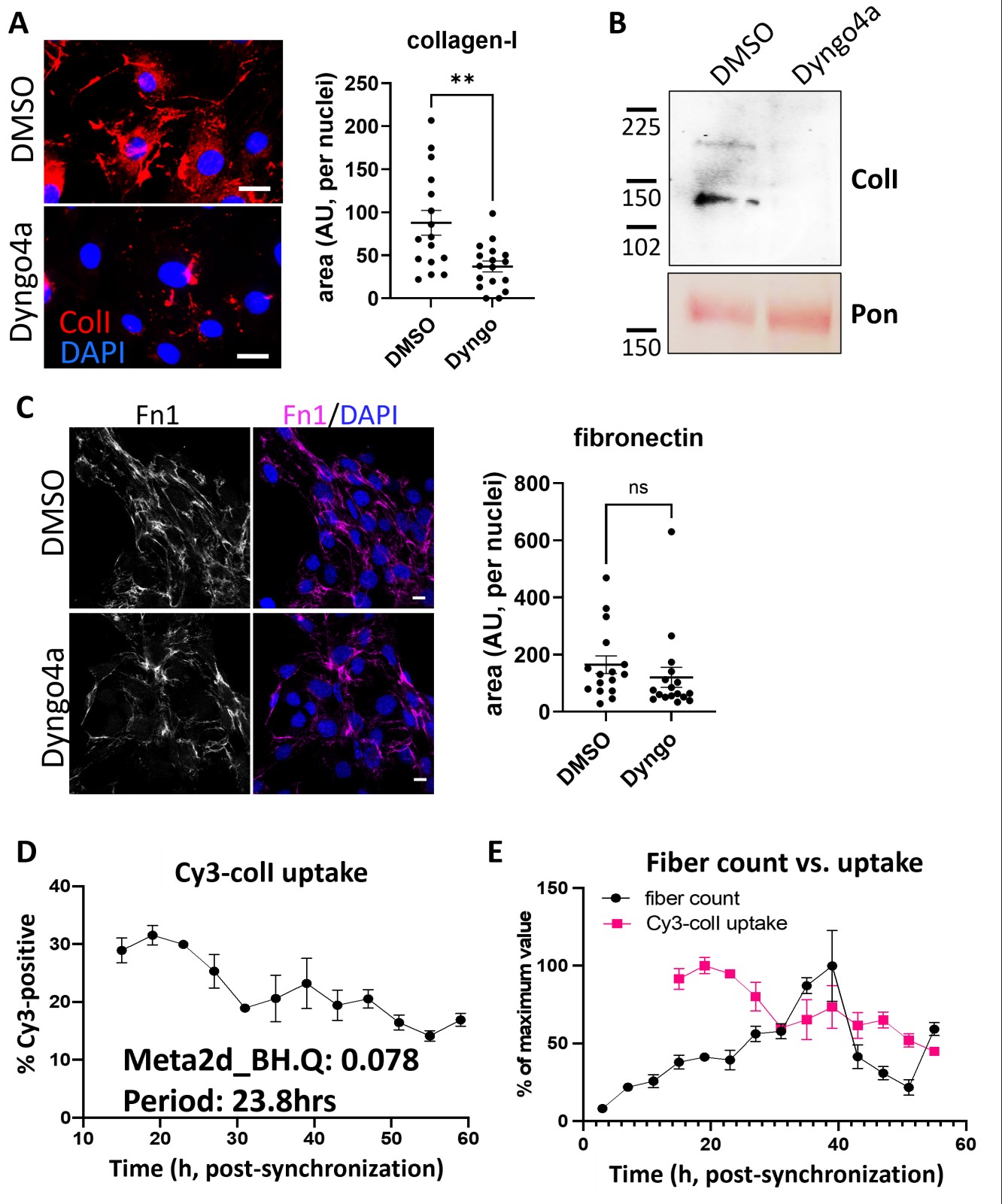

**Figure 2.** Inhibition of endocytosis leads to changes in collagen-I homeostasis, and endocytosis is a rhythmic event. (**A**) Left: fluorescent images of collagen-I (red) counterstained with DAPI (blue) in iTTFs treated with DMSO (top) or Dyng4a (bottom) for 72 hr. Scale bar = 20 μm. Right: quantification of area occupied by collagen-I fibrils, corrected to number of nuclei. N=3 with 5 images from each experiment **p=0.0025. (**B**) Western blot analysis of conditioned media taken from iTTFs treated with DMSO or Dyng4a for 72 hr, showing a decrease in collagen-I secretion. Top: probed with collagen-I

*Figure 2 continued on next page*

*Figure 2 continued*

antibody (Col-I), bottom: counterstained with Ponceau (Pon) as control. Protein molecular weight ladders to the left (in kDa). Representative of N=3. (**C**) Left: fluorescent images of fibronectin (magenta) counterstained with DAPI (blue) in iTTFs treated with DMSO (top) or Dyng4a (bottom) for 72 hr. Scale bar = 20 µm. Right: quantification of area occupied by fibronectin fibrils, corrected to number of nuclei. N=3 with 5 images from each experiment. (**D**) Percentage Cy3-colI taken up by synchronized iTTFs over 48 hr. Meta2d analysis indicates a circadian rhythm of periodicity of 23.8 hr. Bars show mean ± s.e.m. of N=3 per time point. (**E**) Percentage of Cy3-colI taken up by synchronized iTTFs, corrected to the maximum percentage uptake of the time course (pink, bars show mean ± s.e.m. of N=3 per time point), compared to the percentage collagen fibril count over time, corrected to the maximum percentage fibril count of the time course (black, fibrils scored by two independent investigators. Bars show mean ± s.e.m. of N=2 with n=6 repeats at each time point).

The online version of this article includes the following figure supplement(s) for figure 2:

**Figure supplement 1.** Representative images from flow imaging of iTTFs either incubated with Cy5-colI only (top) or both Alexa Fluor 488-labeled 70 kDa dextran and Cy5-colI (bottom), showing very little co-localization between the two markers after being taken up into cells.

**Figure supplement 2.** Representative images from flow imaging of iTTFs either incubated with Cy5-colI only (top) or Cy5-colI and unlabeled colI (bottom), showing that in the presence of excess unlabeled collagen, the majority of the Cy5 signal are restricted to the periphery of the cells.

**Figure supplement 3.** Scatter plot showing Dyngo4a (Dyng), an endocytosis inhibitor, treatment for 1 hr inhibits over 50% of Cy3-colI uptake in iTTFs.

**Figure supplement 4.** Alamar Blue assay showing that prolonged treatment of 20 µM Dyngo4a (Dyng) does not inhibit iTTF proliferation.

**Figure supplement 5.** Full scan of Ponceau stain western blot, corresponding to *Figure 2B*.

**Figure supplement 6.** Quantitative PCR (qPCR) analysis of Col1a1 and Fn1 mRNA levels in DMSO and Dyng-treated iTTFs, corrected to DMSO control, showing a decrease in both collagen-I and fibronectin transcripts.

**Figure supplement 7.** Percentage Cy3-colI taken up by synchronized iTTFs over 48 hr.

were then analyzed by IF microscopy using anti-collagen-I antibodies. As expected CKO- cells synthesized collagen-I, and CKO+ cells showed no collagen-I expression (*Figure 4A*). qPCR analysis of the cells confirmed the absence of Col1a1 mRNAs containing exon 6 to exon 8 sequences, as expected from the location of the LoxP sites in the Col1a1 gene (*Figure 4—figure supplement 1*).

Flow analysis of cells incubated with Cy3-colI revealed collagen-knockout fibroblasts retained the ability to endocytose labeled collagen (*Figure 4—figure supplement 2*). To assess whether collagen-knockout fibroblasts were able to assemble a collagen fibril, Cy3-colI was incubated with cells before the cells were released by trypsin and replated, to ensure that Cy3 signals arose only from Cy3-colI endocytosed by cells. Within 3 days, Cy3-positive collagen fibers could be observed in both CKO- (*Figure 4B*, left column) and CKO+ cells (*Figure 4B*, right column). The observation of fibrillar Cy3 signals in CKO+ cells showed that the cells can repurpose collagen into fibrils without the requirement for intrinsic collagen-I (*Figure 4B*, red arrows). The cells were also probed for collagen-I using an anti-collagen-I antibody with a secondary antibody conjugated to Alexa Fluor 488 (*Figure 4B*, second row). In CKO- cells we detected a diffuse signal over the entire cell and an indication of fibrillar structures. In the CKO+ cells we detected fibrillar structures at the periphery of the cell, and the same diffuse signal seen in CKO- cells (*Figure 4B*, white arrows). We suspect that the diffuse signal observed in CKO+ cells is due to incomplete labeling of collagen-I in our preparation of Cy3-colI, as complete labeling of lysine residues would cause collagen-I to be unable to form fibrils (*Doyle, 2018*). Nonetheless, Dyngo4a treatment led to a lack of Cy3-colI fibrils (*Figure 4C*, right panel and quantification plot); thus, these results demonstrated that fibroblasts can effectively form fibrils from exogenous collagen alone.

## VPS33B controls collagen fibril formation but not protomeric secretion

VPS33B is situated in a post-Golgi compartment where it is involved in endosomal trafficking with multiple functions including extracellular vesicle formation (*Huang et al., 2021*), modulation of p53 signaling (*Liang et al., 2019*), and maintenance of cell polarity (*Wang et al., 2018*), dependent on cell type. We previously identified VPS33B to be a circadian-controlled component of collagen-I homeostasis in fibroblasts (*Chang et al., 2020*). As a result, we decided to further investigate its role in endosomal recycling of collagen-I.

We confirmed our previous finding that CRISPR knockout of VPS33B (VPSko, as verified by western blot analysis and qPCR analysis, *Figure 5—figure supplements 1 and 2*) in tendon fibroblasts led to fewer collagen fibrils without impacting proliferation (*Figure 5—figure supplement 3*), as evidenced by both electron microscopy (*Figure 5A*) and IF imaging (*Figure 5B*). Decellularized matrices showed

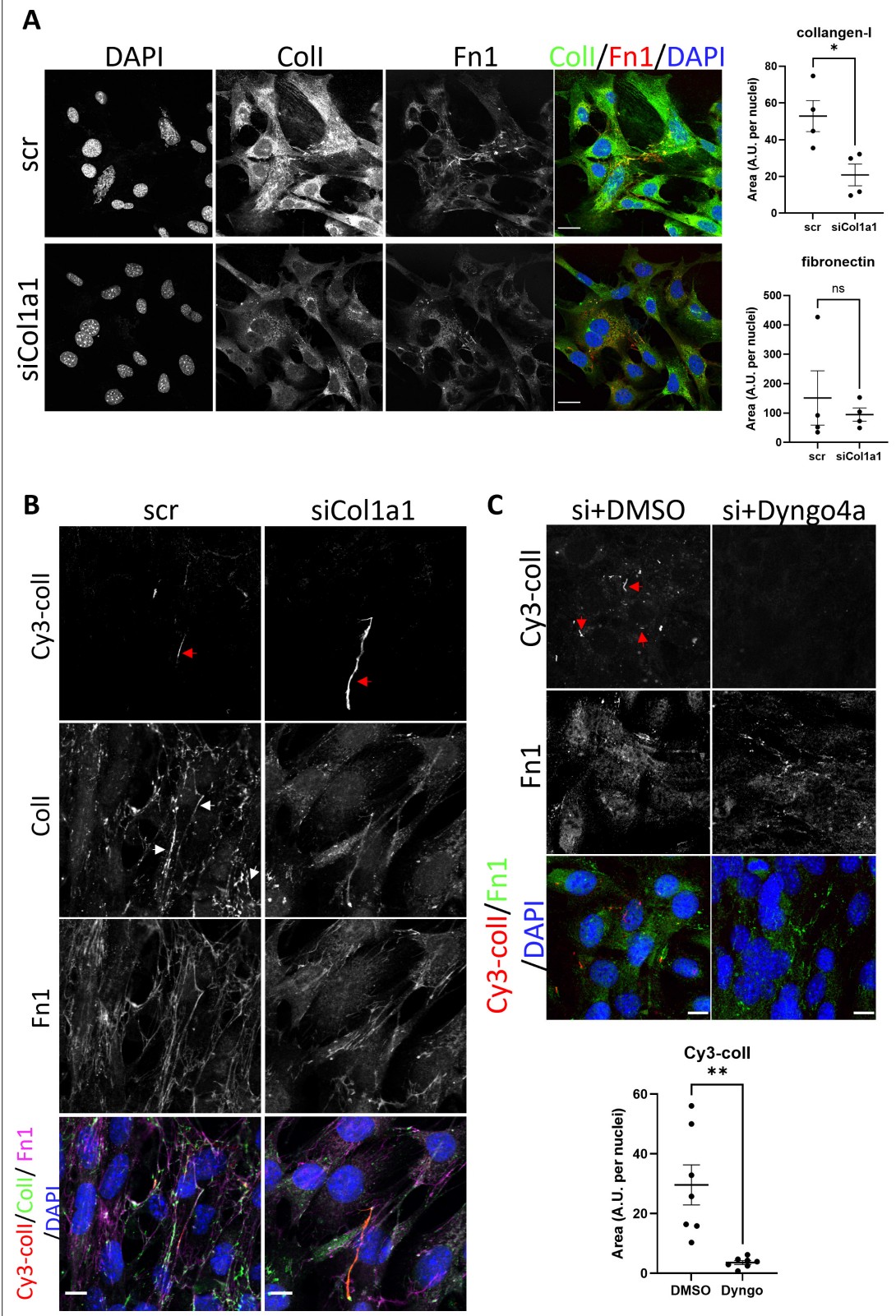

**Figure 3.** Collagen-I recycling can generate fibrils. (**A**) Fluorescent image series of iTTFs treated with scrambled control (top panel, scr), and siRNA against col1a1 (bottom panel, siCol1a1). Labels on top denote the fluorescence channel corresponding to proteins detected (ColI – collagen-I, Fn1 – fibronectin). Quantification of collagen-I and fibronectin signal to the right. Representative of N=4. Scale bar = 25 µm. *p=0.021. (**B**) Fluorescent image series of scr (left column) and siCol1a1 (right column) iTTFs incubated with Cy3-colI. Labels on left denote the fluorescence channel(s) corresponding

*Figure 3 continued on next page*

*Figure 3 continued*

to proteins detected (ColI – collagen-I, Fn1 – fibronectin). Cy3-colI fibrils highlighted by red arrows, and collagen-I fibrils highlighted by white arrows. Both scr cells and siCol1a1 cells can take up exogenous collagen-I and recycle to form collagen-I fibril. Representative of N>3. Scale bar = 10 μm. (**C**) Fluorescent image series of siCol1a1 iTTFs treated with DMSO control (left) or Dyngo4a (right) during Cy3-colI uptake, followed by further culture for 72 hr. Labels on left denote the fluorescence channel corresponding to proteins detected (ColI – collagen-I, Fn1 – fibronectin). Quantification of Cy3-colI signal to the bottom. Dyngo4a treatment led to a reduction of Cy3-colI fibrils. Representative of N>3. Scale bar = 20 μm. **p=0.0022.

The online version of this article includes the following figure supplement(s) for figure 3:

**Figure supplement 1.** Quantitative PCR (qPCR) analysis of scr and siCol1a1 iTTFs.

**Figure supplement 2.** Bar chart showing the percentage of iTTFs that have taken up 5 μg/mL Cy3-colI (left) or 5 μg/mL Cy5-colI (right) after 1 hr incubation.

that VPSko fibroblasts produced less matrix by mass than control, which was mirrored by a reduction in hydroxyproline content (*Figure 5C*). We then stably overexpressed VPS33B in fibroblasts (VPSoe), as confirmed by western blot, qPCR analyses, and flow cytometry of transfected cells (*Figure 5— figure supplements 4–6*). IF staining indicated a greater number of collagen-I fibrils in VPSoe cells (*Figure 5D*), although the mean total matrix and mean total hydroxyproline in VPSoe cultures was not significantly higher than control (*Figure 5E*). As VPSoe cells showed equivalent proliferation to controls (*Figure 5—figure supplement 7*), this suggests VPSoe specifically enhanced the assembly of collagen fibrils. We then performed time-series IF on synchronized cell cultures to quantify the number of collagen fibrils formed. Tendon fibroblasts exhibited an ~24 hr rhythmic fluctuation in collagen-I fibril numbers, whereas VPSoe cells continuously deposited collagen-I fibrils over the 55 hr period (*Figure 5F*). MetaCycle analyses indicated that fluctuation of fibril numbers in control and VPSoe cultures occurred at a periodicity of 22.7 hr and 28.0 hr respectively (*Figure 5—figure supplement 8*), indicating that continuous expression of VPS33B leads to loss of collagen-I fibril circadian homeostasis. Interestingly, when assessing the levels of secreted soluble collagen-I protomers from control and VPSoe cells, VPSoe CM have lower levels of soluble collagen-I (*Figure 5G*). In addition, when VPS33B is knocked down using siRNA, there is an elevation of soluble collagen-I secreted; siVPS33B on VPSoe cells also increased the levels of collagen-I in CM (*Figure 5—figure supplements 9 and 10*), while VPS33B levels are not correlated with *Col1a1* expression levels (*Figure 5—figure supplement 11*). This finding indicates that VPS33B is specifically involved in collagen-I fibril assembly, and not secretion or translation. The reduction of secreted soluble collagen-I in VPSoe cells further supports that VPS33B directs collagen-I toward fibril assembly.

## VPS33B-positive intracellular structures contain collagen-I

We then utilized the split-GFP system (*Magliery et al., 2005*; *Cabantous et al., 2005*) to investigate how VPS33B specifically directs collagen-I to fibril assembly. Here, a GFP signal will be present if VPS33B co-traffics with collagen-I. We, and others, have previously demonstrated that insertion of tags at the N-terminus of proα2(I) or proα1(I) chain does not interfere with collagen-I folding or secretion (*McCaughey et al., 2019*; *Calverley et al., 2020*). To determine which terminus of the VPS33B protein GFP1-10 should be added, we performed computational ΔG analysis (*Hessa et al., 2007*), which predicts that VPS33B contains two regions of extended hydrophobicity toward its C-terminus (*Figure 6—figure supplement 1A*). The first region (residues 565–587, denoted TMD1) has a ΔG of –0.62 kcal/mol, consistent with a single pass type IV transmembrane domain (TMD), known as a tail-anchor. This arrangement locates the short C-terminus of VPS33B inside the lumen of the endoplasmic reticulum (ER), and subsequently within the endosomal compartment (*Rabu et al., 2009*). The second region (residues 591–609, denoted HR1) is significantly less hydrophobic, with a ΔG of +2.61 kcal/mol (*Figure 6—figure supplement 1B*). Its presence raises the possibility that VPS33B may contain two TMDs that, depending on their relative membrane topologies, result in either a luminal or cytosolic C-terminus (*Figure 6—figure supplement 1C*).

To define the topology of VPS33B, we used a well-established in vitro system where newly synthesized and radiolabeled proteins of interest are inserted into the membrane of ER-derived canine pancreatic microsomes, and created constructs with ER luminal modification of either endogenous N-glycosylation sites (N54, N545 in VPS33B), or artificial sites in an appended OPG2 tag (N2, N15 in residues 1–18 of bovine rhodopsin, UniPot: P02699) as a robust reporter for membrane protein topology in the ER (*Figure 6—figure supplements 2 and 3*; *O'Keefe et al., 2022*). Due to the

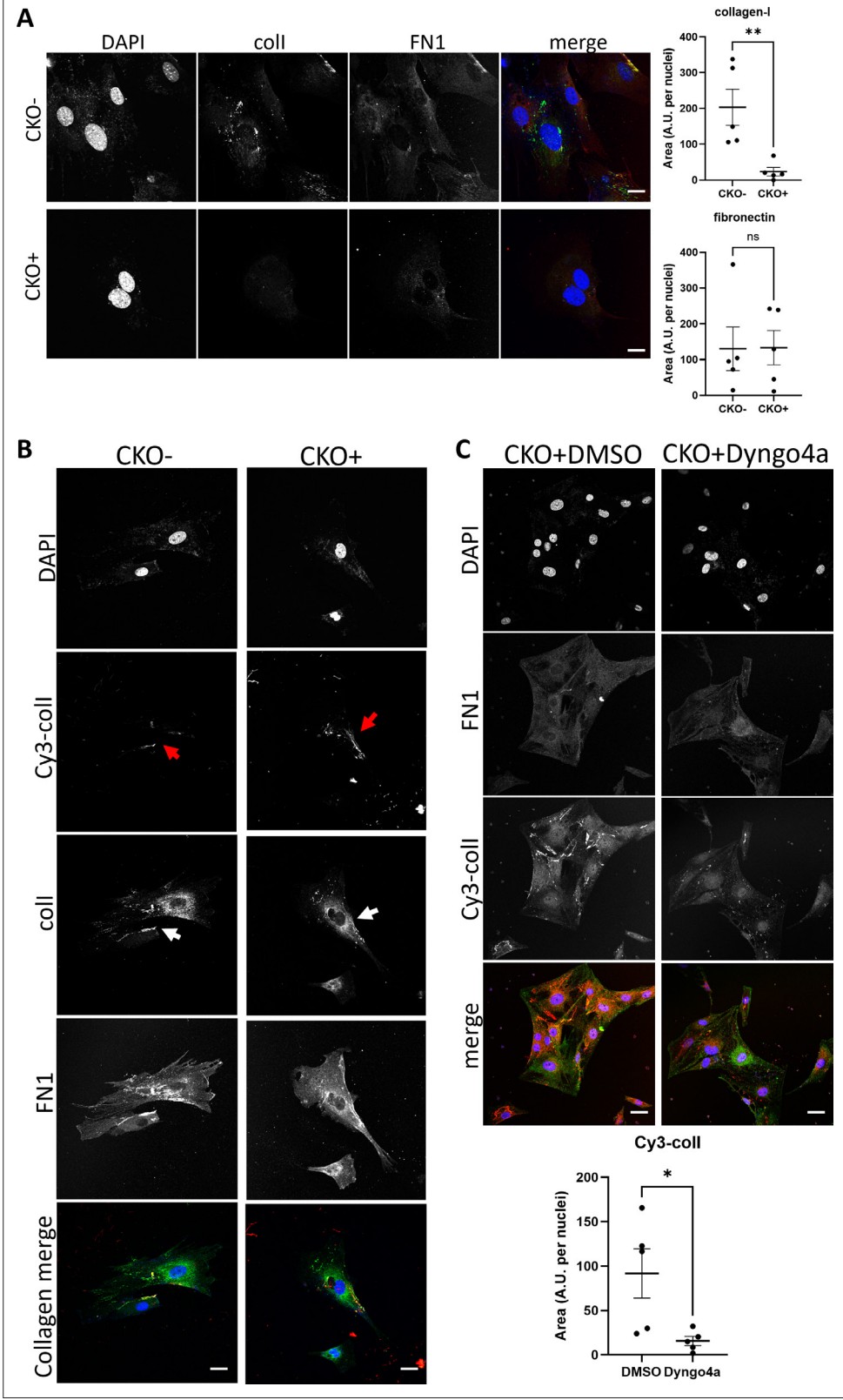

**Figure 4.** Fibroblasts without endogenous collagen-I can effectively make fibrils by endocytic recycling of exogenous collagen. (**A**) Fluorescent images of primary tail tendon fibroblasts isolated from control mice (top panel, CKO-), and tamoxifen-treated collagen-knockout mice (bottom panel, CKO+). Labels on top denote the fluorescence channel corresponding to proteins detected. Quantification of collagen-I and fibronectin

*Figure 4 continued on next page*

*Figure 4 continued*

fluorescence signal to the right. Representative of N=3. Scale bar = 10 μm. **p=0.0084. (**B**) Fluorescent images of CKO-/CKO+ tail tendon fibroblasts incubated with Cy3-colI. Labels on top denote the fluorescence channel corresponding to proteins detected. Cy3-colI fibril highlighted by red arrows, and collagen-I fibril highlighted by white arrows. Both CKO- and CKO+ cells can take up exogenous collagen-I and recycle to form collagen-I fibrils. Representative of N>3. Scale bar = 25 μm. (**C**) Fluorescent image series of CKO+ tail tendon fibroblasts treated with DMSO control (left) or Dyngo4a (right) during Cy3-colI uptake, followed by further culture for 72 hr. Labels on left denote the fluorescence channel corresponding to proteins detected. Quantification of Cy3-colI signal to the bottom. Dyngo4a treatment led to a significant reduction of Cy3-colI fibrils. Representative of N>3. Scale bar = 25 μm. *p=0.00273.

The online version of this article includes the following figure supplement(s) for figure 4:

**Figure supplement 1.** Quantitative PCR (qPCR) analysis of CKO- and CKO+ primary tail tendon fibroblasts.

**Figure supplement 2.** Bar chart showing the % of primary tail tendon fibroblasts that have taken up 5 μg/mL Cy3-colI (left) and Cy5-colI (right) after 1 hr incubation; CKO+ cells have a similar uptake to CKO- when incubated with Cy3-colI, and a slight but significant increase in uptake of Cy5-colI.

---

N-terminal region of VPS33B being highly aggregation-prone in our in vitro system (*Figure 6—figure supplements 4 and 5*), we created three additional chimeric proteins comprised of different regions of VPS33B and Sec61β to investigate the topology of VPS33B (*Figure 6—figure supplement 6*). While a small proportion of each chimera continued to pellet when synthesized in the absence of ER-derived microsomes (*Figure 6—figure supplement 7*, lanes 3, 6, 9), N-glycosylated species were now clearly identifiable for each of the chimeras tested.

In chimera 1, given the respective efficiency of membrane insertion of the tail-anchored region and N-glycosylation of the C-terminal OPG tag of Sec61βOPG2 (*Figure 6—figure supplement 7*, lanes 1–2), we attribute the N-glycosylated species to the C-terminal translocation of OPG2 tag, although it is evident that the remaining short stretch of VPS33B (residues 414–564) still impedes efficient membrane insertion, likely due to aggregation (*Figure 6—figure supplement 7*, lanes 3–5) (*O'Keefe et al., 2021b*; *Zong et al., 2019*). For chimera 2, the efficient modification of its distal N-glycosylation site (N15 of the OPG2 tag) but inefficient use of the proximal site (N2 of the OPG2 tag) reflects the latter residues' close proximity to the ER membrane (*Nilsson and von Heijne, 1993*), and supports the bona fide membrane insertion of the VPS33B TMD1 and thus translocation of the C-terminal OPG2 tag into the ER lumen (*Figure 6—figure supplement 7*, lanes 6–8). Interestingly, the inclusion of HR1 to the C-terminus of TMD1 in chimera 3 results in a substantial qualitative reduction in the amount of protein that is N-glycosylated (*Figure 6—figure supplement 7*, lanes 7 and 10). Further, for the fraction of chimera 3 that is modified, the majority is doubly N-glycosylated; most likely due to the extra length provided by HR1, resulting in both N-glycosylation sites on the OPG2 tag (N2 and N15) now being efficiently modified (*Nilsson and von Heijne, 1993*).

The clear reduction in the proportion of N-glycosylated species obtained with chimera 3 compared to chimera 2 (*Figure 6—figure supplement 7*) indicates that only a small proportion of HR1 and the appended OPG2 tag are successfully translocated into the ER lumen. This is likely due to a combination of the low proportion of hydrophobic residues and the presence of several charged and polar amino acids in HR1 (*Whitley et al., 2021*). We thus propose that, for a minority of VPS33B, TMD1 is inserted into the ER membrane as a tail-anchored region with a luminal HR1 that most likely remains associated with the inner leaflet of the bilayer (N-cytosolic, C-luminal; *Figure 6A*, right). In contrast, the majority of VPS33B likely assumes a 'hairpin' like conformation in the ER membrane where both its N- and C-termini remain in the cytosol (N-cytosolic, C-cytosolic) with either: (1) a partially membrane inserted TMD1, with HR1 associated with the outer leaflet of the ER membrane (*Figure 6A*, left); or (2) a fully membrane inserted TMD1 followed by the marginally hydrophobic HR1 which may span the membrane through the formation of stabilizing hydrogen bonds with TMD1 (*Figure 6A*, middle; *Meindl-Beinker et al., 2006*).

To test our topology findings, we inserted BFP at either the N- (VPSnBFP) or C-terminus (VPScBFP) of VPS33B. In cells expressing VPSnBFP, the fluorescence signal was diffused throughout the cell body, and in some cases appeared to be completely excluded from circular structures (*Figure 6B*, left), suggestive of protein mistargeting. In contrast, VPScBFP-expressing cells have punctate blue fluorescence signals and peripherally blue structures (*Figure 6B*, right), supportive of the topology

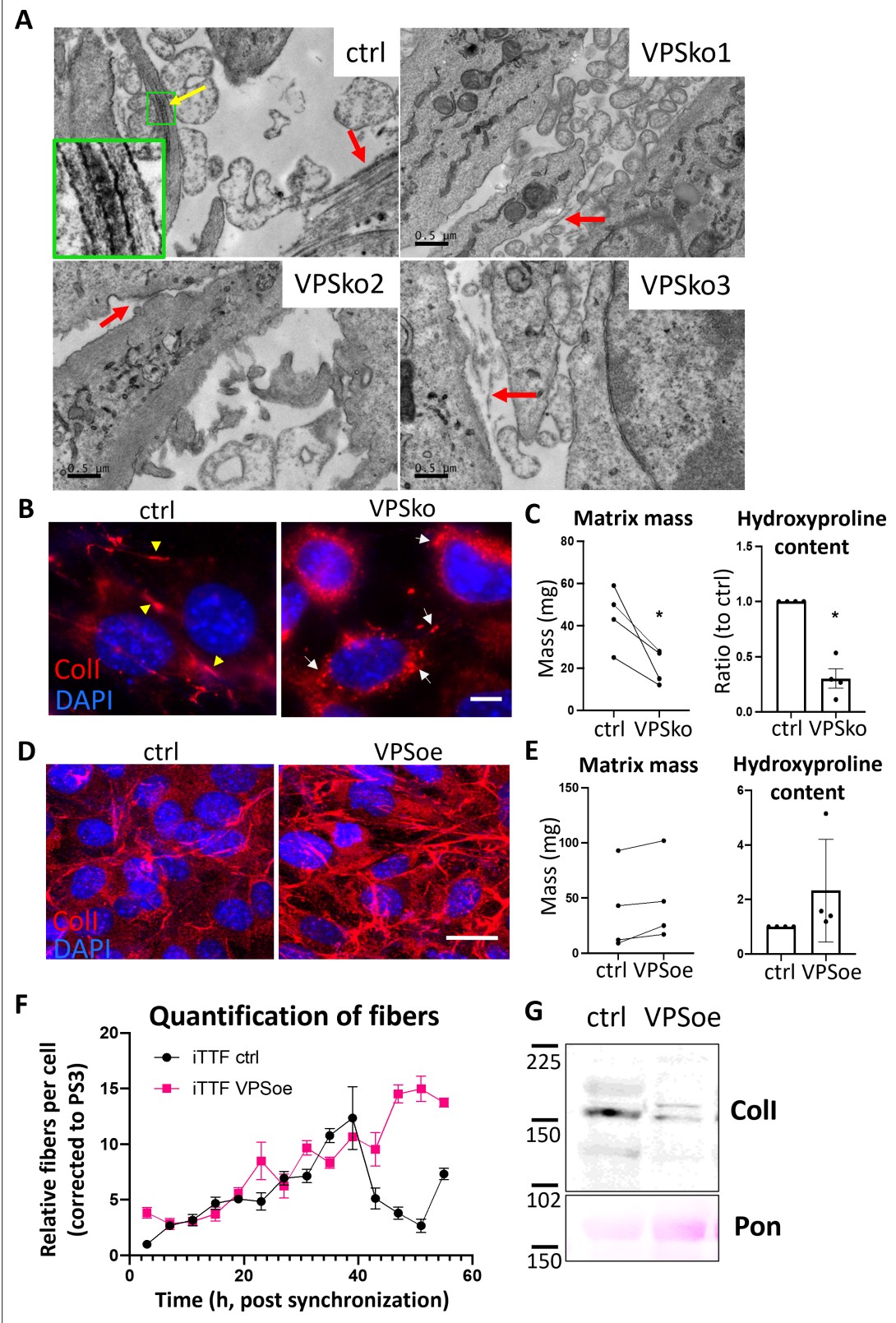

**Figure 5.** VPS33B controls collagen fibril formation at the plasma membrane in a rhythmic manner. (**A**) Electron microscopy images of fibroblasts plated on ACLAR and grown for a week before fixation and imaging. Ctrl culture has numerous collagen-I fibrils, as pointed out by arrows. Yellow arrow points to a fibripositor, and green box is expanded to the left bottom corner, showing the distinct D-banding pattern of collagen-I fibril when observed with electron microscopy. VPSko clones all have fewer and thinner fibrils present in the culture (pointed out by red arrows). Representative of

*Figure 5 continued on next page*

*Figure 5 continued*

N=3. Scale bar = 0.5 µm. (**B**) Fluorescence images of collagen-I (red) and DAPI counterstain in ctrl and VPSko iTTFs. Yellow arrows indicating collagen fibrils, and white arrows pointing to collagen-I presence in intracellular vesicles. Representative of N>6. Scale bar = 25 µm. (**C**) Matrix deposition by ctrl or VPSko iTTFs, after 1 week of culture. Left: decellularized matrix mass. N=4, *p=0.0299. Right: hydroxyproline content presented as a ratio between ctrl and VPSko cells. N=4, *p=0.0254. Ratio-paired t-test used. (**D**) Fluorescence images of collagen-I (red) and DAPI counterstain in ctrl and VPSoe iTTFs. Representative of N>6. Scale bar = 20 µm. (**E**) Matrix deposition by ctrl or VPSoe iTTFs, after 1 week of culture. Left: decellularized matrix mass, N=4. Right: hydroxyproline content presented as a ratio between ctrl and VPSoe cells, N=4. Ratio-paired t-test used. (**F**) Relative collagen fibril count in synchronized ctrl (black) and VPSoe (pink) iTTFs, corrected to the number of fibrils in ctrl cultures at start of time course. Fibrils scored by two independent investigators. Bars show mean ± s.e.m. of N=2 with n=6 at each time point. (**G**) Western blot analysis of conditioned media taken from ctrl and VPSoe iTTFs after 72 hr in culture. Top: probed with collage-I antibody (ColI), bottom: counterstained with Ponceau (Pon) as control. Protein molecular weight ladders to the left (in kDa). Representative of N=3.

The online version of this article includes the following figure supplement(s) for figure 5:

**Figure supplement 1.** Western blot analysis of VPS33B knockout (VPSko) clones compared to control (ctrl) iTTFs.

**Figure supplement 2.** Quantitative PCR (qPCR) analysis of VPS33B expression in the three selected clones.

**Figure supplement 3.** Alamar blue analysis of proliferation rates of ctrl and VPSko iTTFs.

**Figure supplement 4.** Western blot analysis of VPS33B protein levels in control (ctrl) and VPS33B overexpressing (VPSoe) iTTFs.

**Figure supplement 5.** Quantitative PCR (qPCR) analysis of VPS33B expression in ctrl and VPSoe iTTFs.

**Figure supplement 6.** Single parameter histograms of flow cytometry analysis on ctrl (left) and VPSoe (right) iTTFs, showing a shift in increase of RFP fluorescence and thus expression of VPSoe vector.

**Figure supplement 7.** Alamar blue analysis of proliferation rates of ctrl iTTFs and iTTF VPSoe.

**Figure supplement 8.** MetaCycle analyses of the fibril counts showed a rhythmicity of circa 23 hr in ctrl iTTFs compared with circa 28 hr in iTTF VPSoe.

**Figure supplement 9.** Quantitative PCR (qPCR) analysis of VPS33b mRNA expression levels in iTTF or iTTF VPSoe, treated with siRNA scrambled control (iT scr, iToe scr) or siRNA against VPS33b (iT siVPS, iToe siVPS) and cultured for 72 hr.

**Figure supplement 10.** Western blot analysis of conditioned media taken from ctrl and VPSoe iTTFs, treated with either siRNA scrambled control (scr) or siRNA against VPS33B (siVPS) and cultured for 72 hr.

**Figure supplement 11.** Quantitative PCR (qPCR) analysis of Col1a1 mRNA expression levels in iTTF or iTTF VPSoe, treated with siRNA scrambled control (iT scr, iToe scr) or siRNA against VPS33b (iT siVPS, iToe siVPS) and cultured for 72 hr.

findings. Previous studies have also tagged VPS33B at the C-terminus (*Hunter et al., 2018*). Thus, the 214-residue N-terminal fragment (GFP1-10) was cloned onto VPS33B, and the 17-residue C-terminal peptide (GFP11) was cloned onto the proα1(I) chain of collagen-I as previously described (*McCaughey et al., 2019*) (GFP11-proα1(I), *Figure 6C*).

Fibroblasts stably expressing VPS-barrel and GFP11-proα1(I) were imaged to identify any GFP fluorescence. The results showed puncta throughout the cell body (*Figure 6D*). Intriguingly, GFP fluorescence was also observed at the cell periphery (*Figure 6D*, green box zoom). IF staining of endogenous VIPAS39 (a known VPS33B-interacting partner; *Hunter et al., 2018*) also revealed co-localization of VIPAS39 with collagen-I in intracellular punctate structures, where in some of the co-localized puncta, the signal of VIPAS39 is strongest surrounding collagen-I (*Figure 6E*, zoom), suggesting that VIPAS39 is encasing collagen-I, not within the lumen but present in proximity with the external membrane of these structures. VPS33B has been demonstrated to interact with VIPAS in regions before the TMD1 site (*Tornieri et al., 2013*); thus, in all suggested VPS33B topologies herein, VIPAS39 will still be able to interact with VPS33B, and encase collagen-I-containing intracellular structures.

The relationship between VPS33B levels and collagen fibril numbers was confirmed using endogenously tagged Dendra2-collagen-I expressing 3T3 cells (*Pickard et al., 2018*), where notably the number of Dendra2-positive fibrils significantly increased when VPS33BcBFP was expressed; however, the average length of the fibril was not significantly different, suggesting that VPS33B is important in fibril initiation but not elongation (*Figure 6F*). Incubation of iTTFs expressing only the VPS-barrel with CM collected from iTTFs expressing only GFP11-proα1(I) revealed that the GFP signal was detected only in intracellular structures and not along the periphery of the cells, where endocytosis takes place (*Figure 6G*). Flow cytometry analyses of fluorescently labeled-colI endocytosed by control, VPSko, and VPSoe fibroblasts also revealed no consistent change in uptake by VPSko or VPSoe cells (*Figure 6H*). Taken together, these results suggest that VPS33B interacts with endocytosed collagen-I within the cell and trafficks with collagen-I to the extracellular space, and is not involved with collagen-I endocytosis. VPSko cells replated after Cy5-colI uptake have conspicuously fewer fibrils when compared to

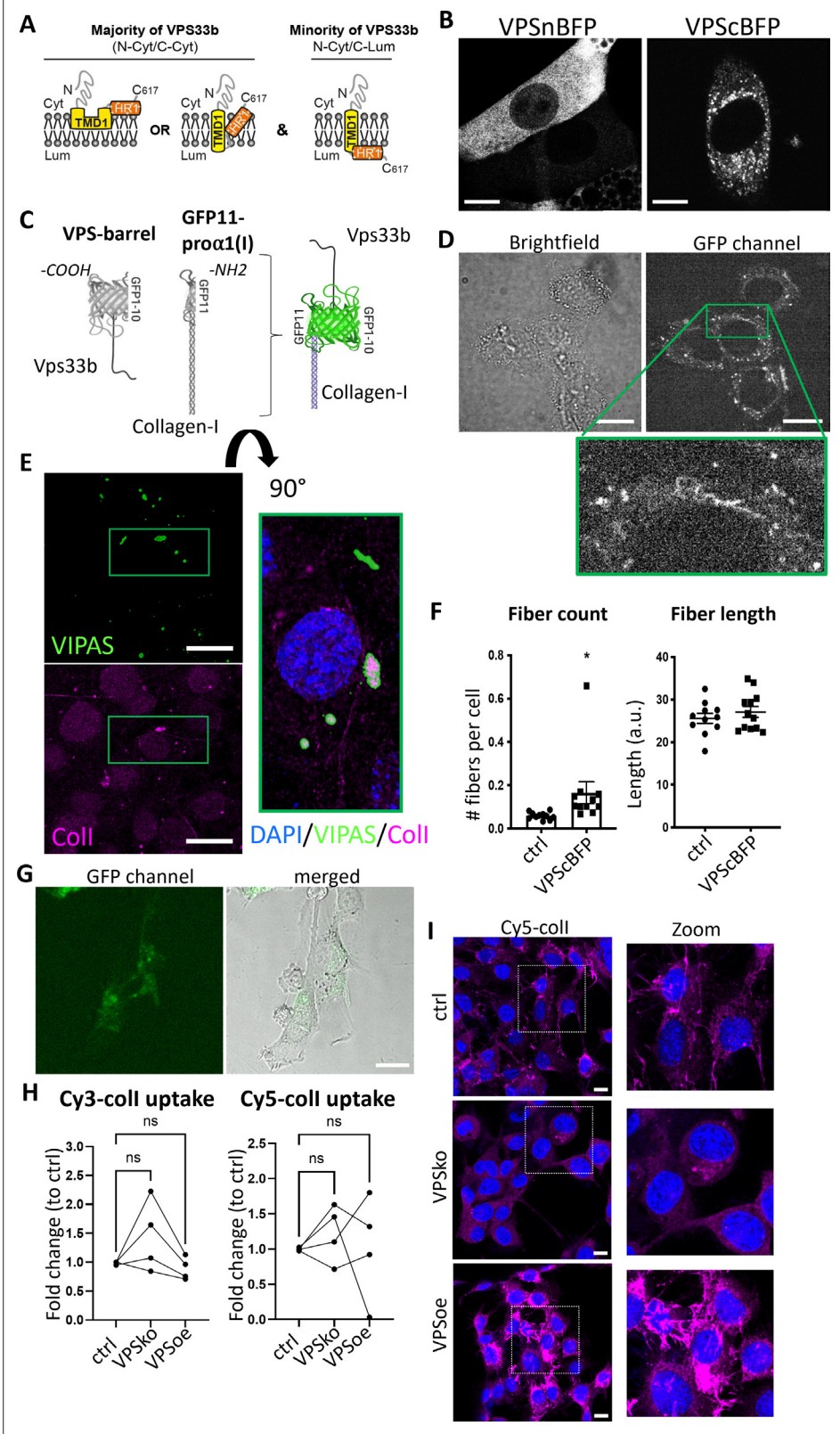

**Figure 6.** Procollagen-I and VPS33B localize to the same compartments. (**A**) Schematic depicting the proposed membrane topologies of VPS33b. (**B**) iTTFs expressing BFP-tagged VPS33B. Left: BFP tagged on the N-terminal end of VPS33B (VPSnBFP). Right: BFP tagged on the C-terminal end (VPScBFP). Images taken in Airy mode. Representative of N>4. Scale bar = 10 μm. (**C**) Schematic of the split-GFP system. GFP1-10 barrel is introduced

*Figure 6 continued on next page*

*Figure 6 continued*

into VPS33B (VPS-barrel), and GFP11 to alpha-1 chain of collagen-I (GFP11-proα1(I)). If the two tagged proteins co-localize (e.g. in a vesicle), a GFP signal will be emitted. (**D**) Brightfield (top) and fluorescence (middle) images of iTTFs expressing both VPS-barrel and GFP11-proα1(I) constructs. Representative of N=5. Green box is expanded to the bottom, to highlight the punctate fluorescence signals within intracellular vesicular structures, as well as fibril-like structures suggestive of fibril assembly sites. Scale bar = 20 µm. (**E**) Fluorescence images of VIPAS (green), collagen-I (red), and DAPI counterstain in iTTFs. Representative of N=3. Green box is expanded to the right (flipped 90°) to show strong VIPAS signal encasing collagen-I. Scale bar = 25 µm. (**F**) Quantification of average number of fibrils per cell (left) and average fibril length (right) in control endogenously tagged Dendra-coll expressing 3T3 cells (ctrl) and Dendra-coll expressing 3T3 overexpressing VPScBFP (VPScBFP). >500 cells quantified per condition. N=12. *p=0.048. (**G**) Brightfield (left) and fluorescence (middle) images of iTTFs expressing VPS-barrel incubated with conditioned media containing Col1a1-GFP11 for 24 hr. Scale bar = 25 µm. (**H**) Line charts comparing the percentage of iTTFs that have taken up 5 µg/mL Cy3-coll (left) and Cy5-coll (right) after 1 hr incubation between control (ctrl), VPS33B-knockout (VPSko), and VPS33B-overexpressing (VPSoe) cells, corrected to control. RM one-way ANOVA was performed. N=4. (**I**) Fluorescence images of iTTFs of different levels of VPS33B expression, fed with Cy5-coll and further cultured for 72 hr. Cultures were counterstained with DAPI. Box expanded to right of images to show zoomed-in images of the fibrils produced by the fibroblasts. Representative of N=2.

The online version of this article includes the following figure supplement(s) for figure 6:

**Figure supplement 1.** Computational prediction on membrane topology of VPS33B.

**Figure supplement 2.** Outline of the in vitro assay using canine pancreatic as a source of endoplasmic reticulum (ER) membrane; following translation, membrane inserted radiolabeled precursor proteins are recovered by centrifugation and analyzed by SDS-PAGE and phosphorimaging.

**Figure supplement 3.** Schematics of endogenous, truncated, and OPG2-tagged VPS33b proteins used in this study.

**Figure supplement 4.** Non-glycosylated and N-glycosylated radiolabeled cell-free translation products are respectively indicated by a yellow or magenta circle.

**Figure supplement 5.** Non-glycosylated and N-glycosylated radiolabeled products are respectively indicated by a yellow or magenta circle.

**Figure supplement 6.** Schematics of OPG2-tagged VPS33b and Sec61β chimeric proteins used in this study; chimera 1: residues 414–564 VPS33b, residues 73–94 Sec61β (TA region), OPG2 tag; chimera 2: residues 1–72 Sec61β (N-terminal region), residues 565–587 VPS33b (TMD1), OPG2 tag; chimera 3: residues 1–72 Sec61β (N-terminal region), residues 565–617 VPS33b (TMD1, HR1 and C-terminus), OPG2 tag.

**Figure supplement 7.** Non-glycosylated and N-glycosylated radiolabeled products are respectively indicated by a yellow or magenta circle.

control. In contrast, VPSoe cells have shorter but more Cy5-coll fibrils (*Figure 6I*), highlighting the role of VPS33B in recycling endocytosed collagen-I to initiate collagen fibrillogenesis.

## Integrin chain α11 mediates VPS33B-dependent fibrillogenesis

Having identified VPS33B as a driver for collagen-I fibril formation but not protomeric secretion, we used biotin cell surface labeling coupled with mass spectrometry protein identification to identify other proteins that may be involved in this process at the cell surface. VPSko and VPSoe fibroblasts were analyzed using a 'shotgun' approach. In total, 4121 proteins were identified in total lysates (*Supplementary file 1*), and 1691 proteins in the enriched-for-surface-protein samples (*Supplementary file 2*). Gene ontology (GO) Functional Annotation analysis (*Huang et al., 2009a*; *Huang et al., 2009b*) identified the top 25 enriched terms (based on p-values) with the top 5 terms all associated with 'extracellular' or 'cell surface' (*Figure 7A*), indicative that the biotin-labeling procedure had successfully enriched proteins at the cell surface interacting or associated with the extracellular matrix (ECM).

We then interrogated the differences between VPSko and control cells, visualizing the results in a semi-quantitative manner using spectral counting (*Supplementary file 3*). Conspicuously, collagen α1(I) and α2(I) chains were detected at a reduced level at the surface of VPSko cells (*Figure 7B*). The reduction of α1(V) and α2(V) chains (which constitute type V collagen) from the cell surface supports the long-standing view that collagen-V nucleates collagen-I containing fibrils. GO pointed to integral

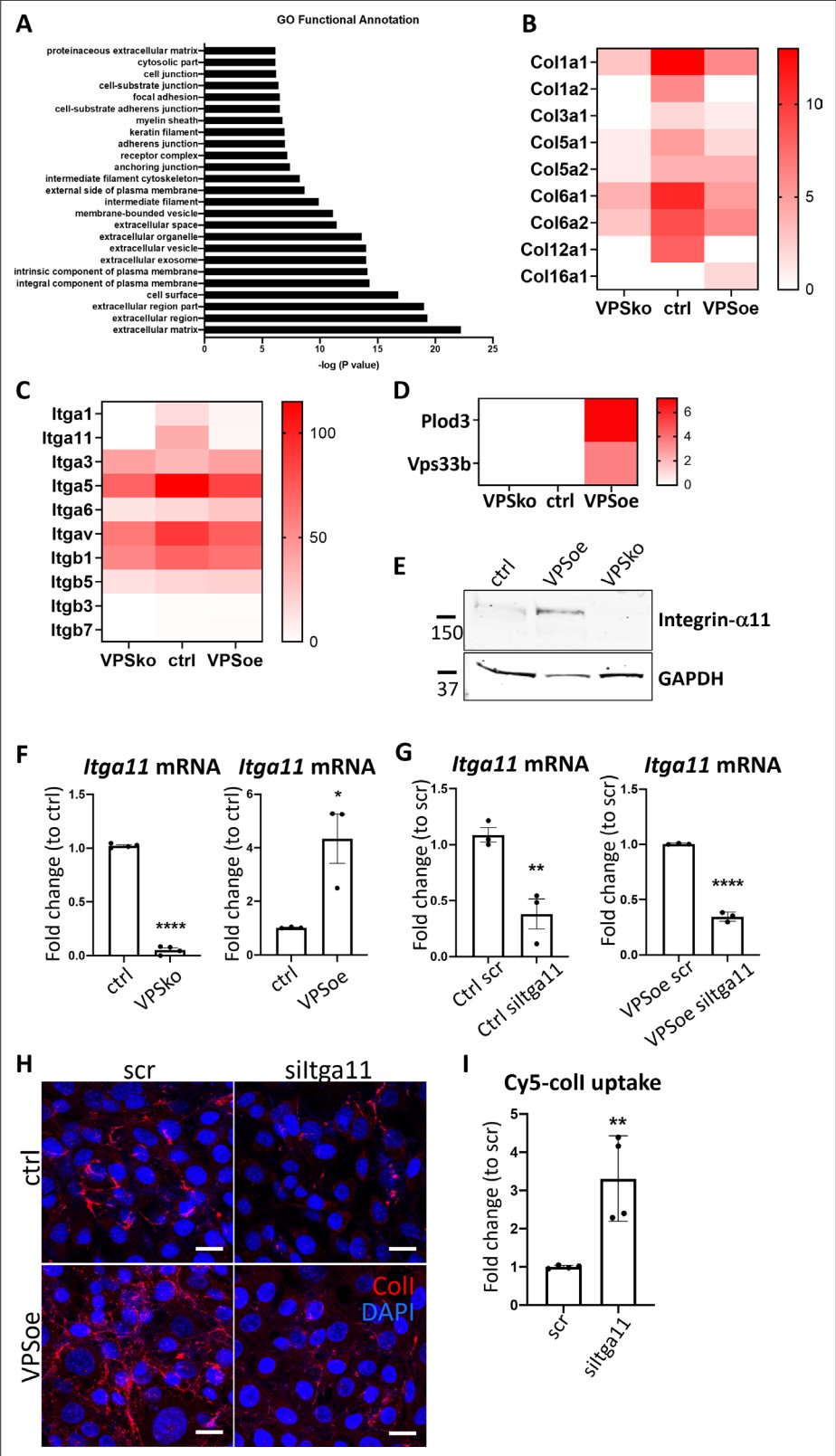

**Figure 7.** Integrin α11 subunit mediates VPS33B-effects and is required for collagen-I fibrillogenesis. (**A**) Top 25 Functional Annotation of proteins detected in biotin-enriched samples when compared to non-enriched samples based on p-values. Y-axis denotes the gene ontology (GO) term, X-axis denotes –log (p-value). (**B**) Heatmap representation of spectral counting of collagens detected in biotin-enriched surface proteins from

*Figure 7 continued on next page*

*Figure 7 continued*

control (ctrl), VPS33B-knockout (VPSko), and VPS33B-overexpressing (VPSoe) iTTFs. Scale denotes quantitative value as normalized to total spectra, as determined by Proteome Discoverer. (**C**) Heatmap representation of spectral counting of integrins detected in biotin-enriched surface proteins from control (ctrl), VPS33B-knockout (VPSko), and VPS33B-overexpressing (VPSoe) iTTFs. Scale denotes quantitative value as normalized to total spectra, as determined by Proteome Discoverer. (**D**) Heatmap representation of spectral counting of Plod3 and VPS33B detected in biotin-enriched surface proteins from control (ctrl), VPS33B-knockout (VPSko), and VPS33B-overexpressing (VPSoe) iTTFs. Scale denotes quantitative value as normalized to total spectra, as determined by Proteome Discoverer. (**E**) Western blot analysis of integrin α11 subunit levels in control (ctrl), VPS33B-overexpressing (VPSoe), VPS33B-knockout (VPSko) iTTFs. Top: probed with integrin α11 antibody, bottom: reprobed with GAPDH antibody. Protein molecular weight ladders to the left (in kDa). Representative of N=3. (**E**) Quantitative PCR (qPCR) analysis of *Itga11* transcript levels in ctrl compared to VPSko iTTFs (left), and ctrl compared to VPSoe iTTFs (right). N>3, ****p<0.0001, *p=0.0226. (**F**) qPCR analysis of *Itga11* mRNA expression in ctrl (left) or VPSoe (right) iTTFs treated with either scrambled control (scr) or siRNA against Itga11 (siItga11), collected after 96 hr. N=3, **p=0.0091, ****p<0.0001. (**G**) Immunofluorescence (IF) images of ctrl and VPSoe iTTFs treated with either control siRNA (scr) or siRNA again Itga11 (siItga11), after 72 hr incubation; collagen-I (red) and DAPI (blue) counterstained. Representative of N=3. Scale bar = 25 µm. (**H**) Bar chart comparing the percentage of iTTFs that have taken up 5 µg/mL Cy5-coll after 1 hr incubation between fibroblasts treated with scrambled control (ctrl) or siRNA against Itga11 (siItga11), corrected to scr. N=3. **p=0.0062.

components of the plasma membrane, which included several integrins. While many integrins were detected in all samples, there was an absence of integrin α1 and integrin α11 subunit in VPSko cultures (*Figure 7C*). It is well established that integrins are cell surface molecules that interact extensively with the ECM (*Humphries et al., 2006*), with integrin α11β1 functioning as a major collagen binding integrin on fibroblasts (*Zeltz and Gullberg, 2016*), which is expressed during development, and is upregulated in subsets of myofibroblasts in tissue and tumor fibrosis (*Zeltz et al., 2023*; *Zeltz and Gullberg, 2023*). Importantly, while the level of integrin β1 chain detected was also reduced in VPSko cultures, it was not as drastic as the reduction of integrin α11 chain. This is likely due to the promiscuous nature of integrin β1 subunit, that is being able to partner with other α subunits for functions other than collagen-I interaction. The absence of integrin α11 subunit from VPSko cells suggested a link between VPS33B-mediated collagen fibrillogenesis and integrin α11β1.

VPS33B was conspicuous by its absence from cell surface labeling studies of control samples (*Figure 7D*). We have previously struggled to detect VPS33B protein using mass spectrometry (*Chang et al., 2020*) and postulate that its absence could be due to low abundance. Regardless, VPS33B was detected at the cell surface in VPSoe samples along with PLOD3 (*Figure 7D*), a lysyl hydroxylase involved in stabilizing collagen and previously identified to be delivered by VPS33B (*Banushi et al., 2016*). Complementary western blot analysis of total cell lysates showed that expression of integrin α11 subunit is significantly reduced in VPSko cells and is elevated in VPSoe cells (*Figure 7E*). This correlation between VPS33B and integrin α11 expression levels was confirmed at the mRNA level (*Figure 7F*), inferring a link between VPS33B, collagen fibril, and integrin α11 abundance. We verified this by siRNA knockdown of *Itga11* in control and VPSoe fibroblasts, where knockdown efficiency is confirmed by qPCR (*Figure 7G*). siItga11 in both control and VPSoe cells reduced the number of collagen-I fibrils in culture (*Figure 7H*). Thus, even with elevated levels of VPS33B, integrin α11 subunit is required for collagen-I fibrillogenesis. Interestingly, knocking down integrin α11 subunit increased exogenous collagen-I uptake as demonstrated by flow cytometry (*Figure 7I*). Thus, we propose that both VPS33B and integrin α11 are involved in directing collagen-I protomers to the formation of collagen-I fibrils, and not mediators of collagen-I endocytosis.

## Integrin α11 subunit is localized to the fibroblastic focus of idiopathic pulmonary fibrosis

We next investigated if VPS33B and integrin α11 are involved in human fibrotic diseases, where accumulation of collagen fibrils is a disease hallmark. Lung fibroblasts isolated from control individuals or individuals suffering from IPF were cultured and the mRNA levels of *COL1A1*, *ITGA11*, and *VPS33B* determined (*Figure 8A*). Although *COL1A1* was similarly expressed, the levels of *VPS33B* and *ITGA11* were significantly increased in IPF fibroblasts compared to control. We next assessed the endocytic capacities of collagen-I in human lung fibroblasts, and confirmed co-localization of

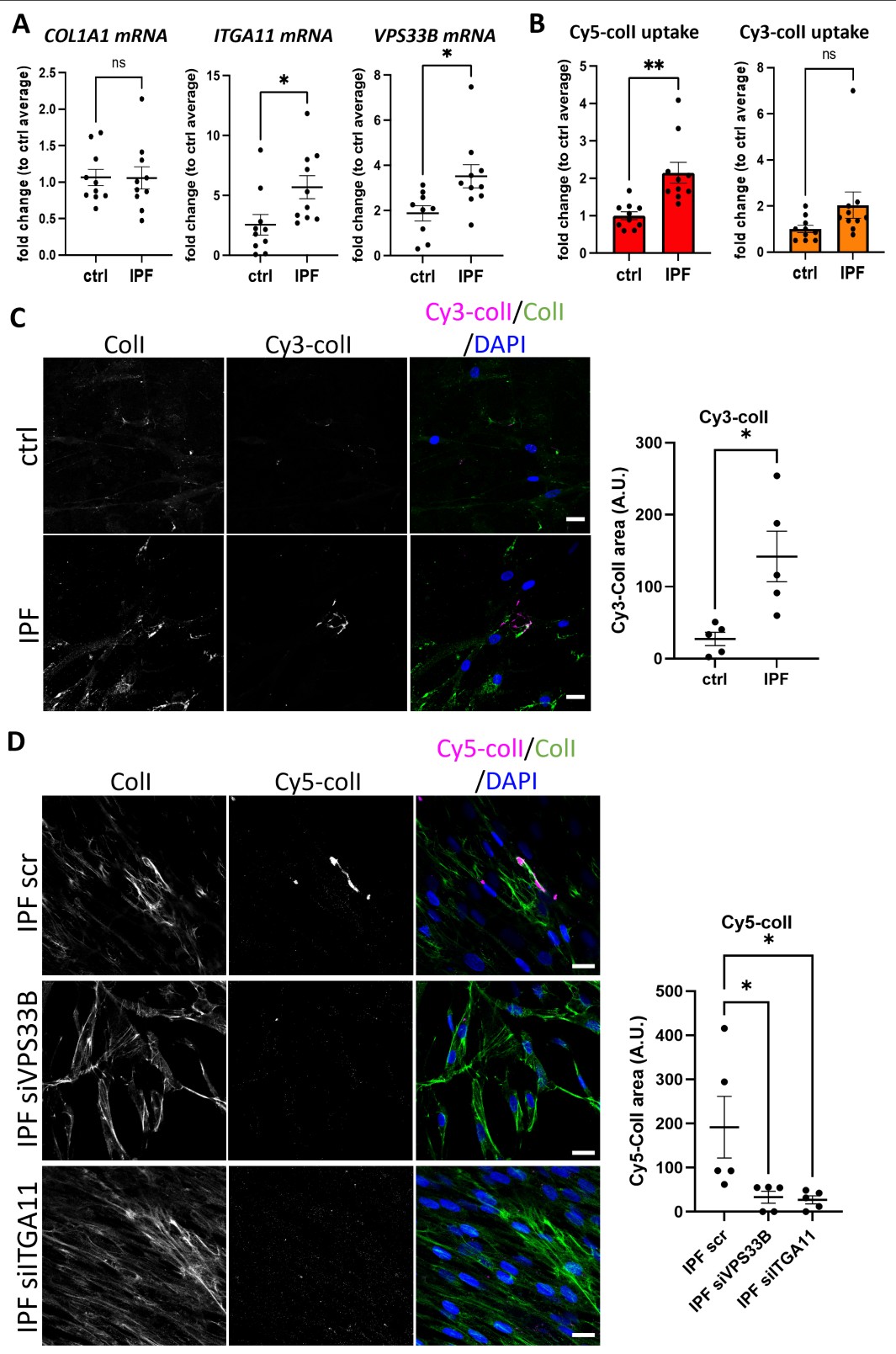

**Figure 8.** Fibroblasts derived from idiopathic pulmonary fibrosis (IPF) patients have higher collagen endocytic recycling capacity that is mediated by VPS33B and ITGA11. (**A**) Quantitative PCR (qPCR) analysis of patient-derived fibroblasts isolated from control (ctrl) or IPF lungs. Bars showing mean ± s.e.m., 5 patients in each group from 2 independent experiments (technical repeats not shown here). *Itga11, *p=0.0259; VPS33B, *p=0.0183. (**B**)

*Figure 8 continued on next page*

*Figure 8 continued*

Fold change of percentage Cy5-colI (left) or Cy3-colI (right) taken up by ctrl or IPF lung fibroblasts, corrected to average of control fibroblasts. Bars showing mean ± s.e.m., 5 patients in each group from 2 independent experiments (technical repeats not shown here). **p=0.003.(**B**) Fluorescent images of ctrl or IPF lung fibroblasts that have taken up Cy3-colI (magenta), followed by further culture for 48 hr in the presence of ascorbic acid, before subjected to collagen-I staining (green). Labels on top denote the fluorescence channel corresponding to proteins detected. Quantification of Cy3-colI signal to the right. IPF fibroblasts produced more Cy3-labeled fibrillar structures. Representative of N=5. Scale bar = 20 µm. *p=0.0135. (**C**) Fluorescent images of IPF lung fibroblasts treated with siRNA scrambled control (scr), siRNA against VPS33B (siVPS33B), or siRNA against ITGA11 (siITGA11) prior to uptake of Cy5-colI (magenta). This was followed by further culture for 48 hr in the presence of ascorbic acid, before subjected to collagen-I staining (green). Labels on top denote the fluorescence channel corresponding to proteins detected. Quantification of Cy5-colI signal to the right. Both siVPS33B and siITGA11 significantly reduced recycled collagen signals. Representative of n=5 across N=2. Ordinary one-way ANOVA with multiple comparisons (to scr) was performed on quantification of Cy5-colI signal. siVPS33B, *p=0.0341; siITGA11, *p=0.0282.

The online version of this article includes the following figure supplement(s) for figure 8:

**Figure supplement 1.** Human lung fibroblasts transiently transfected with GFP-tagged RAB5 (RAB5-GRP, green) and with Cy5-colI added before confocal live imaging.

**Figure supplement 2.** Quantitative PCR (qPCR) analyses of ITGA11 and VPS33B mRNA levels in idiopathic pulmonary fibrosis (IPF) cells treated with siRNA against ITGA11 (siITGA11) or siRNA against VPS33B (siVPS33B).

exogenous Cy5-colI with Rab5-GFP (*Figure 8—figure supplement 1*). We also found that IPF fibroblasts endocytosed significantly more Cy5-labeled exogenous collagen-I when compared to control fibroblasts (*Figure 8B*, left); uptake of Cy3-labeled exogenous collagen-I was also elevated, albeit not significantly (*Figure 8B*, right). Subsequent culture of cells that have taken up exogenous collagen-I revealed that IPF fibroblasts made significantly more fluorescently labeled collagen fibrils, indicating enhanced recycling of endocytosed collagen-I to generate new fibrils (*Figure 8C*). The relationship between VPS33B, ITGA11, and endocytic recycling of collagen-I for fibrillogenesis was further confirmed using siRNA (*Figure 8—figure supplement 2*), where knockdown of either proteins led to a significant decrease in recycling of exogenous Cy5-ColI (*Figure 8D*). This in vitro observation was also represented in IPF pathology, where patient-derived lung samples showed enrichment of collagen-I which overlaps with both integrin α11 subunit and VPS33B within the IPF hallmark lesion (termed the fibroblastic focus [*Herrera et al., 2019*], encircled by red dotted line, *Figure 9A*). In sites of emerging fibrotic remodeling, integrin α11 subunit and VPS33B are also detected (red asterisks, *Figure 9B*; additional N=3 IPF specimens, *Figure 9—figure supplement 1*), indicating a role for VPS33B/integrin α11 chain in collagen-I deposition in the context of IPF. In control lung samples, collagen-I and VPS33B were present whereas integrin α11 subunit was detected at negligible levels (*Figure 9C*, *Figure 9—figure supplement 2*). These results indicate that the proteins required for assembly of collagen into fibrils (i.e. integrin α11, VPS33B) are present at the fibrotic fronts of IPF, and that enhanced endocytic recycling of collagen-I by fibroblasts may be a disease-potentiating mechanism in IPF.

## VPS33B and integrin α11 subunit levels are elevated in chronic skin wounds

Fibrosis has been described as a dysregulation of the normal wound-healing process (*Wynn, 2011*). Thus, we postulated that chronic skin wounds, a similar chronic inflammation unresolved wound-healing condition, may also share a similar pathological molecular signature as IPF. Immunohistochemical (IHC) staining revealed that in human chronic skin wounds, the expressions of integrin α11 subunit and VPS33B are elevated when compared to normal skin areas taken from the same patient (*Figure 10*), where expression was evident at both the fibrotic wound margins and perivascular tissues, indicative of areas under constant collagen remodeling, suggestive of a similar dysregulation in collagen fibrillogenesis pathway utilizing VPS33B and integrin α11.

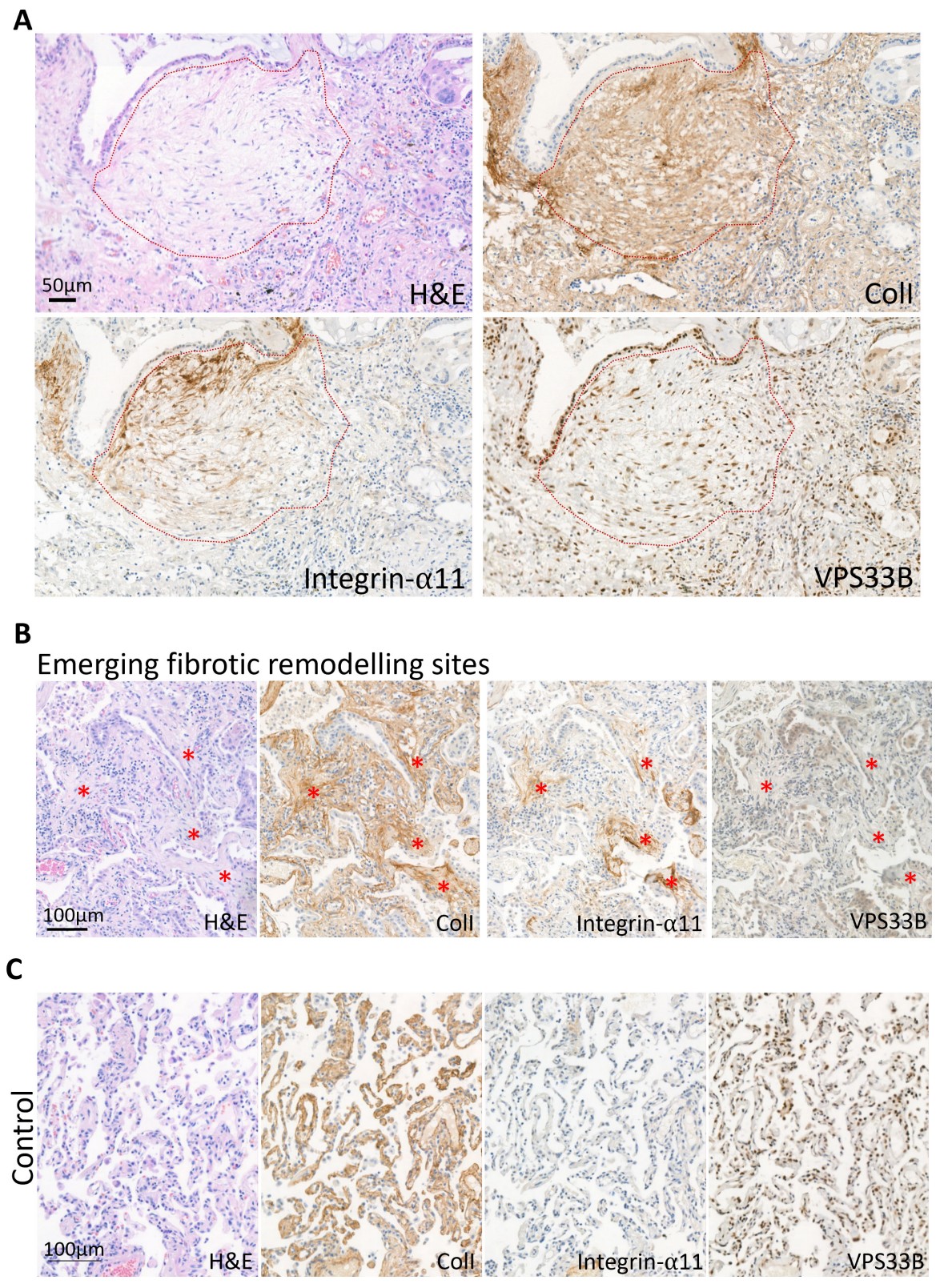

**Figure 9.** The idiopathic pulmonary fibrosis (IPF) fibrotic focus is positive for integrin α11 subunit and VPS33B. (**A**) Immunohistochemistry of IPF patient (patient 1) with red dotted line outlining the fibroblastic focus, the hallmark lesion of IPF. Sections were stained with hematoxylin and eosin (H&E), collagen-I (ColI), integrin α11, VPS33B. Scale bar = 50 μm. (**B**) Immunohistochemistry of IPF patient 4 showing regions of emerging fibrotic remodeling with evidence of fibroblastic foci formation (red asterisks). Sections were stained with H&E, ColI, integrin α11, VPS33B. Scale bar = 100 μm. (**C**)

*Figure 9 continued on next page*

*Figure 9 continued*

Immunohistochemistry of 5 µm thick sequential lung sections taken from lungs classified as control (Control 1). Sections were stained with hematoxylin and eosin (H&E), collagen-I (ColI), integrin α11, VPS33B. Scale bar = 100 µm.

The online version of this article includes the following figure supplement(s) for figure 9:

**Figure supplement 1.** Immunohistochemistry of 5 µm thick sequential lung sections taken from three additional idiopathic pulmonary fibrosis (IPF) patients (IPF2, IPF3, IPF4).

**Figure supplement 2.** Immunohistochemistry of 5 µm thick sequential lung sections taken from an additional lung classified as control (Control 2).

## Discussion

In this study we have identified an endocytic recycling mechanism for type I collagen fibrillogenesis that is under circadian regulation. Integral to this process is VPS33B (a circadian clock-regulated endosomal tethering molecule) and integrin α11 subunit (a collagen-binding transmembrane receptor when partnered with integrin β1 subunit). Collagen-I co-traffics with VPS33B to the plasma membrane for fibrillogenesis, which requires integrin α11. These proteins are all enhanced at active sites of collagen pathologies such as fibrosis and chronic skin wounds, suggestive of a common disease mechanism.

Previous research (*Madsen et al., 2007*; *Knowles et al., 1991*; *Everts et al., 1985*; *Arora et al., 2013*; *Vijayan et al., 2014*) has focused on degradation or signaling as the endpoint of collagen-I endocytosis. However, our results show that collagen-I endocytosis contributes toward fibril formation, which is reduced when endocytosis is inhibited. The decision between degrading or recycling endocytosed collagen may depend on cell type, microenvironment, or collagen type itself. In complex environments, e.g., a wound, degradation of damaged collagen molecules, and recycling of structurally sound collagen molecules could be beneficial. A caveat in this present study is the use of Dyngo4a to inhibit endocytosis. It was demonstrated that there may be non-dynamin targeting effects with Dyngo4a when comparing the effects of dynamin triple knockout and drug treatment (e.g. fluid-phase endocytosis and membrane ruffling) (*Park et al., 2013*) here, we have shown that collagen uptake is likely through receptor-mediated and not fluid-phase endocytosis, although whether periphery membrane ruffling alters collagen deposition remains to be investigated. As total secretion did not appear to be affected by Dyngo4a treatment, as indicated by Ponceau stain on western blots of CM, we take this to suggest that Dyngo4a in this study is not affecting overall secretion and is mostly acting on endocytosis. Further studies targeting multiple endocytosis routes, either through genetic manipulation or drug treatments, will help shed light on how collagen uptake is controlled, and if different forms of collagen (i.e. protomers, fibril) require different mechanisms. Nonetheless, the requirement of VPS33B for fibril assembly confirms the previously observed rhythmicity of collagen fibril formation by fibroblasts, in vivo and in vitro (*Chang et al., 2020*). Our discovery that the endosomal system is involved in fibril assembly explains how collagen fibrils can be assembled in the absence of endogenous collagen synthesis, namely, if collagen can be retrieved from the extracellular space. We also observed a delay between maximum uptake and maximum fibril numbers. A possible function of this delay may be to increase the concentration of collagen-I in readiness for more efficient fibril initiation. This is supported by our findings that fibroblasts can utilize a solution of Cy3-colI lower than the 0.4 µg/mL critical concentration for fibril formation to assemble fibrils. Of note, the comparison between uptake and fibril numbers are not directly from the same experiments due to experimental limitations; the Cy3/Cy5-colI uptake we observed did not reflect total collagen-I levels, as endogenous collage-I was present as well. Nonetheless, we postulate that the fates of the collagen-I that was taken up by fibroblasts are likely: (1) recycled for fibril formation, (2) re-secreted as protomers, or if they are damaged, (3) degradation. Future research will be required to determine the molecular mechanisms that determines these fates. Surface biotin-labeling in fibroblasts identified two integrin α subunits (α1 and α11) to be absent in VPSko cells which could not make collagen fibrils. There are four known collagen-binding integrins (α1β1, α2β1, α10β1, α11β1), where α2β1 and α11β1 have affinity for fibrillar collagens and are expressed in fibroblasts in vivo (*Zeltz and Gullberg, 2016*). α1β1, in contrast, has been demonstrated in vitro to display higher affinity for collagen-IV than collagen-I (*Eble et al., 1993*) it has a wide expression pattern in vivo, including basement membrane-associated smooth muscle cells, strongly suggesting that collagen-IV is the major ligand for α1β1 integrin instead. Thus, we focused on the effects of integrin α11 subunit. Interestingly, both α1 and α11 subunits were also reduced in VPSoe cells as detected by mass spectrometry; however, validation by western blots

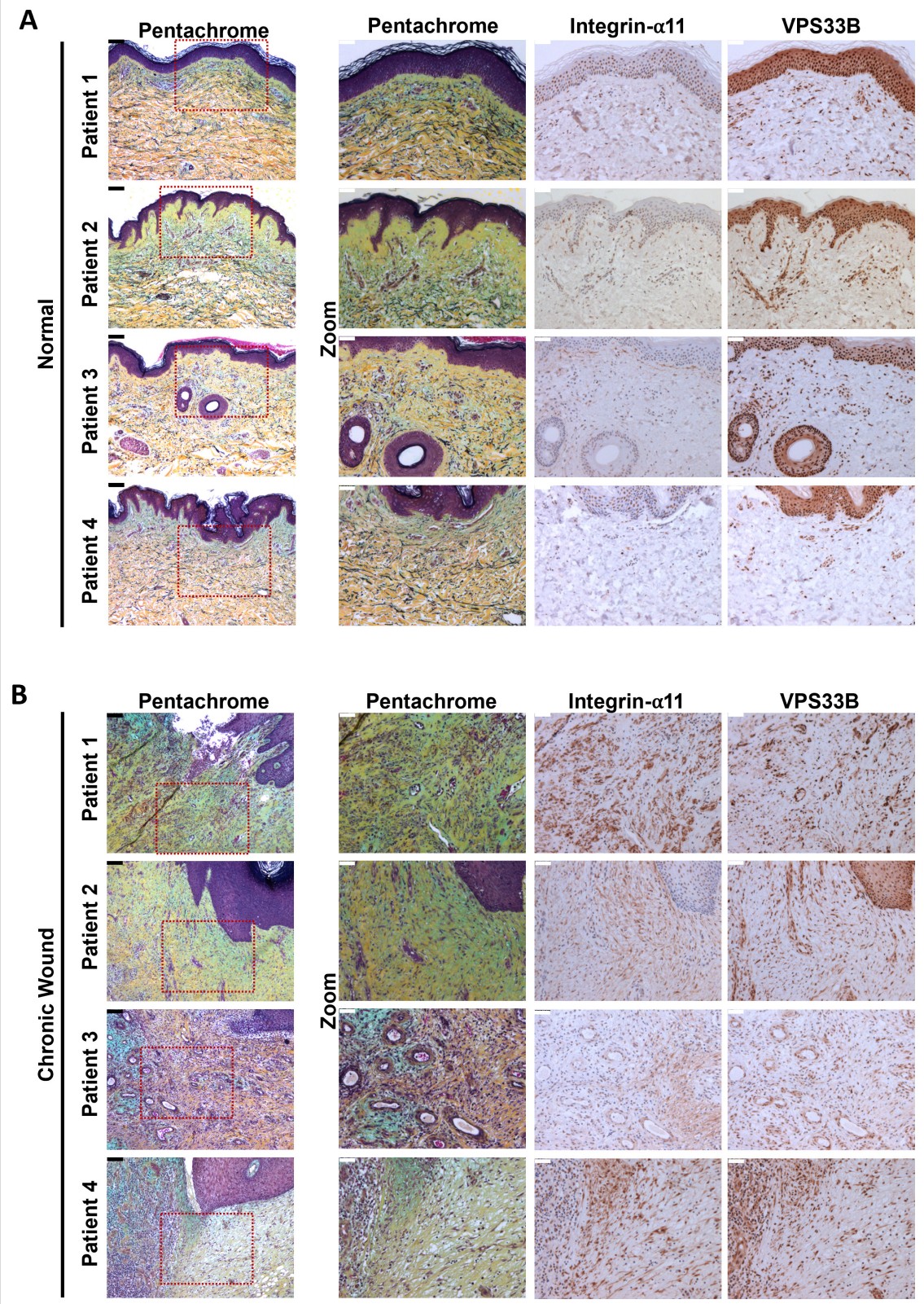

**Figure 10.** Proteins responsible for collagen fibrillogenesis are also co-localized to diseased areas of chronic skin wounds. (**A**) Immunohistochemistry of 5 µm thick sequential skin sections taken from normal skin regions of patients with chronic skin wounds (Patient 1, Patient 2, Patient 3, Patient 4). Sections were stained with pentachrome, integrin α11, VPS33B. Scale bars positioned in top left corner: black (unzoomed pentachrome)=100 µm, white (zoomed sections)=50 µm. (**B**) Immunohistochemistry of 5 µm thick sequential skin sections taken from the chronic wound areas from patients with

*Figure 10 continued on next page*

*Figure 10 continued*

chronic skin wounds (Patient 1, Patient 2, Patient 3, Patient 4). Sections were stained with pentachrome, integrin α11, VPS33B. Scale bars positioned in top left corner: black (unzoomed pentachrome)=100 μm, white (zoomed sections)=50 μm.

---

indicated that protein levels of integrin α11 follows VPS33B and collagen fibril numbers. We postulate that this discrepancy detected through MS might be due to integrin dynamics at the plasma membrane. Regardless, VPS33B and integrin α11β1 are essential for targeting collagen-I to fibril assembly and not for the secretion of soluble triple helical collagen-I molecules (i.e. protomers); there is also no evidence to support an active role for VPS33B or α11β1 in collagen endocytosis. We have also demonstrated involvement of VPS33B and integrin α11 subunit in IPF, particularly at sites of disease progression (fibroblastic foci). The presence of integrin α11 subunit confirms a recent study utilizing spatial proteomics to decipher regions of IPF tissues, where in addition to integrin α11 the authors also identified proteins involved in collagen biogenesis specific to the fibroblastic foci (*Herrera et al., 2022*). Crucially, in our study, levels of *Col1a1* transcript were unchanged between control and IPF fibroblasts, and IHC demonstrated that collagen-I and VPS33B were prevalent in control lungs. This highlights that the production of collagen-I does not equate to fibril formation; indeed, here VPS33B is required for normal collagen homeostasis in the lung, and it is the combination of VPS33B and integrin α11 subunit, coupled with elevated endocytic recycling of exogenous collagen at the fibroblastic focus, that promotes excess collagen-I fibril assemblies, which is a hallmark of fibrosis. Furthermore, this relationship between excessive collagen-I, VPS33B, and integrin α11 is also observed in chronic skin wounds, where, similar to lung fibrosis, there is a chronic unresolved wound-healing response; this is suggestive of a common pathological molecular pathway between the two organs, based on enhanced endocytic recycling of collagen-I protomers directed to fibrillogenesis.

The discovery that VPS33B is required for collagen fibril formation but not collagen protomer secretion highlights an important distinction between 'collagen secretion' and 'collagen fibrillogenesis', especially in the context of collagen pathologies (e.g. fibrosis, cancer metastasis). In this context, it is crucial to keep in mind that elevated collagen levels (i.e. collagen transcripts or protomer levels) does not equate to the formation of an insoluble fibrillar network. This distinction was also suggested in a recent study on *Pten*-knockout mammary fibroblasts, where SPARC protein acts via fibronectin to affect collagen fibrillogenesis but not secretion (*Jones et al., 2021*). Of note, while SPARC and fibronectin were detected in our previously reported time-series mass spectrometry analyses (*Chang et al., 2020*), they were not circadian clock rhythmic in tendon.

The nucleation of a collagen-I fibril is expected to involve collagen-V, a minor fibril-forming collagen that is necessary for the appearance of collagen-I fibrils in vivo (*Wenstrup et al., 2011*). Another long-standing view is that fibronectin may tether collagen to a fibronectin-binding integrin and thereby function as a proteolytically cleavable anchor (*Taylor et al., 2015*, reviewed by *Musiime et al., 2021*). While the present study did not focus on the role of fibronectin, there is a slight indication that fibronectin is decreased when collagen-I is knocked down, although IF staining suggested that collagen-I and fibronectin do not completely overlap, and that inhibition of endocytosis did not affect fibronectin fibril numbers. Further, biotin-surface labeling mass spectrometry analysis in the present study indicated that the levels of the major fibronectin-binding integrins (integrins α5β1, αVβ3) were not drastically altered between control, VPSko, and VPSoe fibroblasts, suggesting that VPS33B controls collagen fibrillogenesis via a fibronectin-independent route. We have recently demonstrated that integrin α11β1 localizes to fibrillar adhesions and contributes to collagen assembly in a mechano-regulated and tensin-1-dependent manner, independent of α5β1 and fibronectin (*Musiime et al., 2024*). In these studies, the α11β1-mediated collagen assembly did not appear to depend on endocytosis. In the future it will be important to establish which factors govern endocytosis-dependent and endocytosis-independent α11β1-mediated collagen fibrillogenesis. Interestingly, in fibronectin knockout liver cells, exogenous collagen-V could be added to initiate collagen-I fibrillogenesis (*Moriya et al., 2011*), which suggests that other matrix proteins could also be recycled in a similar manner as collagen-I. While collagen α1(V) could only be detected in surface biotin-labeled control but not VPSko or VPSoe cells, collagen α2(V) was only detected in VPSoe cells. Of note, we have previously shown that collagen α2(V) protein was rhythmic and in phase with collagen α1(I) and collagen α2(I). Whether collagen-V co-traffics with collagen-I in VPS33B-compartments to fibrillogenic sites at the cell periphery, or collagen-V is directed to the nucleation site by another route, is an important question to

address in future studies. Indeed, endocytic recycling of other factors required for collagen fibrillogenesis (e.g. other nucleation factors such as collagen-V, collagen-XI, integrins), in addition to collagen-I protomers, may be required for fibril assembly by fibroblasts.

Previous research has demonstrated that collagen-I can be secreted within minutes (*Canty et al., 2004*), while other studies have demonstrated a relatively slow emergence of collagen-I fibrils over days in culture (*Pickard et al., 2018*). The delay in assembly of fibrils could be due to the requirement to reach the threshold concentration of collagen needed for fibrillogenesis (*Kadler et al., 1987*). Procollagen-I can be converted to collagen-I within the secretory pathway (*Canty et al., 2004*; *Canty-Laird et al., 2012*) in a process that requires giantin for intracellular N-terminal processing of procollagen-I (*Stevenson et al., 2021*). Thus, it is possible that triple helical collagen-I protomers destined for fibril formation are: (1) directed straight to a VPS33B-positive compartment from the Golgi apparatus, or (2) quickly secreted and then recaptured, via endocytosis, prior to recycling for fibrillogenesis. Such a mechanism would provide the cell with additional opportunities to determine the number of fibrils, the rate at which they are to be initiated, and their required growth in the ECM (see *Figure 11*). We have not identified the specific proteins that mediate collagen-I uptake into the cell, however as inhibition of clathrin-mediated endocytosis did not completely inhibit collagen endocytosis, it is likely that collagen-I has multiple routes to enter the cells (*Madsen et al., 2017*; *Rainero, 2016*; *Lee et al., 2014*; *Leitinger, 2014*). With regard to the fibrillogenesis route, although a recent model of VPS33B/VIPAS39 protein complexes does not consider the possibility that VPS33B can associate directly with vesicular membranes (*Liu et al., 2023*), our in vitro studies suggest its hydrophobic C-terminal region can act as a tail-anchor domain. Importantly, the proposed model by Liu et al. of the VPS33B/VIPAS39 complex suggests the C-terminus of VPS33B would be available for membrane association, consistent with our in vitro findings. Further studies will be required to establish whether differences between our models reflect alternative experimental systems (i.e. yeast vs. mammalian), and/or the behavior of VPS33B alone vs. in complex with VIPAS39. We found that endogenous VIPAS39 encases collagen-I in intracellular puncta, consistent with a model where VPS33B and VIPAS39 co-traffic with collagen-I at the same endosomal compartment (see also *Liu et al., 2023*). Our data suggest two membrane-bound populations of VPS33B, the majority with a hairpin-like structure and cytosolic C-terminus and a minority that spans the membrane with the C-terminus inside the lumen. Whether or not alternative topologies of VSP33B are associated with different biological functions remains to be determined. Nonetheless, here we have identified a previously unknown role for endocytosis in collagen fibrillogenesis, which is exploited in collagen pathologies.

## Materials and methods

### Procurement of human lung tissue and human lung-derived fibroblasts

The use of human lung tissue was approved by the National Health Research Authority with patient consent (NRES14/NW/0260). The specimens used for this study met the criteria for IPF diagnosis (*Lynch et al., 2018*), as previously described (*Herrera et al., 2020*). Four IPF patient samples and two control lung samples were used in this study. The patient-derived fibroblasts, isolated from five IPF patient samples and five control samples used here, were a kind gift from Prof. Peter Bitterman and Prof. Craig Henke at the University of Minnesota. Briefly, control and IPF-derived fibroblasts were isolated by allowing fibroblasts to propagate from lung tissue explants as previously described [*Herrera et al., 2018*]. The fibroblasts used in the described experiments were between passages 4 and 6.

### Procurement of human skin tissues

Human chronic wound samples (as defined by *Kyaw et al., 2018*) were collected alongside healthy skin samples from four consenting patients undergoing surgical debridement and reconstruction at Manchester University NHS Foundation Trust. All procedures were approved by the National Health Research Authority through the ComplexWounds@Manchester Biobank (NRES 18/NW/0847).

### Mice

The care and use of all mice in this study was carried out in accordance with the UK Home Office regulations, UK Animals (Scientific Procedures) Act of 1986 under the Home Office Licence (#70/8858

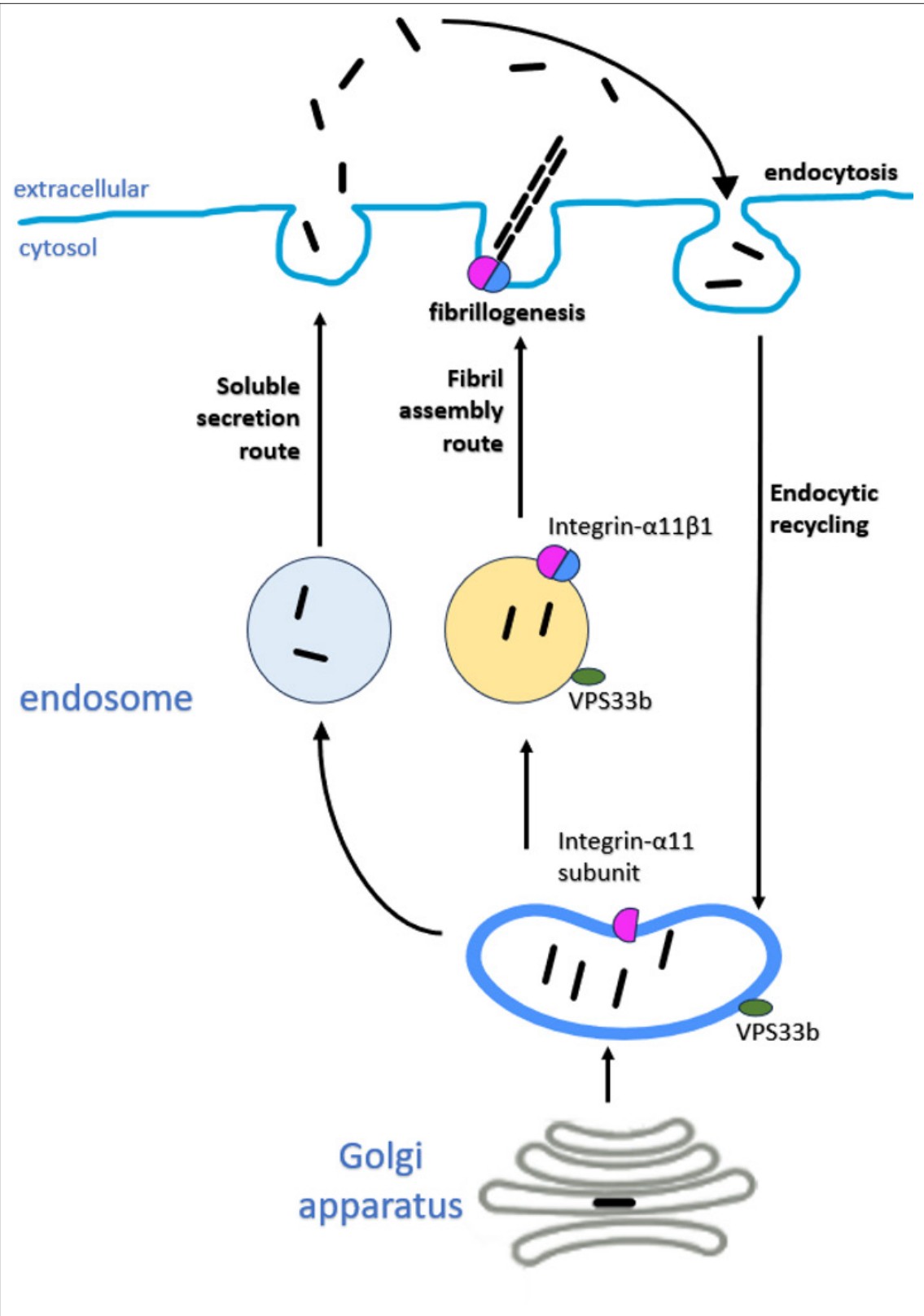

**Figure 11.** Proposed working model of collagen homeostasis in fibroblasts. Endogenous collagen is either secreted as protomers (soluble secretion route, not circadian rhythmic) or made into fibrils (fibril assembly route, circadian rhythmic). Secreted collagen protomers can be captured by cells through endocytosis (circadian rhythmic) and recycled to make new fibrils. Integrin α11 and VPS33b direct collagen to fibril formation.

or I045CA465). The permission included the generation of conditional knockout animals. C57 bl/6 Col1a1-fl/fl mice (*Li et al., 2017*; *Yang et al., 2013*), kind gift from Dr Kevin K. Kim (Pulmonary and Critical Care Medicine, University of Michigan Medical School, Ann Arbor, USA) were crossed with C57 bl/6 Col1a2-ER$^{T2}$ mice (kind gift from Prof. George Bou-Gharios, University of Liverpool, UK) to generate a tamoxifen-inducible Col1a2-ER$^{T2}$:: Col1a1-fl/fl strain. Ten-week-old male Col1a2-ER$^{T2}$:: Col1a1-fl/fl mice were treated with tamoxifen by intraperitoneal injections according to approved UK Home Office regulations (Project licence number I045CA465; Personal licence number P08B76E2B). Mice were humanely sacrificed by experienced personal in the animal facility.

## Cell culture

Unless otherwise stated, all cell culture reagents were obtained from Gibco, and all cells were maintained at 37°C, 5% $CO_2$ in a humidified incubator. Immortalized murine tail tendon fibroblasts (iTTFs, isolated from 8- to 10-week-old C57BL/6 male mice [*Chang et al., 2020*]), and NIH3T3 cells with Dendra2-tagged collagen-I expression (Dendra2-3T3 [*Pickard et al., 2018*]) were cultured in DMEM/F12 with sodium bicarbonate and L-glutamine (supplemented with 10% fetal calf serum [FCS], 200 μM L-ascorbate-2-phosphate [Sigma], and 10,000 U/mL penicillin/streptomycin), and high-glucose DMEM with sodium bicarbonate and L-glutamine (supplemented with 10% newborn calf serum, 200 μM L-ascorbate-2-phosphate, and 10,000 U/mL penicillin/streptomycin) media, respectively. Human patient-derived fibroblasts were cultured in low glucose DMEM with sodium bicarbonate and L-glutamine (supplemented with 10% FCS, 200 μM L-ascorbate-2-phosphate, and 10,000 U/mL penicillin/streptomycin). HEK293T cells were cultured in DMEM with sodium bicarbonate and L-glutamine, supplemented with 10% FCS.

To synchronize cells in culture, 100 μM dexamethasone was added to each sample, then incubated for an hour before the media was removed and replaced with fresh culture media. This marks t=0 in the experiments (i.e. post-synchronization time 0, or PS0).

Murine tail tendons were isolated by first removing the skin from the mouse tails by cutting it from tail root and degloving the whole tail. Once tendons become exposed, tendon fibers were pulled out using sterile forceps, breaking every other vertebra. The vertebra was then trimmed with scalpels leaving just the tail tendons. These were then placed on a cell culture insert and equilibrated in full DMEM/F12 culture media (see above section) overnight, before the fluorescently labeled collagen were added at 5 μg/mL for 3 days, washed extensively with PBS, and then removed before embedding in low-melting-point agarose for confocal imaging (Leica SP8 upright confocal microscopy). For the pulse-chase experiment, Cy3-labeled collagen was first added and incubated, before tendons were washed and 5FAM-labeled collagen was added for a further 48 hr prior to imaging.

## CRISPR-Cas9-mediated knockout

iTTFs were treated with CRISPR-Cas9 to delete *VPS33B* gene as previously described (*Chang et al., 2020*). Gene knockout was confirmed by western blotting and qPCR.

## Constructs

The cDNA for mouse VPS33B (UniProt: P59016) was purchased from GenScript (OMu07060D). Sec61β modified with a C-terminal OPG2 tag (residues 1–18 of bovine rhodopsin [UniProt: P02699]) was as previously described (*McKenna et al., 2016*). An artificial N-glycosylation site (VPS33B-A606T) and OPG2-tag (VPS33B-OPG2) were incorporated into parent VPS33B by site-directed mutagenesis (Stratagene QuikChange, Agilent Technologies) using the relevant forward and reverse primers (Integrated DNA Technologies; see reagents table) and confirmed by DNA sequencing (GATC, Eurofins Genomics). Linear DNA templates were generated by PCR and mRNA transcribed using T7 polymerase or SP6 RNA polymerase as appropriate (see reagents table). In order to improve the signal intensity of all radiolabeled proteins, an additional five methionine residues (5M) were appended to the C-terminus of all VPS33B linear DNA templates by PCR. All primer combinations used for mutagenesis and PCR are listed in the tables below.

## Transfection and stable infection of overexpression vectors in cells

Three different vectors were used to stably overexpress VPS33B protein in iTTF and 3T3 cells – untagged VPS33B overexpression (with red fluorescent protein expression as selection), N-terminus

BFP-tagged VPS33B overexpression, and C-terminus BFP-tagged VPS33B overexpression. All proteins were cloned into pLV V5-Luciferase expression vector, where luciferase was excised from the backbone prior to gel purification, or into pCMV expression vector, followed by Gibson Assembly (NEB) (*Campeau et al., 2009*). Lentiviral particles were generated in HEK293T cells, and cells were infected in the presence of 8 µg/mL polybrene (Millipore). Cells were then sorted using Flow Cytometry to isolate RFP-positive or BFP-positive cells. Alternatively, cells were treated with puromycin (Sigma) or G418 (Sigma) depending on selection marker on the vectors.

## Fluorescence labeling of collagen-I

Cy3 or Cy5 NHS-ester dyes (Sigma) were used to fluorescently label rat tail collagen-I (Corning), using a previously described method (*Doyle, 2018*). Briefly, 3 mg/mL collagen-I gels were made, incubated with 50 mM borate buffer (pH 9) for 15 min, followed with incubation with either Cy3-ester or Cy5-ester (Sigma) dissolved in borate buffer in dark overnight at 4°C, gently rocking. Dyes were then aspirated and 50 mM Tris (pH 7.5) were added to quench the dye reaction, and incubated in the dark rocking for 10 min. Gels were washed with PBS (with calcium and magnesium ions) 6×, incubating for at least 30 min each wash. The gels were then resolubilized using 500 mM acetic acid and dialyzed in 20 mM acetic acid.

## Circular dichroism of rat tail collagen-I

Circular dichroism spectra of rat tail collagen in 10 mM acetic acid was recorded on a Jasco J810 spectrometer using 0.2 mm quartz coverslip sample holders. Spectra were recorded using approximately 0.5 mg/mL rat tail collagen for WT, Cy3 labeled and Cy5-labeled collagens. The melting curves were performed by monitoring at 223 nm only which is the positive triple helical peak and recording between 30°C and 70°C, and capturing data points every 30 s with a temperature increment of 1° every 60 s in a 1 mm quartz cuvette. All outputs are recorded in machine units rather than being converted to molar units because of the difficulty in ascertaining accurate concentrations of labeled collagen.

## Mass photometry of rat tail collagen-I

The Refeyn mass photometer was used to assess the mass of the collagen molecules. The instrument was calibrated with BSA and PTX3 with a mass range of 67 kDa, 135 kDa, and 340 kDa. The WT, Cy3-labeled and Cy5-labeled collagens were diluted to ~20 nM in 10 mM acetic acid and counts read over the course of 60 s. The ratiometric contrast was converted to mass using the calibration standards.

## RNA isolation and quantitative real-time PCR

RNA was isolated using TRIzol Reagent (Thermo Fisher Scientific) following the manufacturer's protocol, and concentration was measured using a NanoDrop OneC (Thermo Fisher Scientific). Complementary DNA was synthesized from 1 µg RNA using TaqMan Reverse Transcription kit (Applied Biosystems) according to the manufacturer's instructions.

SensiFAST SYBR kit reagents were used in qPCRs. Primer sequences can be found in reagents section.

## Protein extraction and western blotting

For lysate experiments, proteins were extracted using urea buffer (8 M urea, 50 mM Tris-HCl pH 7.5, supplemented with protease inhibitors and 0.1% β-mercaptoethanol). For CM, cells were plated out at 200,000 cells per six-well plate, and left for 48–72 hr before 250 µL was sampled. Samples were mixed with 4xSDS loading buffer with 0.1% β-mercaptoethanol and boiled at 95°C for 5 min. The proteins were separated on either NuPAGE Novex 10% polyacrylamide Bis-Tris gels with 1XNuPAGE MOPS SDS buffer or 6% Tris-glycine gels with 1XTris-glycine running buffer (all Thermo Fisher Scientific), and transferred onto polyvinylidene difluoride membranes (GE Healthcare). The membranes were blocked in 5% skimmed milk powder in PBS containing 0.01% Tween 20. Antibodies were diluted in 2.5% skimmed milk powder in PBS containing 0.01% Tween 20. The primary antibodies used were: rabbit polyclonal antibody (pAb) to collagen-I (1:1000; Gentaur), mouse mAb to vinculin (1:1000; Millipore), rabbit pAb to integrin α11 subunit (1:1000; see *Popova et al., 2004*), and mouse mAb to VPS33B (1:500; Proteintech). Horseradish-peroxidase-conjugated antibodies and Pierce ECL western blotting

substrate (both from Thermo Fisher Scientific) were used and reactivity was detected on GelDoc imager (Bio-Rad). Alternatively, Li-Cor goat-anti-mouse 680, mouse-anti-rabbit 800 were used and reactivity detected on an Odyssey Clx imager.

## In vitro ER import assays

Translation and ER import assays were performed in nuclease-treated rabbit reticulocyte lysate (Promega) as previously described (*O'Keefe et al., 2022*; *O'Keefe et al., 2021b*; *O'Keefe et al., 2021a*): briefly, in the presence of EasyTag EXPRESS $^{35}$S Protein Labelling Mix containing [$^{35}$S] methionine (Perkin Elmer) (0.533 MBq; 30.15 TBq/mmol), 25 μM amino acids minus methionine (Promega), 6.5% (vol/vol) nuclease-treated ER-derived canine pancreatic microsomes (from stock with $OD_{280}$=44/mL) or an equivalent volume of water, and 10% (vol/vol) of in vitro transcribed mRNA (~1 μg/μL) encoding relevant precursor proteins. Translation reactions for Sec61βOPG2 (20 μL; 1×) were performed at 30°C for 30 min whereas VPS33B and its variants (20 μL; 2×) were performed at 30°C for 1 hr. Irrespective of the precursor protein, all translation reactions were finished by incubating with 0.1 mM puromycin for 10 min at 30°C to ensure translation termination and the ribosomal release of newly synthesized proteins prior to analysis. Microsomal membrane-associated fractions were recovered by centrifugation through an 80 μL high-salt cushion (0.75 M sucrose, 0.5 M KOAc, 5 mM $Mg(OAc)_2$ and 50 mM HEPES-KOH, pH 7.9) at 100,000×$g$ for 10 min at 4°C, the pellet suspended directly in SDS sample buffer and, where indicated, treated with 1000 U of endoglycosidase Hf (New England Biolabs, P0703S). All samples were denatured for 10 min at 70°C and resolved by SDS-PAGE (10% or 16% PAGE, 120 V, 120–150 min). Gels were fixed for 5 min (20% MeOH, 10% AcOH), dried for 2 hr at 65°C, and radiolabeled products were visualized using a Typhoon FLA-700 (GE Healthcare) following exposure to a phosphorimaging plate for 24–72 hr. Images were opened using Adobe Photoshop and annotated using Adobe Illustrator.

## Flow cytometry and imaging

Cy3- or Cy5-tagged collagen was added to cells and incubated at 37°C, 5% $CO_2$ in a humidified incubator for predetermined lengths of times before washing in PBS, trypsinized, spun down at 2500 rpm for 3 min at 4°C, and resuspended in PBS on ice. For labeled dextran experiment, 200 μg/mL Oregon Green 488-labeled 70 kDa dextran and 10 μg/mL Cy5-colI were added to the cells together and incubated for 1 hr; for unlabeled collagen-I saturation experiments, 100 μg/mL rat tail collagen-I were added together with 10 μg/mL labeled collagen-I and incubated for 1 hr. Cells were then processed as described above. Cells were analyzed for Cy3-/Cy5-collagen/488-dextran uptake using LSRFortessa (BD Biosciences). For imaging, cells were prepared as described, and analyzed on Amnis Image-Stream$^X$Mk II (Luminex).

## Decellularization of cells to obtain ECM

Cells were seeded out at 50,000 in a six-well plate and cultured for 7 days before decellularization. Extraction buffer (20 mM $NH_4OH$, 0.5% Triton X-100 in PBS) was gently added to cells and incubated for 2 min. Lysates were aspirated and the matrix remaining in the dish were washed gently twice with PBS, before being scraped off into $ddH_2O$ for further processing.

## Hydroxyproline assay

Samples were incubated overnight in 6 M HCl (diluted in $ddH_2O$ [Fluka]; approximately 1 mL per 20 mg of sample) in screw-top tubes (StarLab) in a sand-filled heating block at 100°C covered with aluminum foil. The tubes were then allowed to cool down and centrifuged at 12,000×$g$ for 3 min. Hydroxyproline standards were prepared (starting at 0.25 mg/mL; diluted in $ddH_2O$) and serially diluted with 6 M HCl. Each sample and standard (50 μL) were transferred into fresh Eppendorf tubes, and 450 μL chloramine T reagent (0.127 g chloramine T in 50% N-propanol diluted with $ddH_2O$; made up to 10 mL with acetate citrate buffer [120 g sodium acetate trihydrate, 46 g citric acid, 12 mL glacial acetic acid, 34 g sodium hydroxide] adjusted to pH 6.5 and then made to 1 L with $dH_2O$; all reagents from Sigma) was added to each tube and incubated at room temperature for 25 min. Ehrlich's reagent (500 μL; 1.5 g 4-dimethylaminobenzaldehyde diluted in 10 mL N-propanol:perchloric acid (2:1)) was added to each reaction tube and incubated at 65°C for 10 min and then the absorbance at 558 nm was measured for 100 μL of each sample in a 96-well plate format.

## Electron microscopy

Unless otherwise stated, incubation and washes after EM fixation were done at room temperature. Cells were plated on top of ACLAR films and allows to deposit matrix for 7 days. The ACLAR was then fixed in 2% glutaraldehyde/100 mM phosphate buffer (pH 7.2) for at least 2 hr and washed in ddH$_2$O 3×5 min. The samples were then transferred to 2% osmium (vol/vol)/1.5% potassium ferrocyanide (wt/vol) in cacodylate buffer (100 mM, pH 7.2) and further fixed for 1 hr, followed by extensive washing in ddH$_2$O. This was followed by 40 min of incubation in 1% tannic acid (wt/vol) in 100 mM cacodylate buffer, and then extensive washes in ddH$_2$O, and placed in 2% osmium tetroxide for 40 min. This was followed by extensive washes in ddH$_2$O. Samples were incubated with 1% uranyl acetate (aqueous) at 4°C for at least 16 hr, and then washed again in ddH$_2$O. Samples were then dehydrated in graded ethanol in the following regime: 30%, 50%, 70%, 90% (all vol/vol in ddH$_2$O) for 8 min at each step. Samples were then washed 4×8 min each in 100% ethanol, and transferred to pure acetone for 10 min. The samples were then infiltrated in graded series of Agar100Hard in acetone (all vol/vol) in the following regime: 30% for 1 hr, 50% for 1 hr, 75% for overnight (16 hr), 100% for 5 hr. Samples were then transferred to fresh 100% Agar100Hard in labeled moulds and allowed to cure at 60°C for 72 hr. Sections (80 nm) were cut and examined using a Tecnai 12 BioTwin electron microscope.

## Surface biotinylation-pulldown mass spectrometry

Cells were grown in six-well plates for 72 hr prior to biotinylating. Briefly, cells around 90% confluence were kept on ice and washed in ice-cold PBS, following with incubation with ice-cold biotinylating reagent (prepared fresh, 200 µg/mL in PBS, pH 7.8) for 30 min at 4°C gently shaking. Cells were then washed twice in ice-cold TBS (50 mM Tris, 100 mM NaCl, pH 7.5), and incubated in TBS for 10 min at 4°C. Cells were then lysed in ice-cold 1% Triton-X (in PBS, with protease inhibitors) and lysates cleared by centrifugation at 13,000×*g* for 10 min at 4°C. Supernatant was then transferred to a fresh tube and 1/5 of the lysates were kept as a reference sample. Streptavidin-Sepharose beads was aliquoted into each sample and incubated for 30 min at 4°C rotating. Beads were then washed 3× in ice-cold PBS supplemented with 1% Triton-X, followed by one final wash in ice-cold PBS and boiled at 95°C for 10 min in 2× sample loading buffer, followed by centrifugation at 13,000×*g* for 5 min followed by sample preparation for mass spectrometry analysis. The protocol used for sample preparation was as described previously (*Herrera et al., 2020*). Mass spectrometry results files were exported into Proteome Discoverer for identification and spectral counting. All searches included the fixed modification for carbamidomethylation on cysteine residues resulting from IAA treatment to prevent cysteine bonding. The variable modifications included in the search were oxidized methionine (monoisotopic mass change, +15.955 Da); hydroxylation of asparagine, aspartic acid, proline, or lysine (monoisotopic mass change, +15.955 Da); and phosphorylation of threonine, serine, and tyrosine (79.966 Da). A maximum of two missed cleavages per peptide was allowed. The minimum precursor mass was set to 350 Da with a maximum of 5000. Precursor mass tolerance was set to 10 ppm, fragment mass tolerance was 0.02 Da, and minimum peptide length was 6. Peptides were searched against the Swissprot database using Sequest HT with a maximum false discovery rate of 1%. Proteins required a minimum FDR of 1% and were filtered to remove known contaminants and to have at least two unique peptides. Missing values were assumed to be due to low abundance. For spectral counting, enriched samples were normalized and spectral matches were compared. Peptide spectral matches were only included if they were calculated to have a q-value of less than 0.01. Mass spectrometry data are available via ProteomeXchange with identifier PXD034394.

## Imaging and IF

For fluorescence live imaging of Cy3-colI internalization, iTTF cells were seeded for 24 hr onto Ibidi µ-plates and stained using CellMask Green Plasma Membrane Stain (Invitrogen C37608) according to the manufacturer's protocol. Cy3-colI was added to cells and images were collected in a 37°C chamber using a Zeiss 3i spinning disk (CSU-X1, Yokagowa) confocal microscope with a ×63/1.40 Plan-Apochromat oil objective. 7.5 µm z-stacks with a step size of 0.5 µm were captured at 2 min intervals using 488 nm (100 power, 150 ms exposure) and 561 nm (100 power, 100 ms exposure) lasers with SlideBook 6.0 software (3i) and a front illuminated Prime sCMOS camera. Imaris 9.9.1 software was then used to generate 3D reconstructions.

For fixed IF imaging, cells plated on coverslips or Ibidi μ-plates were fixed with 100% methanol at –20°C and then permeabilized with 0.2% Triton-X in PBS. Primary antibodies used were as follows: rabbit pAb collagen-I (1:400, Gentaur OARA02579), rabbit pAb VIPAS (1:50, Proteintech 20771-1-AP), mouse mAb FN1 (1:400, Sigma F6140 or Abcam ab6328). Secondary antibodies conjugated to Alexa Fluor 488, Alexa Fluor 647, Cy3, and Cy5 were used (Thermo Scientific), and nuclei were counterstained with DAPI (Sigma). Coverslips were mounted using Fluoromount G (Southern Biotech). Images were collected using a Leica SP8 inverted confocal microscope (Leica) using an ×63/0.50 Plan Fluotar objective. The confocal settings were as follows: pinhole, 1 Airy unit; scan speed, 400 Hz bidirectional; format 1024×1024 or 512×512. Images were collected using photomultiplier tube detectors with the following detection mirror settings: DAPI, 410–483; Alexa Fluor 488, 498–545 nm; Cy3, 565–623 nm; Cy5, 638–750 nm using the 405 nm, 488 nm, 540 nm, 640 nm laser lines. Alternatively, images were collected using EVOS M7000 (Thermo Fisher) using ×40/1.3 NA oil objective. Images were collected in a sequential manner to minimize bleed-through between channels.

For fluorescence live imaging with Rab5-GFP, cells were plated onto Ibidi μ-plates and imaged using Zeiss LSM880 NLO (Zeiss). For split-GFP experiments, cells were seeded for 24 hr onto Ibidi μ-plates before imaging with an Olympus IXplore SpinSR (Olympus) with ×100 oil magnification. Prior to imaging, media was changed to FluoroBrite media with the appropriate supplements.

## Immunohistochemistry

Human lung samples were formalin-fixed and paraffin-embedded (FFPE). 5 μm FFPE sections were obtained and mounted onto Superfrost Plus (Thermo Scientific) slides and subjected to antigen heat retrieval using citrate buffer (Abcam, ab208572), in a pre-heated steam bath for 20 min, before cooling to room temperature in a water bath for 20 min. Slides were then treated with 3–4% hydrogen peroxide (Leica Biosystems RE7101) for 10 min, blocked in SuperBlock (TBS) blocking buffer for a minimum of 1 hr (Thermo Scientific; 37581), and probed with primary antibodies overnight at 4°C in 10% SuperBlock solution in Tris Buffered Saline Tween 20 solution (TBS-T, pH 7.6). Primary antibodies used were as follows: Collagen-I A1/A2 (Rockland Immunochemicals Inc 600-401-103.0.5), integrin α11 (integrin α11 mAb 210F4 [*Smeland et al., 2020*]), VPS33B (Atlas antibodies).

After overnight incubation, specimens were subjected to Novolink Polymer Detection Systems (Leica Biosystems RE7270-RE, as per the manufacturer's recommendations), with multiple TBS-T washes. Sections were developed for 5 min with DAB Chromagen (Cell Signaling Technology, 11724/5) before being counterstained with hematoxylin for 1 min, followed by acid alcohol and blueing solution application. Slides were dehydrated through sequential ethanol and xylene before being coverslipped with Permount mounting medium (Thermo Scientific, SP15).

## Histological imaging

Stained slides were imaged using a DMC2900 Leica camera along with Leica Application Suite X software (Leica).

## Quantification and statistical analysis

Data are presented as the mean ± s.e.m. unless otherwise indicated in the figure legends. The sample number 'N' indicates the number of independent biological samples in each experiment, and 'n' indicates the number of technical repeats, and are indicated in the figure legends. Data were analyzed as described in the legends. The data analysis was not blinded, apart from quantification of fibril numbers over time, and differences were considered statistically significant at $p < 0.05$, using Student's t-test or one-way ANOVA, unless otherwise stated in the figure legends. Analyses were performed using GraphPad Prism 8 or 10.2.0 software. Significance levels are: $*p < 0.05$; $**p < 0.01$; $***p < 0.005$; $****p < 0.0001$. Where applicable, normality test was performed using the Shapiro-Wilk method. For periodicity, analysis was performed using the MetaCycle package (*Wu et al., 2016*) in the R computing environment (*Ihaka and Gentleman, 1996*) with the default parameters.

## Acknowledgements

The proteomics was performed at the Biological Mass Spectrometry Facility with the assistance of Stacey Warwood and Ronan O'Cualain, imaging was performed at the Bioimaging Facility, flow cytometry analyses were performed at the Flow Cytometry Facility with the assistance of Gareth Howell, all

in the Faculty of Biology, Medicine and Health (University of Manchester). This report is independent research supported by the North West Lung Centre Charity and National Institute for Health Research Clinical Research Facility at Manchester University NHS Foundation Trust. The views expressed in this publication are those of the author(s) and not necessarily those of the NHS, the North West Lung Centre Charity, National Institute for Health Research or the Department of Health. The authors would like to acknowledge the Manchester Allergy, Respiratory and Thoracic Surgery Biobank and the North West Lung Centre Charity for supporting this project. We would also like to thank the study participants for their contribution. The authors would like to acknowledge and thank Prof. David Stephens for his feedback on this manuscript. Wellcome Senior Investigator Award 110126/Z/15/Z (KEK). MRC Career Development Award MR/W016796/1 (JC). Wellcome Centre Award 203128/Z/16/Z (KEK). Wellcome Investigator Award in Science 204957/Z/16/Z (SH) Norwegian Centre of Excellence grant 223250 (DG). Norwegian Cancer Society grant 223052 (DG). Nasjonalföreningen for folkhelsen grant, project 16216 (DG). Wellcome Institutional Strategic Support fund 204796/Z/16/Z (AR, JW). Wellcome Translational Partnership Award 209741/Z/17/Z (AR, JW).

## Additional information

### Funding

| Funder | Grant reference number | Author |
|---|---|---|
| Medical Research Council | MR/W016796/1 | Joan Chang |
| Wellcome Trust | 110126 | Karl E Kadler |
| Wellcome Trust | 203128 | Karl E Kadler |
| Wellcome Trust | 204957 | Stephen High |
| Norwegian Centre of Excellence | 223250 | Donald Gullberg |
| Norwegian Cancer Society | 223052 | Donald Gullberg |
| Nasjonalföreningen for folkhelsen | 16216 | Donald Gullberg |
| Wellcome Trust | 204796 | Jason Wong Adam Reid |
| Wellcome Trust | 209741/Z/17/Z | Jason Wong Adam Reid |

The funders had no role in study design, data collection and interpretation, or the decision to submit the work for publication. For the purpose of Open Access, the authors have applied a CC BY public copyright license to any Author Accepted Manuscript version arising from this submission.

### Author contributions

Joan Chang, Conceptualization, Data curation, Formal analysis, Supervision, Funding acquisition, Validation, Investigation, Visualization, Methodology, Writing – original draft, Project administration, Writing – review and editing; Adam Pickard, Conceptualization, Investigation, Methodology, Writing – review and editing; Jeremy A Herrera, Validation, Investigation, Visualization, Methodology, Writing – review and editing; Sarah O'Keefe, Matthew Hartshorn, Yinhui Lu, Formal analysis, Investigation, Methodology, Writing – review and editing; Richa Garva, Formal analysis, Investigation, Visualization, Methodology, Writing – review and editing; Anna Hoyle, Data curation, Formal analysis, Validation, Investigation, Methodology, Writing – review and editing; Lewis Dingle, Investigation, Writing – review and editing; John Knox, Validation, Investigation, Methodology, Writing – review and editing; Thomas A Jowitt, Madeleine Coy, Formal analysis, Validation, Investigation, Methodology, Writing – review and editing; Jason Wong, Adam Reid, Resources, Funding acquisition, Writing – review and editing; Cédric Zeltz, Resources, Visualization, Writing – review and editing; Rajamiyer V Venkateswaran, Resources; Patrick T Caswell, Stephen High, Supervision, Funding acquisition, Investigation, Methodology, Writing – review and editing; Donald Gullberg, Resources, Supervision, Investigation, Writing

– review and editing; Karl E Kadler, Conceptualization, Supervision, Funding acquisition, Writing – original draft, Writing – review and editing

### Author ORCIDs
Joan Chang ⓘ https://orcid.org/0000-0002-7283-9759
Richa Garva ⓘ https://orcid.org/0000-0001-7752-8936
Madeleine Coy ⓘ https://orcid.org/0009-0004-7592-861X
Patrick T Caswell ⓘ https://orcid.org/0000-0002-2633-2324
Karl E Kadler ⓘ https://orcid.org/0000-0003-4977-4683

Reviewer #1 (Public review): https://doi.org/10.7554/eLife.95842.3.sa1
Reviewer #3 (Public review): https://doi.org/10.7554/eLife.95842.3.sa2
Author response https://doi.org/10.7554/eLife.95842.3.sa3

---

## Additional files

### Supplementary files
Supplementary file 1. Excel spreadsheet of proteins identified in total lysates, report from Proteome Discoverer.

Supplementary file 2. Excel spreadsheet of proteins identified in biotin-enriched samples, exported from Proteome Discoverer.

Supplementary file 3. Excel spreadsheet of proteins quantified through spectral counting between control, VPSko and VPSoe biotin-enriched samples, report from Proteome Discoverer. Presented in quantitative value as normalized to total spectra. Gene names of proteins detected shown here.

MDAR checklist

### Data availability
Mass spectrometry data are available via ProteomeXchange with identifier PXD034394.

The following dataset was generated:

| Author(s) | Year | Dataset title | Dataset URL | Database and Identifier |
| --- | --- | --- | --- | --- |
| Chang J | 2024 | Surface biotin-labelling mass spec | https://www.ebi.ac.uk/pride/archive/projects/PXD034394 | PRIDE, PXD034394 |

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

# Appendix 1

## Appendix 1—key resources table

| Reagent type (species) or resource | Designation | Source or reference | Identifiers | Additional information |
|---|---|---|---|---|
| Antibody | Rabbit polyclonal antibody (pAb) to collagen-I | Aviva Systems Biology | Cat# OARA02579, RRID:AB10873334 | 1:500 (WB) |
| Antibody | Rabbit polyclonal antibody (pAb) to collagen-I | Kerafast | Cat# ENH018 | 1:400 (IF) |
| Antibody | Mouse monoclonal antibody (mAb) to mouse vinculin | Sigma-Aldrich | Cat# V9131, RRID:AB_477629 | 1:1000 (WB) |
| Antibody | Rabbit polyclonal antibody (pAb) to mouse integrin α11 subunit | This paper | Generated from Donald Gullberg's lab | 1:1000 (WB) |
| Antibody | Mouse monoclonal antibody (mAb) to mouse VPS33B | Proteintech | Cat# 12195-1-AP, RRID:AB_2215198 | 1:500 (WB) |
| Antibody | Rabbit polyclonal antibody (pAb) to mouse VIPAS | Proteintech | Cat# 20771-1-AP, RRID:AB_10695764 | 1:200 (IF) |
| Antibody | Mouse monoclonal antibody (mAb) to FN1 | Sigma-Aldrich | Cat# F6140, RRID:AB_476981 | 1:400 (IF) |
| Antibody | Mouse monoclonal antibody (mAb) to FN1 | Abcam | Cat# ab6328, RRID:AB_305428 | (Include dilution) |
| Antibody | Rabbit polyclonal antibody (pAb) to human VPS33B | Atlas antibodies | Cat# HPA040415, RRID:AB_10795419 | 1:200 (IHC) |
| Antibody | Mouse monoclonal antibody (mAb) integrin α11 210F4 | DOI: 10.1002/cjp2.148 | Generated from Donald Gullberg's lab | 1:200 (IHC) |
| Antibody | Rabbit polyclonal antibody (pAb) to Collagen-I | Rockland Immunochemicals | 600-401-103.0.5, RRID:AB_217595 | 1:200 (IHC) |
| Antibody | IRDye 680RD Goat anti-Mouse IgG (H+L) | Li-Cor | Cat# 925-68070, RRID:AB_2651128 | 1:10,000 (WB) |
| Antibody | IRDye 800RD Goat anti-Rabbit IgG (H+L) | Li-Cor | Cat# 925-32211, RRID:AB_2651127 | 1:10,000 (WB) |
| Antibody | Goat anti-Mouse IgG (H+L) Secondary Antibody, HRP | Thermo Fisher Scientific | Cat# 32430, RRID:AB_1185566 | 1:1000 (WB) |
| Antibody | Goat anti-Rabbit IgG (H+L) Secondary Antibody, HRP | Thermo Fisher Scientific | Cat# 32460, RRID:AB_1185567 | 1:1000 (WB) |
| Antibody | Goat anti-Rabbit IgG (H+L) Cross-Adsorbed Secondary Antibody, Cyanine3 | Thermo Fisher Scientific | Cat# A10520, RRID:AB_2534029 | 1:500 (WB) |
| Antibody | Goat anti-Rabbit IgG (H+L) Cross-Adsorbed Secondary Antibody, Cyanine5 | Thermo Fisher Scientific | Cat# A10523, RRID:AB_2534032 | 1:500 (WB) |
| Antibody | Goat anti-Mouse IgG (H+L) Cross-Adsorbed Secondary Antibody, Cyanine3 | Thermo Fisher Scientific | Cat# A10521, RRID:AB_2534030 | 1:500 (WB) |
| Antibody | Goat anti-Mouse IgG (H+L) Highly Cross-Adsorbed Secondary Antibody, Alexa Fluor 647 | Thermo Fisher Scientific | Cat# A-21236, RRID:AB_2535805 | 1:500 (WB) |
| Antibody | Goat anti-Mouse IgG (H+L) Highly Cross-Adsorbed Secondary Antibody, Alexa Fluor 488 | Thermo Fisher Scientific | Cat# A-11029, RRID:AB_2534088 | 1:500 (WB) |
| Antibody | Goat anti-Rabbit IgG (H+L) Highly Cross-Adsorbed Secondary Antibody, Alexa Fluor 488 | Thermo Fisher Scientific | Cat# A-11034, RRID:AB_2576217 | 1:500 (WB) |
| Recombinant DNA reagent | pLV-V5-VPS33B-GFP1-10 | This paper | N/A | pLV vector expressing VPS33b-GFP barrell. Available on request. |

*Appendix 1 Continued on next page*

*Appendix 1 Continued*

| Reagent type (species) or resource | Designation | Source or reference | Identifiers | Additional information |
|---|---|---|---|---|
| Recombinant DNA reagent | pLV-V5-VPS33B-BFP | This paper | N/A | pLV vector expressing VPS33B tagged with BFP at C terminus. Available on request. |
| Recombinant DNA reagent | pLV-V5-BFP-VPS33B | This paper | N/A | pLV vector expressing VPS33B tagged with BFP at N terminus. Available on request. |
| Recombinant DNA reagent | pCMV-SBP-GFP11-COL1A1 | This paper | N/A | pCMV vector expression COL1A1 tagged with GFP11. Available on request. |
| Recombinant DNA reagent | pcDNA3.1+/C-(K)-DYK | This paper | N/A | pcDMA3.1 vector expression for in vitro translation. Available on request. |
| Recombinant DNA reagent | pcDNA3.1-eGFP-Rab5 | This paper | N/A | pcDNA3.1 vector with Rab5 tagged with eGFP. Available on request. |
| Biological samples (human) | Human control lung tissues | Manchester University NHS Foundation Trust | NRES14/NW/0260 | Not available due to HTA restrictions |
| Biological samples (human) | Human IPF lung tissues | Manchester University NHS Foundation Trust | NRES14/NW/0260 | Not available due to HTA restrictions |
| Biological samples (human) | Human chronic skin wound tissue | Manchester University NHS Foundation Trust | NRES 18/NW/0847 | Not available due to HTA restrictions |
| Biological samples (human) | Human healthy skin wound tissue | Manchester University NHS Foundation Trust | NRES 18/NW/0847 | Not available due to HTA restrictions |
| Chemical compound, drug | Dyngo4a | Abcam | Ab120689 | |
| Chemical compound | TRIzol reagent | Invitrogen | 15596-018 | |
| Chemical compound | Rat tail collagen-I | Corning | 354259 | |
| Chemical compound | Oregon Green 488-Dextran, 70 kDa | Life Technologies | D7173 | |
| Chemical compound | Cy3 NHS Ester | Sigma | GEPA13101 | |
| Chemical compound | Cy5 NHS Ester | Sigma | GEPA15101 | |
| Recombinant protein | Recombinant *S. pyogenes* Cas9 nuclease protein | IDT | 1081059 | |
| Commercial assay or kit | Gibson Assembly Cloning kit | NEB | E5510S | |
| Commercial assay or kit | Site-Directed Mutagenesis QuikChange kit | Agilent | 200513 | |
| Commercial assay or kit | Novolink Polymer Detection Systems | Leica Biosystems | RE7270-RE | |
| Commercial assay or kit | Rabbit reticulocyte lysate, nuclease treated | Promega | L4960 | |
| Commercial assay or kit | TaqMan Reverse Transcription kit | Applied Biosystems | N8080234 | |
| Commercial assay or kit | SensiFASTSYBR No-ROX kit | Bioline | BIO-98005 | |
| Cell lines (mouse) | NIH3T3-Dendra2-ColI | This paper | N/A | Available on request |
| Cell lines (mouse) | Immortalized mouse tail tendon fibroblasts | This paper | N/A | Available on request |
| Cell lines (human) | Primary human IPF lung fibroblasts | University of Minnesota | N/A | Not available due to HTA restrictions |

*Appendix 1 Continued on next page*

*Appendix 1 Continued*

| Reagent type (species) or resource | Designation | Source or reference | Identifiers | Additional information |
|---|---|---|---|---|
| Cell lines (human) | Primary human control lung fibroblasts | University of Minnesota | N/A | Not available due to HTA restrictions |
| Cell lines (human) | HEK293T | This paper | N/A | Available on request |
| Genetic reagent (mouse) | siCol1a1, Mission esiRNA | Merck | EMU069551 | |
| Genetic reagent (mouse) | siItga11, Mission esiRNA | Merck | EMU042761 | |
| Genetic reagent (human) | siITGA11, Mission esiRNA | Merck | EHU145321 | |
| Genetic reagent (mouse) | VPS33B mouse cDNA | GenScript | Omu07060D | |
| Genetic reagent (mouse) | Sec61β modified with a C-terminal OPG2 tag | This paper | N/A | Available on request from Sarah O'Keefe/Steve High |
| Genetic reagent (mouse) | VPS33B A-606-T | This paper | N/A | Available on request from Sarah O'Keefe/Steve High |
| Genetic reagent (mouse) | VPS33B-OPG2 | This paper | N/A | Available on request from Sarah O'Keefe/Steve High |
| Genetic reagent (mouse) | VPS33B-Sec61βTMD | This paper | N/A | Available on request from Sarah O'Keefe/Steve High |
| Genetic reagent (mouse) | VPS33B-Sec61βTMD-OPG2 | This paper | N/A | Available on request from Sarah O'Keefe/Steve High |
| Software, algorithm | Fiji ImageJ | doi:10.1038/nmeth.2019 | https://imagej.net/software/fiji/ | |
| Software, algorithm | MetaCycle | doi:10.1093/bioinformatics/btw405, Version: 1.2.0 | https://github.com/gangwug/MetaCycle; *Wu, 2022* | |
| Software, algorithm | Scaffold Proteome Software | doi:10.1002/pmic.200900437 | https://www.proteomesoftware.com/products | |
| Software, algorithm | GraphPad Prism 8 | https://www.graphpad.com/scientific-software/prism/ | https://www.graphpad.com/scientific-software/prism/ | |
| Sequence-based reagent | VPS33B A-606-T | This paper | N/A | Site-directed mutagenesis of VPS33B (then cloned into pcDNA3.1+/C-(K)-DYK) Forward: ACTGCTGTTACAAACAGTACCCGCCTCATGGAAGCC Reverse: GGCTTCCATGAGGCGGGTACTGTTTGTAACAGCAGT |
| Sequence-based reagent | VPS33B-OPG2 | This paper | N/A | Site-directed mutagenesis of VPS33B (then cloned into pcDNA3.1+/C-(K)-DYK) Forward: GCCAACGGAACAGAAGGACCAAACTTCTACGTACCATTCAGCAACAAAACAGGCTAATCCGATTACAAGGATGACGAC Reverse: CTATTAGCCTGTTTTGTTGCTGAATGGTACGTAGAAGTTTGGTCCTTCTGTTCCGTTGGATTTCACCTCACTCATGGCTTC |
| Sequence-based reagent | VPS33B-Sec61βTMD | This paper | N/A | Site-directed mutagenesis of VPS33B (then cloned into pcDNA3.1+/C-(K)-DYK) Forward: GTATTGGTTATGTGTCTTCTGTTCATCGCTTCTGTATTTATGTTGCACATTTGGGGCAAGTACACTCGTTCGTAGCTGCGCCTCATCTTGGTGGTGTTCC Reverse: CTACGAACGAGTGTACTTGCCCCAAATGTGCAACATAAATACAGAAGCGATGAACAGAAGACACATAACCAATACTGACTCACTGGAAGCCTTGTCTTCC |

*Appendix 1 Continued on next page*

*Appendix 1 Continued*

| Reagent type (species) or resource | Designation | Source or reference | Identifiers | Additional information |
|---|---|---|---|---|
| Sequence-based reagent | VPS33B-Sec61βTMD-OPG2 | This paper | N/A | Site-directed mutagenesis of VPS33B (then cloned into pcDNA3.1+/C-(K)-DYK) Forward: AACGGAACAGAAGGA CCAAACTTCTACGTACCATTCA GCAACAAAACAGGCTAATAGCTGC GCCTCATCTTGGTGGTGTTCCTG Reverse: CTATTAGCCTGTTTTGTTG CTGAATGGTACGTAGAAG TTTGGTCCTTCTGTTCCGTTCGAAC GAGTGTACTTGCCCCAAATGTGCAAC |
| Sequence-based reagent | 414-564-5M | This paper | N/A | For PCRs to create transcription templates Forward: GCCAACGGAACAGA AGGACCAAACTTCTAC GTACCATTCAGCAACAAAACAG GCTAATCCGATTACAAGGATGACGAC Reverse: CTACATCATCATCATCATTGACTC ACTGGAAGCCTTGTC |
| Sequence-based reagent | 414-587-5M | This paper | N/A | For PCRs to create transcription templates Forward: GCCAACGGAACAG AAGGACCAAACTTCTACGTACC ATTCAGCAACAAAACAGGCTA ATCCGATTACAAGGATGACGAC Reverse: CTACATCATCATCATCATCAGGAA GCGCAGGGCTGATAT |
| Sequence-based reagent | 414-FL-5M | This paper | N/A | Forward: GCCAACGGAACAGAA GGACCAAACTTCTACGTACCA TTCAGCAACAAAACAGGCTAAT CCGATTACAAGGATGACGAC Reverse: CTACATCATCATCATCATGGAT TTCACCTCACTCATGGCTTC |
| Sequence-based reagent | 414-N603-FL-5M | This paper | N/A | For PCRs to create transcription templates Forward: GCCAACGGAACAG AAGGACCAAACTTCTACGTACC ATTCAGCAACAAAACAGGCTA ATCCGATTACAAGGATGACGAC Reverse: CTACATCATCATCATCATGGATTT CACCTCACTCATGGCTTC |
| Sequence-based reagent | 414-FL-OPG2-5M | This paper | N/A | Forward: GCCAACGGAACAGAAGGACC AAACTTCTACGTACCATTCAGC AACAAAACAGGCTAATCCGAT TACAAGGATGACGAC Reverse: CTACATCATCATCATCATGCCT GTTTTGTTGCTGAATG GTACGTAGAAGTTTGGTCCTT CTGTTCCGTT |
| Sequence-based reagent | 1-564-5M | This paper | N/A | For PCRs to create transcription templates Forward: CGCAAATGGGC GGTAGGCGTG Reverse: CTACATCATCATCATCATTG ACTCACTGGAAGCCTTGTC |
| Sequence-based reagent | 1-587-5M | This paper | N/A | Forward: CGCAAATGGGCG GTAGGCGTG Reverse: CTACATCATCATCATCATCA GGAAGCGCAGGGCTGATAT |
| Sequence-based reagent | 1-FL-5M | This paper | N/A | For PCRs to create transcription templates Forward: CGCAAATGGG CGGTAGGCGTG Reverse: CTACATCATCATCATC ATGGATTTCACCTCACTCATGGCTTC |
| Sequence-based reagent | 1-N603-5M | This paper | N/A | For PCRs to create transcription templates Forward: CGCAAATGGGC GGTAGGCGTG Reverse: CTACATCATCATCATCATG GATTTCACCTCACTCATGGCTTC |

*Appendix 1 Continued on next page*

*Appendix 1 Continued*

| Reagent type (species) or resource | Designation | Source or reference | Identifiers | Additional information |
|---|---|---|---|---|
| Sequence-based reagent | 1-FL-OPG2-5M | This paper | N/A | For PCRs to create transcription templates Forward: CGCAAATGGGCGGTAGGCGTG Reverse: CTACATCATCATCATCATGC CTGTTTTGTTGCTGAAT GGTACGTAGAAGTTTGGTCCTTCTGTTCCGTT |
| Sequence-based reagent | msCol1a1 | This paper | N/A | Primers for qPCRs (ms – mouse, hu – human) Forward: AGAGCATGACCGATGG ATTC Reverse: AGGCCTCGGTGGACA |
| Sequence-based reagent | msItga11 | This paper | N/A | Forward: AGATGTCGCAGACTGGCTTT Reverse: CCCTAGGTATGCTGCATGGT |
| Sequence-based reagent | msRplp0 | This paper | N/A | Primers for qPCRs (ms – mouse, hu – human) Forward: ACTGGTCTAGGACCCGAGAAG Reverse: CTCCCACCTTGTCTCCAGTC |
| Sequence-based reagent | msGapdh | This paper | N/A | Forward: CAGCCTCGTCCCGTAGACAA Reverse: CAATCTCCACTTTGCCACTGC |
| Sequence-based reagent | msVPS33B | This paper | N/A | Primers for qPCRs (ms – mouse, hu – human) Forward: GCATTCACAGACACGGCTAAG Reverse: ACACCACCAAGATGAGGCG |
| Sequence-based reagent | huCOL1A1 | This paper | N/A | Forward: GGGATTCCCTGGACCTAAAG Reverse: GGAACACCTCGCTCTCCA |
| Sequence-based reagent | huGAPDH | This paper | N/A | Primers for qPCRs (ms – mouse, hu – human) Forward: GAGTCAACGGATTTGGTCGT Reverse: GACAAGCTTCCCGTTCTCAG |
| Sequence-based reagent | huACTB | This paper | N/A | Forward: GATCATTGCTCCTCCTGAGC Reverse: AAAGCCATGCCAATCTCATC |
| Sequence-based reagent | huITGA11 | This paper | N/A | Primers for qPCRs (ms – mouse, hu – human) Forward: CACGACATCAGTGGCAATAAG Reverse: GACCCTTCCCAGGTTGAGTT |
| Sequence-based reagent | huVPS33B | This paper | N/A | Primers for qPCRs (ms – mouse, hu – human) Forward: GAGCTGCCTGACTTCTCCAT Reverse: GCTTGTCTACTTCGTGTTGCTG |

