## [Editor Report · eLife Assessment]

This study describes a novel mechanism for how collagen fibrils are formed. The authors present **compelling** evidence that collagen-I fibrillogenesis relies on a functional endocytic system for recycling collagen-I, with circadian-regulated VPS33b and integrin-α11 being critical for fibril assembly. This is an **important** study for the understanding of the pathophysiology of collagen fibrillogenesis.

---

## [Referee Report · Reviewer #1 (Public review)]

Summary:

The authors describe that the endocytic pathway is crucial for ColI fibrillogenesis. ColI is endocytosed by fibroblasts, prior to exocytosis and formation of fibrils, which can include a mixture of endogenous/nascent ColI chains and exogenous ColI. ColI uptake and fibrillogenesis are regulated by circadian rhythm as described by the authors in 2020, thanks to the dependence of this pathway on circadian-clock-regulated protein VPS33B. Cells are capable of forming fibrils with recently endocytosed ColI along when nascent chains are not available. Previously identified VPS33B is demonstrated not to have a role in endocytosis of ColI, but to play a role in fibril formation, which the authors demonstrate by showing the loss of fibril formation in VPS33B KO, and an excess of insoluble fibrils - along-side a decrease in soluble ColI secretion - in VPS33B overexpression conditions. A VPS33B binding protein VIPAS39 is also shown to be required for fibrillogenesis and to colocalise with ColI. The authors thus conclude that ColI is internalised into endosomal structures within the cell, and that ColI, VPS33B and VIPA39 are co-trafficked to the site of fibrillogenesis, where along with ITGA11, which by mass spectrometric analysis is shown to be regulated by VPS33B levels, ColI fibrils are formed. Interestingly, in involved human skin sections from idiopathic pulmonary fibrosis (IPF) patients, ITGA11 and VPS33B expression is increased compared to healthy tissue, while in patient-derived fibroblasts, uptake of fluorescently-labelled ColI is also increased. This suggests that there may be a significant contribution of endocytosis-dependent fibrillogenesis in the formation of fibrotic and chronic wound-healing diseases in humans.

Strengths:

This is an interesting paper that contributes an exciting novel understanding of the formation of fibrotic disease, which despite its high occurrence, still has no robust therapeutic options. The precise mechanisms of fibrillogenesis are also not well understood, so a study devoted to this complex and key mechanism is well appreciated. The dependence of fibrillogenesis on VPS33B and VIPA39 is convincing and robust, while the distinction between soluble ColI secretion and insoluble fibrillar ColI is interesting and informative.

Weaknesses:

There are a number of limitations to this study in its current state. Inhibition of ColI uptake is performed using Dyngo4a, which although proposed as an inhibitor of Clathrin-dependent endocytosis is known to be quite un-specific. This may not be a problem however, as the endocytic mechanism for ColI also does not seem to be well defined in the literature, in fact, the principle mechanism described in the papers referred to by the authors is that of phagocytosis. It would be interesting to explore this important part of the mechanism further, especially in relation to the intracellular destination of ColI. The circadian regulation does not appear as robust as the authors last paper, however, there could be a larger lag between endocytosis of ColI and realisation of fibrils. The authors state that the endocytic pathway is the mechanism of trafficking and that they show ColI, VPS33B and VIPA39 are co-trafficked. However, the only link that is put forward to the endosomes is rather tenuously through VPS33B/VIPA39. There is no direct demonstration of ColI localisation to endosomes (ie. immunofluorescence), and this is overstated throughout the text. Demonstrating the intracellular trafficking and localisation of ColI, and its actual relationship to VPS33B and VIPA39, followed by ITGA11, would broaden the relevance of this paper significantly to incorporate the field of protein trafficking. Finally, the "self-formation" of ColI fibrils is discussed in relation to the literature and the concentration of fluorescently-tagged ColI, however as the key message of the paper is the fibrillogenesis from exocytosed colI, I do not feel like it is demonstrated to leave no doubt. Specific inhibition of intracellular trafficking steps, or following the progressive formation of ColI fibrils over time by immunofluorescence would demonstrate without any further doubt that ColI must be endocytosed first, to form fibrils as a secondary step, rather than externally-added ColI being incorporated directly to fibrils, independent of cellular uptake.

---

## [Referee Report · Reviewer #3 (Public review)]

Summary:

Chang et al. investigated the mechanisms governing collagen fibrillogenesis, firstly demonstrating that cells within tail tendons are able to uptake exogenous collagen and use this to synthesize new collagen-1 fibrils. Using an endocytic inhibitor, the authors next showed that endocytosis was required for collagen fibrillogenesis and that this process occurs in a circadian rhythmic manner. Using knockdown and overexpression assays, it was then demonstrated that collagen fibril formation is controlled by vacuolar protein sorting 33b (VPS33b), and this VPS33b-dependent fibrillogenesis is mediated via Integrin alpha-11 (ITGA11). The authors also demonstrated increased expression of VPS33b and ITGA11 at the gene level in fibroblasts from patients with idiopathic pulmonary fibrosis (IPF), and greater expression of these proteins in both lung samples from IPF patients and in chronic skin wounds, indicating that endocytic recycling is disrupted in fibrotic diseases. Finally, the authors performed knockdown assays in patient derived IPF fibroblasts to confirm that silencing of VPS33b and ITGA11 results in a decrease in recycling of exogenous collagen-1

Strengths:

The authors have performed a comprehensive functional analysis of the regulators of endocytic recycling of collagen, providing compelling evidence that VPS33b and ITGA11 are crucial regulators of this process, and that this endocytic recycling becomes disrupted in fibrotic diseases.

---

## [Author Response]

The following is the authors’ response to the original reviews.

Overall authors’ response

We would like to thank the 3 reviewers for a thorough critique of our manuscript, and acknowledging the novelty and importance of our studies, in particular the relevance to collagenrelated pathologies such as idiopathic pulmonary fibrosis and chronic skin wound. We appreciate that there are shortcomings in these studies, as highlighted by reviewers; we have rewritten parts of our manuscript to clarify any misunderstandings, and conducted additional experiments to address concerns raised by reviewers (please see below red text within each response), which have been incorporated into our revised manuscript (modified text highlighted in yellow in revised manuscript). We believe that the revision had made our manuscript stronger in support of our original conclusions.

**Public Reviews:**

**Reviewer #1 (Public Review):**
Summary:The authors describe that the endocytic pathway is crucial for ColI fibrillogenesis. ColI is endocytosed by fibroblasts, prior to exocytosis and formation of fibrils, which can include a mixture of endogenous/nascent ColI chains and exogenous ColI. ColI uptake and fibrillogenesis are regulated by circadian rhythm as described by the authors in 2020, thanks to the dependence of this pathway on circadian-clock-regulated protein VPS33B. Cells are capable of forming fibrils with recently endocytosed ColI when nascent chains are not available. Previously identified VPS33B is demonstrated not to have a role in endocytosis of ColI, but to play a role in fibril formation, which the authors demonstrate by showing the loss of fibril formation in VPS33B KO, and an excess of insoluble fibrils - along-side a decrease in soluble ColI secretion - in VPS33B overexpression conditions. A VPS33B binding protein VIPAS39 is also shown to be required for fibrillogenesis and to colocalise with ColI. The authors thus conclude that ColI is internalised into endosomal structures within the cell, and that ColI, VPS33B, and VIPA39 are co-trafficked to the site of fibrillogenesis, where along with ITGA11, which by mass spectrometric analysis is shown to be regulated by VPS33B levels, ColI fibrils are formed. Interestingly, in involved human skin sections from idiopathic pulmonary fibrosis (IPF) patients, ITGA11 and VPS33B expression is increased compared to healthy tissue, while in patient-derived fibroblasts, uptake of fluorescently-labelled ColI is also increased. This suggests that there may be a significant contribution of endocytosis-dependent fibrillogenesis in the formation of fibrotic and chronic wound-healing diseases in humans.Strengths:This is an interesting paper that contributes an exciting novel understanding of the formation of fibrotic disease, which despite its high occurrence, still has no robust therapeutic options. The precise mechanisms of fibrillogenesis are also not well understood, so a study devoted to this complex and key mechanism is well appreciated. The dependence of fibrillogenesis on VPS33B and VIPA39 is convincing and robust, while the distinction between soluble ColI secretion and insoluble fibrillar ColI is interesting and informative.Weaknesses:There are a number of limitations to this study in its current state. Inhibition of ColI uptake is performed using Dyngo4a, which although proposed as an inhibitor of Clathrin-dependent endocytosis is known to be quite un-specific. This may not be a problem however, as the endocytic mechanism for ColI also does not seem to be well defined in the literature, in fact, the principle mechanism described in the papers referred to by the authors is that of phagocytosis.

We thank the reviewer for pointing this out. Macropinocytosis or phagocytosis could be modelled using high molecular weight dextran, and we have used fluorescently-labelled dextran to investigate potential co-localisation with exogenous collagen to investigate the involvement of these mechanisms in addition to endocytosis, and showed very little co-localisation (revised Figure S2B, lines 123-126). Further, we have performed a competition experiment where unlabelled collagen was added in excess at the same time as labelled collagen and showed that excess unlabelled collagen led to a retention of labelled collagen at the cell periphery (revised Figure S2C, lines 126-129). This is suggestive of collagen-I uptake utilises a different pathway to dextran (i.e. fluid-phase endocytosis) and is a receptor-mediated process.

It would be interesting to explore this important part of the mechanism further, especially in relation to the intracellular destination of ColI.

We agree with the reviewer that the intracellular destination of ColI is very interesting, which is what the current Chang lab is investigating, although we believe the research findings fall out of scope for the revised manuscript here. However, we have included additional immunofluorescence data to support that collagen is indeed taken up into endosomal compartments using GFP-tagged Rab5 constructs (revised Figure 1D, Figure S6A).

The circadian regulation does not appear as robust as the authors' last paper, however, there could be a larger lag between endocytosis of ColI and realisation of fibrils.The authors state that the endocytic pathway is the mechanism of trafficking and that they show ColI, VPS33B, and VIPA39 are co-trafficked. However, the only link that is put forward to the endosomes is rather tenuously through VPS33B/VIPA39.

We would like to clarify that we meant the post-Golgi compartment. We did not mean VPS33b/VIPAS39 as an endosome marker; however as we see collagen entering the cell in intracellular compartments, which is then recycled, we take that as convention, the endosome would be involved. This is further supported that we see some colocalisation with the classic Rab5 endosome marker.

There is no direct demonstration of ColI localisation to endosomes (ie. immunofluorescence), and this is overstated throughout the text.

We appreciate the comment and have modified overstatements in the revised manuscript as appropriate. As stated above, we have included additional immunofluorescence data to support that collagen is indeed taken up into endosomal compartments.

Demonstrating the intracellular trafficking and localisation of ColI, and its actual relationship to VPS33B and VIPA39, followed by ITGA11, would broaden the relevance of this paper significantly to incorporate the field of protein trafficking. Finally, the "self-formation" of ColI fibrils is discussed in relation to the literature and the concentration of fluorescently-tagged ColI, however as the key message of the paper is the fibrillogenesis from exocytosed colI, I do not feel like it is demonstrated to leave no doubt. Specific inhibition of intracellular trafficking steps, or following the progressive formation of ColI fibrils over time by immunofluorescence would demonstrate without any further doubt that ColI must be endocytosed first, to form fibrils as a secondary step, rather than externally-added ColI being incorporated directly to fibrils, independent of cellular uptake.

We appreciate the concern raised here. This is precisely why we trypsinised and replated cells as part of the workflow, so we can make sure that there is no residual exogenous collagen which is not endocytosed being incorporated onto pre-existing fibrils. We have new data using flow imaging, which showed that cells that don’t endocytose exogenous collagen has accumulation of said collagen at the periphery of the cells, which is greatly reduced after trypsinisation. This new data is in a more detailed methodology-based study which is under preparation, which will allow future studies to further dissect the collagen intracellular trafficking process, and thus is not included in the revised manuscript.

**Reviewer #2 (Public Review):**
Summary:In this manuscript, the authors describe a mechanism, by which fluorescently-labelled Collagen typeI is taken up by cells via endocytosis and then incorporated into newly synthesized fibers via an ITGA11 and VPS33B-dependent mechanism. The authors claim the existence of this collagen recycling mechanism and link it to fibrotic diseases such as IPF and chronic wounds.Strengths:he manuscript is well-written, and experimentally contains a broad variation of assays to support their conclusions. Also, the authors added data of IPF patient-derived fibroblasts, patient-derived lung samples, and patient-derived samples of chronic wounds that highlight a potential in vivo disease correlation of their findings.The authors were also analyzing the membrane topology of VPS33B and could unravel a likely 'hairpin' like conformation in the ER membrane.Weaknesses:Experimental evidence is missing that supports the non-degradative endocytosis of the labeled collagen.

We thank the reviewer for raising this. We would like to clarify that we do not think that all endocytosed collagen-I is recycled, but rather sorted in the endosome which determines the fate of endocytosed collagen. Interestingly, results from Kadler’s group has shown that blocking lysosome function (through chloroqine and bafilomycin) significantly reduced endogenous collagen fibril formation (https://www.biorxiv.org/content/10.1101/2024.05.09.593302v1), suggesting a nondegradative role for lysosome in fibrillogenesis.

The authors show and mention in the text that the endocytosis inhibitor Dyngo4a shows an effect on collagen secretion. It is not clear to me how specific this readout is if the inhibitor affects more than endocytosis. This issue was unfortunately not further discussed.

We thank the reviewer for this comment and have included in discussion the specificity of Dyngo4a (revised manuscript lines 383392). The ponceau stain suggests that Dyngo4a treatment did not affect global secretion and thus the effects are specific to collagen-I (Fig 2B).

The authors use commercial rat tail collagen, it is unclear to me which state the collagen is in when it's endocytosed. Is it fully assembled as collagen fiber or are those single heterotrimers or homotrimers?

We apologise for the confusion and will clarify in our revision. These would be single helical trimers from acid-extracted rat tail collagen. We have performed additional light scattering and CD spectra to confirm the molecular weight and helicity, and confirm that adding fluorescent tags did not alter the readout. We have included this in the revised manuscript (revised Figure S1A-C, manuscript lines 82-86).

The Cy-labeled collagen is clearly incorporated into new fibers, but I'm not sure whether the collagen is needed to be endocytosed to be incorporated into the fibers or if that is happening in the extracellular space mediated by the cells.

We appreciate the concern raised here, which is also raised by reviewer 1. As answered above, this is why we trypsinised and replated cells as part of the workflow, so we can make sure that there is no residual exogenous collagen being incorporated onto pre-existing fibrils. We also have new data using flow imaging, which shows that cells that don’t endocytose exogenous collagen has accumulation of said collagen at the periphery of the cells, which is greatly reduced after trypsinisation. This new data is in a methodology-based manuscript which is under preparation, thus will not be included in the revised manuscript.

In general for the collagen blots, due to the lack of molecular weight markers, what chain/form of collagen type I are you showing here?

Apologies for the lack of molecular weight markers, it was an oversight by the authors and have been included in the revised figures.

Besides the VPS33B siRNA transfected cells the authors also use CRISPR/Cas9-generated KO. The KO cells do not seem to be a clean system, as there is still a lot of mRNA produced. Were the clones sequenced to verify the KO on a genomic level?

Yes, the clones were verified and used in our previous paper on circadian control of collagen homeostasis. There are instances where despite knockout at the protein level, mRNA is still persistent; however these transcripts are likely then directed to degradation through nonsense-mediated mRNA decay. To fully understand this mechanism is beyond the scope of this paper.

For the siRNA transfection, a control blot for efficiency would be great to estimate the effect size. To me it is not clear where the endocytosed collagen and VPS33B eventually meet in the cells and whether they interact. Or is ITGA11 required to mediate this process, in case VPS33B is not reaching the lumen?

This is an interesting question. We have conducted experiments with Col1-GFP11 containing conditioned media incubated with VPS33b-barrell in the revised paper, which showed that they interact within the cell and not at the cell periphery (revised Figure 6G, lines 293-296), again highlighting that VPS33b is not involved in the endocytosis step but interacts with endocytosed collagen-I intracellularly. We have attempted colocliasation studies using the split GFP approach with VPS33B and ITGA11 to investigate where they interact, but as the ITGA11 construct we used did not localise to the cell surface as expected, we are not confident that this system is appropriate for investigating how/if VPS33B interacts with ITGA11, and there are simply no good antibody for VPS33B for staining.

The authors show an upregulation of ITGA11 and VPS33B in IPF patients-derived fibroblasts, which can be correlated to an increased level of ColI uptake, however, it is not clear whether this increased uptake in those cells is due to the elevated levels of VPS33B and/or ITGA11.

We would like to clarify here that we do not think collagen-I uptake is due to VPS33B and/or ITGA11, as siITGA11 and VPS33B in fibroblasts showed no consistent changes in uptake as determined by flow cytometry, which was included in the original manuscript (now revised Figure 6H, 7I). VPS33B and ITGA11 are involved in the ‘outward’ arm of recycled collagen-I, i.e. directing to fibrillogenesis route. We agree that the inclusion of additional functional studies using IPF patient-derived patient fibroblasts would add to the manuscript, and have performed siRNA against VPS33B and ITGA11 on IPF fibroblasts, and demonstrated a late of endocytic recycling events (revised Figure 8D, S6B, lines 351-353).

**Reviewer #3 (Public Review):**
Summary:Chang et al. investigated the mechanisms governing collagen fibrillogenesis, firstly demonstrating that cells within tail tendons are able to uptake exogenous collagen and use this to synthesize new collagen-1 fibrils. Using an endocytic inhibitor, the authors next showed that endocytosis was required for collagen fibrillogenesis and that this process occurs in a circadian rhythmic manner. Using knockdown and overexpression assays, it was then demonstrated that collagen fibril formation is controlled by vacuolar protein sorting 33b (VPS33b), and this VPS33b-dependent fibrillogenesis is mediated via Integrin alpha-11 (ITGA11). Finally, the authors demonstrated increased expression of VPS33b and ITGA11 at the gene level in fibroblasts from patients with idiopathic pulmonary fibrosis (IPF), and greater expression of these proteins in both lung samples from IPF patients and in chronic skin wounds, indicating that endocytic recycling is disrupted in fibrotic diseases.Strengths:The authors have performed a comprehensive functional analysis of the regulators of endocytic recycling of collagen, providing compelling evidence that VPS33b and ITGA11 are crucial regulators of this process.Weaknesses:Throughout the study, several different cell types have been used (immortalised tail tendon fibroblasts, NIHT3T cells, and HEK293T cells). In general, it is not clear which cells have been used for a particular experiment, and the rationale for using these different cell types is not explained. In addition, some experimental details are missing from the methods.

We thank the reviewer for pointing out the lack of clarity, and have filled in missing information in the methods. HEK293T cells were used for virus production for the VPSoe system, and we have clarified the cell types used in figure legends (predominantly iTTF). We have also provided justification when NIH3T3 cells were used (revised lines 290-291).

There is also a lack of functional studies in patient-derived IPF fibroblasts which means the link between endocytic recycling of collagen and the role of VPS33b and ITGA11 cannot be fully established.

We thank the reviewer for this comment, which was also raised by reviewer 2 above. We agree that the inclusion of additional functional studies using IPF patient-derived patient fibroblasts would add to the manuscript and have performed siRNA against VPS33B and ITGA11 on IPF fibroblasts, and demonstrated a late of endocytic recycling events (revised Figure 8D, S6B, lines 351-353).

**Recommendations for the authors:**

**Reviewer #1 (Recommendations For The Authors):**
The authors inhibit Clathrin-dependent endocytosis with dyngo4a. It is well known that this inhibitor is not highly specific for this pathway. It is also not explained why the authors only inhibit the Clathrin uptake pathway, and not pinocytosis or Clathrin-independent endocytosis too. The authors refer to papers that describe pinocytosis for collagen endocytosis.

We thank the reviewer for raising this question. Based on the fact that inhibition of clathrin-dependent pathway does not completely abrogate endocytosis of collagen-I, we anticipate that other pathways are involved in mediating collagen-I uptake, although additional data suggested this is unlikely through fluid-phase endocytosis, and is receptor mediated (revised Figure S2B, C).

Where does the ColI go in the cell? Depending on the uptake pathway, it is likely to pass through endocytic carriers to endosomes, where it may be recycled to the PM or degraded. From the start, the authors describe the ColI as being in vesicular structures, however, the imaging data that this is based on is not co-labelled with anything to determine the potential structure/localisation. This is not done at any point in the paper, until IF is shown of ColI with VIPA39, however without the relevant controls, this IF is unconvincing, as the general pattern of ColI and VIPA39 as an endosomal marker are not classically recognisable. Additionally, VPS33B is described as a late endosome/lysosome marker, which would have different connotations on ColI trafficking or destination than other types of endosomes.

We thank the reviewer for pointing out the weaknesses in our original IF. We have included new confocal images showing labelled collagen co-localisation with GFP-tagged Rab5 through transient transfection, which is a more traditional endosome marker (revised Figure 1D, Figure S6A).

We are currently characterising the compartments to where ColI is trafficked to, which is being prepared as part of a methodology-based manuscript. We believe that this characterisation would be too detailed to be included in a revised version of this manuscript. The Kadler lab also have data suggesting that the lysosome is involved in collagen fibrillogenesis instead of its canonical degradation function, which is in another submitted manuscript (https://www.researchsquare.com/article/rs-1336021/v1). It was not included in this manuscript due to our focus (i.e. endocytic-recycling).

In Figure 5H, the pattern of Cy5-ColI staining looks like it could even be ER/Golgi in the VPSKO zoom panel, but in the absence of co-labelling, we cannot conclude anything. In order for the authors to conclude that ColI is within the endosomes, co-labelled If should be performed to demonstrate ColIendosomal colocalization. Likewise for the role of VPS33B in ColI fibrillogenesis: dependence of the process is demonstrated, but the relationship is not defined. This could be clarified using IF. This would also support the authors' statements of co-trafficking between ColI, VPS33B, and VIPA39, which as the paper stands, is not demonstrated.

We would like to clarify that our hypothesis is that the endosome controls how collagen is being deposited outside the cell, i.e. whether it’s protomeric secretion or fibrillogenesis, and that the decision of whether an endocytosed collagen is recycled or degraded lies in this compartment. The reviewer is correct that it may not be just the endosome that endocytosed collagen-I ends up in, as we have new data suggesting involvement of other intracellular compartment, although the detailed mechanism is beyond the scope of this manuscript. Nonetheless, we have included new data showing co-localisation of endocytosed collagen with Rab5 in this revised manuscript (revised Figure 1D, Figure S6A).

The basis of this paper is that endocytosis of ColI must occur before re-exocytosis as fibrillar ColI. The authors show this through pulse-chase experiments, with a trypsinisation step to remove any externally bound ColI. The authors also show nice time progression by flow cytometry, but it would truly demonstrate this point if they showed 0 timepoint, or low timepoint of IF to show progressive lengthening of ColI fibrils. This is used early on in Figure 1D, although the presentation here is not very clear. This is especially important as the authors address the self-seeding capabilities of Collagen in cell-free conditions in Figure 1F.

We would like to thank the reviewer for this suggestion. From previous endogenously tagged collagen data, we know that the appearance of collagen fibrils is rather rapid, thus it may not be a gradual lengthening as expected, but rather a depletion of endocytosed collagen in the initial seeding/growth step (please see https://www.researchsquare.com/article/rs-1336021/v1). We have included an image of replated fibroblasts after 18 hours showing no appearance of extracellular collagen, endogenous or otherwise (revised Figures S2A, line 110).

Finally, although the involvement of ITGA11 is interesting, it is not well described, and its role is not well demonstrated. This could likely be clarified by an additional introduction to ITGA11 and its role in collagen exocytosis/fibrillogenesis.

We would like to thank the reviewer for pointing this out and have included additional sentences to specifically introduce ITGA11 and its role in fibrillogenesis (see lines 320, 321; 446-450).

Specific points:Line 73: You haven't compared reuse vs production, so you can't say that reuse is central rather than production. They may be both as important or production still may be the most crucial, maybe it depends on cell/collagen type. Using the ColI KD or CHX to block nascent synthesis, you could directly compare the impact of both.

We would like to clarify that we are not referring to reuse/recycling here. We meant that production of collagen (i.e. single hetero/homotrimer molecules within the cell) is not as crucial as the utilisation (i.e. are these being secreted as protomers, or assembled into fibrils) of these building blocks by the cells, which was supported by our finding that production (as suggested by mRNA levels) of IPF fibroblasts are similar to that in control fibroblasts (now revised Figure 8A). We have conducted ColI siRNA to block nascent synthesis in the original manuscript and showed that fibroblasts can efficiently make new fibrils by recycling exogenous collagen (Figure 3B, C), although we appreciate that siRNA may not completely inhibit endogenous production. Thus, we have also included new data using collagen-I knockout cells to support our hypothesis that without endogenous production, fibroblasts can still effectively make collagen fibrils if they can reuse what is available in the extracellular space (revised Figure 4, Figure S3C, D; lines 178-199).

Lines 83-87: The rationale for this experiment is not clear. Cy3-ColI is added, taken up into cells, and incorporated into fibrils coming from cells. 5FAM-ColI is added at a later stage, then at 2 days (when incorporation is demonstrated in Fig 1B), it is also incorporated into cells as expected. Why does this comment on ColI not being degraded any more than Cy3-ColI alone?

We believe that the pulse chase experiment using the differently tagged collagen demonstrated a dimension of dynamics that is not demonstrated with Cy3-ColI alone. In this case, Cy3-ColI was initially added, and removed after 3 days; 5FAM-ColI is then added and incubated for 2 more days. Thus after 5 days since the initial pulse, the Cy3-ColI persisted and was not degraded. We would like to apologise for causing this confusion, and have clarified in the revised manuscript (lines 542-549; Figure S1D figure legend).

Figure 1A: I would like to see a negative control: either dark colI or no Cy3-Col, or timescale. Is B quantified from these images?

We thank the reviewer for this comment. We have added the nocollagen control image in our revision (revised Figure S1D). 1B is not quantified from the ex vivo tendon experiments, but rather the in vitro cell culture experiments (i.e. those from 1D-1F, although they are all from independent experiments).

Figure 1B: in iTTF cells (immortalised tendon cells) Corrected to max: What does that mean?

As there are variations between individual experiments (e.g. changes in the amount of collagen added due to pipetting) we have normalised to the maximum value obtained in each individual experiments so that we can display all biological repeats within the same graph.

Figure 1C: You can't say ColI is in vesicular structures from this, they are spots, yes, but that could also be in Golgi/ER (unlikely to be cytosolic but not impossible).

We appreciate this comment and have change the wording accordingly and call them intracellular/punctate structures.

Figure 1D: Not the best presentation: The cell mask has structures: what are these? It's not clear if this is a single cell, would be better with a defined marker (endocytic marker, lysosome etc). Instead of a low-resolution 3D view, it would be clearer with normal confocal XY and zooms of "vesicular structures" using appropriate markers as 3D reconstructions I think it could be removed.

This is a single cell and the cell mask is staining plasma membrane. We didn’t use defined marker as we wanted to visualise the whole intracellular cell compartment. We appreciate that further proof is needed to verify the location of the endocytosed collagen, and have included additional confocal imaging data to support the localisation of collagen into Rab5 positive intracellular compartments (revised Figure 1D, Figure S6B).

Figure 1 E/F: Cy3 is only visible in extracellular structure, not also intracellular. Why? Would be useful to see the time points of incorporation at the end of the pulse, then at an early point into the chase, to demonstrate 1 Cy3-ColI uptake into cells and progressive incorporation rather than potential direct binding of ColI-Cy3 to ECM, or other non-specific factors. Showing the image at 0t would demonstrate an absence of external labelled colI and therefore its appearance later could be presumed that it had been internalised before.

As the cells were trypsinized and replated after one hour labelled collagen feeding to ensure we are only tracking endocytosed collagen, t=0 in this case would be cells that are unattached. We have included t=18hr images post replate instead to show baseline level of collagen (revised Figures S2A, line 110).

Figure S1A: yellow box: doesn't show only Cy3-ColI, there is red and yellow in the central cell, and large yellow blobs in the cell above. These images do not support this claim, including the Fiber Zoom box. They should also be shown in single channels to demonstrate the authors' points better.

Apologies for the confusion – this is to show that newly added FAM5 Collagen is also co-localising with previously endocytosed Cy3-ColI, i.e. the Cy3-ColI is persisting rather than being degraded.

Line 92: endocytosed into distinct structures: These images are very vague, but I don't think you can call them distinct structures, all you can say from this is that they are spots.

We have changed the wording to ‘distinct puncta’.

It is not clear why the authors use Cy3, Cy5, and 5FAM labelled colI. A brief explanation would be useful.

Apologies for the confusion, we initially included our justification (to show that the fluorescence labels do not change the way collagen is internalised) but removed it in the final manuscript due to length. We have added the justification (revised line 101-102).

Figure 1F: It would be useful to see a quantification of the Cy3 channel here: I agree with the conclusions, and find the 0.5 ug/ml condition more convincing than 0.1 actually, although there is some feint Cy3 in cell-free samples there seems to be quite a big increase in the presence of cells, and this would look more convincing if quantified.

We thank the reviewer for this suggestion and have included quantification in the revised manuscript (revised Figure 1G-I).

Figure 2B: Dyng is not an abbreviation of Dyng. Standardise Dyng/Dyngo/Dyngo4a. WB is soluble colI and represents little (if any) insoluble col. IF is more or less the other way round. How do they compare this?

Thank you for pointing out the inconsistencies, we have corrected this in the revised manuscript. We took the conditioned media from the same experiment where cells are fixed for IF and carried out Western blot analyses. The IF showed some collagen still present, albeit significantly reduced. This is in agreement with the western blot results (i.e. Dyng4a inhibits both soluble and insoluble forms of collagen deposition).

Figure 2C: not an image series. Quant: no cells/independent exps and STATS?

Apologies for the missing experimental details in figure legends, it should say ‘representative of N=3 experiments’. We are not sure what the reviewer meant by Figure 2C not being an image series, as we meant it to be an image series of the individual fluorescence channels. We have changed this terminology to avoid confusion, and have included statistical analyses in the methods section. The statistical analyses of the fibril quantification is next to the fluorescence images.

Figures 2D/E: The authors show that internalised ColI peaks at 20h and decreases to 60h, Fibers peak at 40h. How is this measured? ECM removed? Why would there be less in the cells, degradation? Whats the synchronisation?

We apologise for omitting the synchronisation method in methods section, and have included in our revised manuscript (revised lines 542-544). This is through dexamethasone addition (and removal after 1hr incubation) as standard. The internalised Col-I is measured using Cy3ColI so the cells would have both nascent and external collagen. Total intracellular collagen at the different time points would likely be higher than represented as a result, but here we are demonstrating that internalisation is a rhythmic event using the external labelled collagen. Fibers are measured using standard IF and then fibril counting.

Please note that we are only overlaying the two graphs to form our hypothesis that endocytosis may be used for accumulation of collagen protomers that then allows for efficient fibrillogenesis. They are not directly comparable as the quantification are of different things (internalised Cy3-ColI, total collagen fibrils). We have clarified this in our discussion (revised lines 399-401).

Discussion: Where does the ColI go? Solubilised? Degraded? Taken up by other cells?The inverse correlation is not very tight. In fact, at 38h where fiber count peaks, Cy3-ColI also peaks (esp in normalised data, Figure S2D).

We thank the reviewer for this comment and have reworded our main text to reflect this, and included additional discussion in our revised manuscript (revised lines 401-404).

Line 123: What is the turnover rate of Fibrils? Don't know for how long the transcription has been done, or when this would affect the fibril number. You have the quant for Fn1, where is the quant for ColI?

We have included the quantification of collagen-I in original Figure 2A. We appreciate that it might cause confusion in Figure 2C (as we co-stained ColI and Fn1 in the same experiment) we have removed the collagen-I panel from the revised Figure 2C. We know from previous results that the number of fibrils fluctuate over 24hour period, although the turnover of one specific fibril is unlikely going to be 24 hours (https://www.biorxiv.org/content/10.1101/331496v2)

Line 124: no accumulation of col in extracellular space, but you don't know how much endogenous colI (or other endogenous ECM proteins) they're taking up as it isn't measured here. If the author wants to comment on this, should use either exogenous col to monitor take up and resection or block transcription/translation to show fibril formation endo/exocytosis independent of endogenous synthesis.

This experiment has been done in the original manuscript – siCol1a1 experiment was done with two rounds of siRNA, first round is normal transfection followed by reverse transfection onto fresh coverslips (this will ensure no prior ECM is being deposited, see Figure 3). However we appreciate that there may still be low levels of endogenous collagen-I, and thus have included new data using collagen-I knock-out fibroblasts to strengthen our findings (revised Figure 4).

Line 142: Why is fibronectin synthesis also decreased in Col KD? This is clear in the image but no explanation/reference is given.

Due to the dynamic and complex nature of ECM, it is unsurprising if there is a knockon effect when knocking down one matrix protein. However, we have quantified the amount of fibronectin fibril deposited by scr and siCol1a1 fibroblasts, and showed that there was in fact no significant change between the two treatments (revised Figure 3A).

Figure 3A: Need labels for which colour/protein is shown. Needs quantifying, especially as the Fn1 decrease is not so obvious here, it is consistent between Figure 3A and 2C?

We have provided quantification in the revision (revised Figure 3A). Figure 3A and 2C are two separate experiments (one is Dyngo treatment and one is siCol1a1), and neither showed significant changes in fibronectin fibril areas.

Figure 3B: Line 151: the text states that "The observation of fibrillar Cy3 signals in siCol1a1 cells showed that the cells can repurpose collagen into fibrils without the requirement for intrinsic collagen-I production (red arrow Figure 3B), however, there is clearly endogenous colI here too (along the fiber and also strongly at each end). Does the ColI antibody recognise the exogenous ColI?

In our hands the ColI antibody does not recognise exogenous ColI, as the cell-free Cy3-ColI images were also stained with ColI antibody to ensure the two experimental conditions were treated exactly the same.

This conclusion could only be made in the true absence of collagen: either in knock-out cells, or where collagen production/trafficking has been blocked (ie knockout of ColI chaperone or ERES block), or in a cell type that produces collagens but not ColI. Alternatively, if there are any fibrils seen that are completely negative, they should be shown in the figure and quantified (number of Cy3-ColI+-ColI+ vs Cy3-ColI+-ColI-).

We thank the reviewer for this suggestion. We have included new data from collagen knock-out fibroblasts in this revision (revised Figure 4).

Figure S4A: the quality of this blot isn't very high, the result is not very clear and the high intensity (unspecific?) band below confounds the interpretation. In the author's previous paper (NCB 2020) the blots for VPS33B were much clearer, as is Fig S4D. It would be nice to include a clearer blot, maybe from the other repeats.

This is the only blot that we used to select which knockout clones to use for our previous paper, which is why the quality is not as high. Knockout clones were all verified with additional western blots, and we do not think that endogenous VPS33b is expressed at high levels (also verified by MS analyses). Fig S4D is overexpression of VPS33b, which is much easier to detect.

Figure S4D: This blot is much clearer, it would be useful to include a high gain to show the VPS33B band in CT to be able to understand the true increase.

From the qPCR data one can see that the increase at mRNA is 20+ fold increase; we’ve always had problems trying to detect endogenous VPS33b using western blot or mass spectrometry analysis.

Figure 4A: The fibrils here in the CT are not obvious, and the difference between CT and KOs is not appreciable. Would this be clearer shown at a lower magnification, with zooms where needed? Or immunogold labelling/CLEM to label the ColI?

It is not trivial to carry out immunogold labelling/CLEM. These are cell-derived matrices in culture and thus lower magnification may not show as many collagen fibrils as one would expect. We are not confident that lower magnification will provide more information as the characteristic D-banded collagen pattern will be lost.

Line 167/Figure 4B: It looks like there is more internal ColI in KO, but the images are not good enough to tell. This could be better shown by flow cytometry.

We have previously seen that VPSKO leads to accumulation of collagen-I in intracellular punctas (NCB2020) which is also seen here. Flow cytometry data for internalisation of external collagen is already included in original Figure 5G (revised Figure 6H).

Again you mention intercellular vesicles, but based on these images, it is not possible to conclude this. These large spots could be aggregation elsewhere in the cell. Specific localisation should be shown by co-labelled IF/confocal, or it could be nicely shown by EM + fluorescent element (CLEM / Immunogold), or these statements removed from the text.

We appreciate that the term ‘vesicles’ is very defined in the trafficking field, and have changed it to ‘intracellular compartments’.

Line 173-174 / Figure 4E: Why do you think the matrix mass is not increased in VPSoe by the approach shown in E when there is seemingly a huge increase by IF? E must also measure other ECM matrix proteins, which do you expect to be secreted by these cells? Could this confound the data if they too are affected by VPSoe?

IF is showing specifically collagen-I. Hydroxyproline detects multiple collagens, and shows a trend of increase (although not significant due to one outlier). Matrix mass is a very generic measurement of total ECM deposited based on decellularized ECM weight. The reviewer is correct that VPSoe may also affect other ECM deposition, however here we are focussing specifically with its effect on collagen-I. How VPSoe changes other types of ECM deposition would be something that could be addressed in future studies and is not within scope of this manuscript.

Are the results in E paired?

Individual values between control and VPSoe in each separate experiments are paired.

Figure 4F: Is quantification from IF shown in D? Specify which kind of microscopy it is based on.

Quantification is based on fibril counting using standard fluorescence microscopy, as used in our previous paper. D is independent of F, as F is specifically looking at synchronised circadian effects, and D (and elsewhere) we are looking at global collagen deposition effects, irrespective of what time of day the cells are in.

Figure S5F: What do the yellow/red spots in the blots represent?

We apologise for the initial unclear description of what the yellow/magenta circles depict in relation to the phosphoimages of the radiolabelled cell free translation products displayed in Supplementary Figure 5, panels F, G and I. These circles indicate non-glycosylated (yellow) and N-glycosylated (magenta) species respectively, as is now clearly descried in the revised manuscript.

Figure 5 title: You can't conclude this from these images, need confocal and PM or cytosolic marker.

We have changed the title to ‘VPS33B co-trafficks with collagen-I”. There is no good commercial VPS33b antibody for immunofluorescence staining, which is why we used the split GFP approach in this paper, and the images were acquired using confocal imaging (Olympus SpinSR system).

Figure 5E: The authors describe that ColI is in endosomes throughout most of the paper, and this is based on the involvement of VPS33B in the colI pathway. VPS33B is thought to be at the late endosome/lysosome. However, these images do not look like classic endosomes or lysosomes, or other normal organelle IF phenotypes. The fluorescent intensity looks saturated, and it is difficult to conclude anything from these images. It is unclear where in the cell the largest blob in the zoom would be localised and in which cell. I would suggest that this image is replaced and proper controls included (IgG controls and single channels) as well as using different markers for other potential intracellular structures.

We appreciate the reviewers comment with regards to the classification of VPS33b localisation in the endosome compartment. We did not mean to use VPS33b as an endosome marker, as the focus of our studies are the function of VPS33b in directing endogenous or exogenous collagen to fibrillogenesis. With live imaging we could see endocytosed collagen moving in intracellular compartments, and have conducted additional staining to show co-localisation with Rab5 (revised Figure 1), which we take to indicate, through convention, that it is occupying an endosome compartment. We have included single channel images in the revised manuscript (revised Figure 6E).

Line 255/ Figure 5G: no consistent change in uptake. Why are the results so varied in the KO and oe, here and in Fig 4C/E? N=4, what does that mean? 4 cells? 4 independent exps?

In all cases, “N” represents independent biological experiments in this manuscript. Thus “N=4” in this case is 4 independent biological experiments, with at least 10,000 cells analysed per experiment.

We don’t know why there is a variation in response, however that is also why we concluded that it is unlikely that VPS33B is directly involved with collagen uptake. We have changed 5G (now revised Figure 5H) to a paired line graph for better representation.

Figure 5H shows the uptake of Cy5ColI. At this resolution, VP2ko looks like the col is ER, in one of the cells in the zoom, it looks like it is at Golgi. I think that the uptake route of ColI needs to be better defined, as there is no way to tell here where the colI goes. ColI being recycled/degraded would be most likely. But this figure looks like that might not be the case. It is also not clear where the zooms come from, they should be indicated with dashed boxes in the lower mag image

We thank the reviewer for this comment, and agree that we need to define the uptake route of ColI. This is currently being assembled as a methodology manuscript, and how ColI is being recycled/degraded is one major research area of the Chang lab.

We have added dashed boxes in the lower mag images to indicate where the zooms derived from, and we would also like to thank the reviewer for pointing this out as we realised we have accidentally cropped the image to a slightly different area for the VPSko image, and have now corrected this.

Line 257: Based on this data, it could be trafficking through the cell as well as into the extracellular space.

We think that VPS33B is involved in trafficking collagen through the cell to plasma membrane but not secreted, as based on our split-GFP experiment we never observed extracellular GFP signal, which suggests VPS33b is not deposited extracellularly.

Line 259: "highlighting the role in recycling col to fibril formation sites" is an overstatement based on the data shown here, there is no data on colI trafficking or its regulation

We respectfully disagree that we have not shown data on col-I trafficking or regulation by VPS33b – split GFP highlighted cotrafficking to the plasma membrane, and we have shown a clear relationship between VPS33b and collagen-I fibril formation, with minimal changes to collagen-I mRNA levels. We acknowledge that we have not shown specifically the location of VPS33b at fibrillogenic sites and have modified this statement in revised manuscript (revised line 302).

Line 262: "Having identified VPS33B as specifically driving collagen-I fibril formation" is also an overstatement.

We refer here the data that VPS33b is not controlling collagen-I secretion (as demonstrated by the CM westerns) and specifically fibrillogenesis. We have clarified this in the revised text (revised line 304).

Line 286: It would be useful to have a brief intro to PLOD3.

We have included a brief intro to PLOD3 in the introduction, as well as the results highlighted by the reviewer, in our revised manuscript (revised line 54-58).

Line 289/290: There could be other explanations for disruption to exo-endocytosis when disrupting col trafficking. Is VPS33B controlling exocytosis in general? Why should it be specific to col? Likewise with siITGA11 KD? Hypothesis for ITGA11 and fibrillogenesis?

The relationship between ITGA11 and collagen fibrillogenesis is currently in a manuscript by Donald Gullberg and Cedric Zeltz, under revision at Matrix Biology (see reference 63 in revised manuscript). We do not think that VPS33b is controlling exocytosis in general, which is supported by the minimal change in ponceau stain of the western blots in the manuscript. Previously it has been shown that VPS33B co-trafficks with PLOD3, a collagen-I modifier.

Figure 6I: Why only quant Scr + siITGA11, not in VPSoe? It looks like there is still an increase in intracellular or fibril formation in VPSoe + siITGA11, which would be a key result to discuss.

We would like to clarify that 6I (now revised Figure 7I) is on the endocytosis of exogenous collagen-I, not quantification of Figure 6H.

Line 307: Discuss fibrillogenic sites, what are they?

As we have not shown direct evidence of VPS33B delivering endocytosed collagen at the site of fibrillogenesis, we have decided to alter the text to avoid overstatement, as suggested from previous reviewers’ comments.

Figure 8: What does pentachrome label?

Pentachrome staining allows for simultaneous staining of multiple species: collagen in red, sulphated mucopolysaccharides in violet, red blood cells in yellow, muscle in orange, nuclei in green.

Line 326: "In this study we have identified the endosome as a major protagonist in..." This is an overstatement and cant be drawn from this data.

We have modified this statement to “In this study we have identified an endocytic recycling mechanism for type I collagen fibrillogenesis that is under circadian regulation”

Line 330/331: "Collagen-I co-traffics with VPS33B in a VIPAS-containing endosomal compartment that directs collagen-I to sites of fibril assembly," This is also an overstatement that cannot be drawn from this data.

We have modified this statement to “Collagen-I co-traffics with VPS33B to the plasma membrane for fibrillogenesis”.

Line 340: again, the demonstration of the involvement of the endocytic pathway is very limited.

We have provided new evidence in the revised manuscript that support the involvement of classical endosomal compartments.

Line 366: You cant conclude this, you have not manipulated these proteins to show a functional effect or modulation of fibrillogenesis, it could still be a secondary effect.

We have provided new evidence in the revised manuscript that supports this conclusion.

Line 569: "Unless otherwise stated, incubation and washes were done at room temperature." Which incubations? Specify if this is just post-fixation during the EM prep or during cell culture.

This is specific to the EM preparation and we have clarified in the revised manuscript (revised line 663).

Small text alterations:

Overall we would like to thank the reviewer for highlighting these errors and mistakes in our manuscript, and have corrected them in our revised manuscript.

Figure 1E: Fluoro image series? This is only one image.

We wrote this to mean single channel images, we have corrected the terminology.

Line 111: Ref for Dyngo4a?

We have included this in the revised manuscript

Line 121: introduction/abbreviation definition for Fn1? Instead it is on Line 140.

Thank you for highlighting this, we have corrected this in revised manuscript.

Figure S2C: Alignment of labels cleaves x-axis.

We thank the reviewer for catching this and have corrected this with our revised manuscript.

Figure S4F and G should be inverted to mention sequentially in the text.

We thank the reviewer for catching this and have corrected this in our revised manuscript.

Line 182: Figure 4J should be G.

We thank the reviewer for catching this and have corrected this in our revised manuscript.

Line 209: typo: N-glycosylated.

We have corrected this typo in our revised manuscript.

Fig 6E: Very big as a figure element compared to others.

We have made this smaller in the revised manuscript to fit better with rest of the figure.

Line 313: Figure 7E not F.

Thank you for spotting this, we have corrected it.

Line 555: Typo: Scraped.

We have corrected this typo in our revised manuscript.

Line 562: missing

We have corrected this typo in our revised manuscript.

Standardise

We thank the reviewer for spotting the mistakes below and have corrected in our revised manuscript.

Legends: Include numbers of repeats and STATs throughout.Terminology: Dyng etc.Scale bars: some included as editable lines, some with size on top, small/large etc.

In certain cases we have positioned the scale bars in different regions of the figures to ensure no obscuring of the images.

VPS33b v B.
**Reviewer #2 (Recommendations For The Authors):**
The authors can improve the experimental part of the manuscript the following:- For all the western blots please include molecular weight markers.

We thank the reviewer for noticing this omission and have included molecular weight markers in the revised manuscript.

- Performing immunofluorescence and western blot analysis of endocytosed collagen -/+ inhibitors for lysosomal degradation (BafA1 or E64d+PepstatinA) in order to exclude endocytosis for degradation.

We thank the reviewer for this comment, another paper from the lab has identified lysosome to be involved in collagen fibrillogenesis (https://www.biorxiv.org/content/10.1101/2024.05.09.593302v1), thus

- Figure out how Dyngo4a is affecting Col1 secretion in the first place? Does it interfere with the secretory pathway. Alternatively, use a different model to block endocytosis (e.g. siRNA Dynamin).

We thank the reviewer for raising this. The Dyngo CM blot for total ponceau stain (revised Figure 2B) showed minimal changes, which suggest that global secretion is not affected.

- Further characterization of the VPS33B / collagen vesicles by immunofluorescence containing markers for early, late, and recycling endosomes. Block endocytic recycling by depletion of either Rabs or e.g. EHD1.

There are no good VPS33b antibody for staining. We have included images of GFP-tagged Rab5 co-localisation with labelled collagen-I (revised Figure 1D, Figure S6B).

- Further clarify the status of the VPS33B knockouts e.g. by sequencing. also provide a readout of the siRNA KD, besides the mRNA levels, since there the difference is not striking.

The knockout cell lines were characterised previously in our 2020 paper, which is referred to in our revised manuscript. We have always had issues detecting endogenous VPS33b due to reagents limitations, which is why we resorted to mRNA as the key readout.

- Doing siRNA knockdowns and endocytosis inhibition in the IPF fibroblasts to further strengthen the link between elevated expression of VPS33B/ ITGA11 and increased collagen uptake.

We thank the reviewer for suggesting these experiments. Due to limitations of the patient-derived fibroblasts (cell numbers and passage numbers) we had to prioritise experiments, and thus have performed siRNA against VPS33B and ITGA11 in the IPF fibroblasts. We showed that in both cases the amount of recycled labelled-collagen in collagen fibrils is significantly reduced (revised Figure 8D).

**Reviewer #3 (Recommendations For The Authors):**
Major points(1) Choice of cells: Please provide a rationale for why each cell line was used, and make sure that it is clear throughout the manuscript which cell line was used for each particular experiment. The HEK293T cell line is also missing from the reagent table.

We thank the reviewer for pointing out this omission, and have clarified in our revised manuscript which cell lines were used in each experiment. We used HEK293T to generate lentiviruses as described in the methods section.

(2) Missing information from methods. Experimental details are missing from the methods in several places, making it difficult for someone to replicate an experiment. For example, no details are given in the methods describing the explant culture of murine tail tendons (described in results lines 78100), and there are no details on how the skin samples were obtained or stained. Further, no ethical approval details are provided for the use of human skin tissue.

We apologise for leaving the ethical approval details and skin sample collection out, this was an oversight and will be included in the revised manuscript. We have also included the method to how murine tail tendons were cultured ex vivo (revised lines 527-531, 546-553).

(3) Functional studies in patient-derived cells. To fully establish the role of VPS33b and ITGA11 in fibrotic diseases, functional studies including the knockdown/overexpression of these genes could be performed to establish if the same response is seen as in non-diseased cells.

We agree that this will add much to the paper, and have performed siRNA against VPS33B and ITGA11 in the IPF fibroblasts. We showed that in both cases the amount of recycled labelled-collagen in collagen fibrils is significantly reduced (revised Figure 8D).

Minor Points

We thank the reviewer for pointing out these mistakes, and have corrected and included additional details in the revised manuscript.

(1) Lines 51-52. Wording of this sentence is unclear, please rephrase.(2) Line 182. Should this be Fig 4G rather than J?(3) Line 209. Correct spelling of glycosylated.(4) Line 463. Incomplete brackets and details missing?(5) Line 590. Correct tense - was rather than are.(6) Line 593. Specify centrifugation speed.(7) Line 619. Nuclei rather than nucleus.(8) Ln 650. Statistical analysis - was normality tested?(9) Figure 1e - Difficult to read labels for coll/DAPI.